# RETHINKING PRE-TRAINING IN TABULAR DATA: A NEIGHBORHOOD EMBEDDING PERSPECTIVE

## ABSTRACT

Pre-training is prevalent in deep learning for vision and text data, acquiring knowledge from other datasets to improve the downstream tasks. However, when it comes to tabular data, the inherent heterogeneity in the attribute and label spaces among datasets makes it hard to learn shareable knowledge and encode it in a model. We propose **Tab**ular data **P**re-**T**raining via **M**eta-representation (TABPTM), aiming to pre-train a general tabular model over a set of heterogeneous datasets. The key is to embed data instances from any dataset into a common feature space, in which an instance is represented by its distance to a fixed number of nearest neighbors and their labels. Such a meta-representation standardizes heterogeneous tasks into homogeneous local prediction problems, enabling training a model to infer the label (or the score to each possible label) of an input instance based on its neighborhood information. As such, the pre-trained TABPTM can be *directly applied to new datasets without further fine-tuning, regardless of their diverse attributes and labels*. Extensive experiments on 72 tabular datasets validate TabPTM's effectiveness (with and without fine-tuning) in both tabular classification and regression tasks.

## 1 INTRODUCTION

Pre-training has been the driving force for recent advances in artificial intelligence (Devlin et al., 2019; Liu et al., 2019; Kirillov et al., 2023; Dosovitskiy et al., 2021). Foundation models in vision and language all benefit from pre-training, using abundant images and documents collected from multiple sources (Zhou et al., 2023a; Oquab et al., 2023; Radford et al., 2021). Once trained, these models demonstrate remarkable generalizability to new tasks, even without fine-tuning. Such a success, however, has yet to be witnessed in tabular data, even though the data format is ubiquitous in many real-world applications such as financial prediction (Cao & Tay, 2001), recommendation system (Richardson et al., 2007), and healthcare (Ogunleye & Wang, 2020).

One fundamental challenge is the *inherent heterogeneity* in tabular data from different sources and tasks. A tabular dataset is typically represented by a matrix, with rows corresponding to instances and columns to attributes (features) (Borisov et al., 2022). Beyond this homogeneity, tabular datasets can be quite different in their dimensionality (*i.e.*, numbers of columns) and the semantic meaning of each dimension, even if they are for the same application. For example, different healthcare datasets can encode different granularities and aspects of patient information; even at the same feature entry (*e.g.*, the $d$-th column), the meaning is not necessarily the same (*e.g.*, "age" or "height"). This is drastically different from vision and text data (of the same language), in which different data sources typically share the same "vocabulary" (*e.g.*, pixel, patch, or sub-word definition) and similar relationships between vocabulary "elements" (*e.g.*, nearby pixels usually have similar colors). The lack of shared vocabulary and relationships in tabular data hampers the joint training of a model on multiple datasets, let alone the direct application of the pre-trained model to new downstream tasks.

> We thus hypothesize, *to pre-train a useful model for tabular data, one must first answer the question: What is the shareable vocabulary across datasets, and what transferable knowledge could be learned from these datasets?*

Some recent work addresses this by taking advantage of the semantic meanings of columns (*i.e.*, attributes). By transforming a data instance into the textual form (*e.g.*, "Age is 20, height is 170, ..."), one could tackle tabular data using language models (Hegselmann et al., 2023; Wang & Sun, 2022;

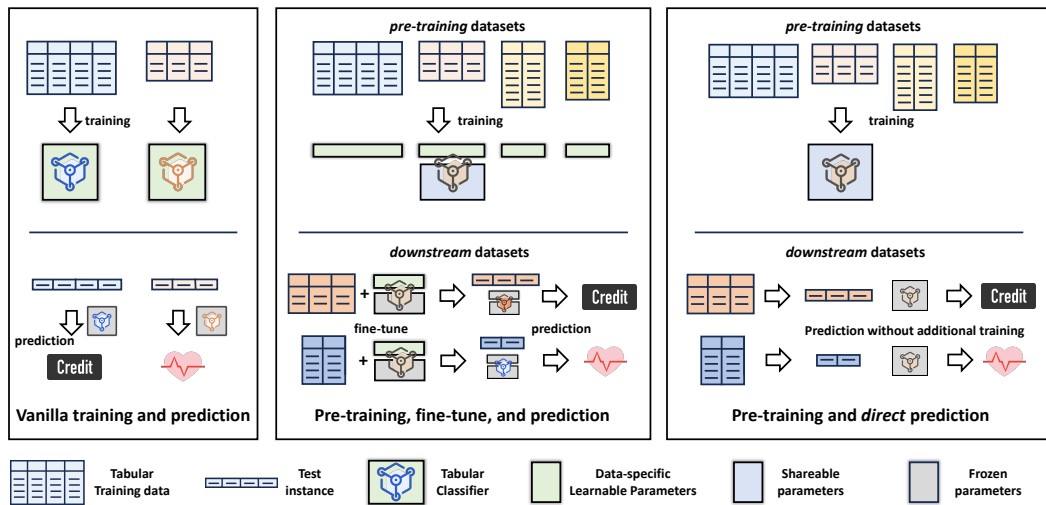

Figure 1: An illustration of TABPTM vs. other training strategies on tabular data. Left: the vanilla training and prediction pipeline, where tabular models are trained on each dataset separately. Middle: pre-training of a joint model on top of the learned dataset-specific token representations (of each attribute), in which the representation must be re-trained for downstream datasets. Right: TABPTM unifies heterogeneous tabular datasets via the meta-representation using neighborhood information, allowing pre-training a shareable model to predict labels based on such information, even without fine-tuning it on downstream datasets.

Liu et al., 2022a; Wang et al., 2023). When the semantic meanings are inaccessible or ill-defined — which is quite common in the real world, such as healthcare or measurement data — some approaches attempt to learn dataset-specific token representation for each attribute to transform different datasets into a shareable feature space (Iwata & Kumagai, 2020; Kumagai et al., 2022; Liu et al., 2022b; Wydmanski et al., 2023; Zhu et al., 2023). However, for new downstream tasks, these approaches must first re-train task-specific token representations before applying the pre-trained model.

Seeing the limitations of this line of approach, in this paper, we explore a novel direction to pre-train a general model for tabular data. We take inspiration from traditional work in *dimensionality reduction* and *manifold learning* and recent work in *comparing different neural network representations* to answer the question about **shareable vocabulary across heterogeneous tabular data**. In multidimensional scaling, regardless of what the original tabular datasets encode, as long as they lead to the same pairwise distance matrix between instances, they can be embedded into the same, low-dimensional tabular matrix (Davison & Sireci, 2000; Carroll & Arabie, 1998; Torgerson, 1952). Likewise, in manifold learning, a high-dimensional dataset can be re-represented by a lower-dimensional one if they encode the same neighborhood relationship (Weinberger et al., 2004; Yan et al., 2006; Van der Maaten & Hinton, 2008; Hinton & Roweis, 2002). Such cross-instance relationships are also used in recent work to analyze learned deep representations: two pre-trained neural networks encode similar knowledge if their features lead to similar affinity matrices among data instances, even if they are built upon different architectures (Kornblith et al., 2019; Huh et al., 2024). In short, **the relationships among instances** within a dataset could effectively serve as the shareable vocabulary across different tabular datasets, regardless of their dimensionality and semantic meanings.

**What transferable knowledge** can we learn on top of the shareable vocabulary, *i.e.*, the relationship among instances? We draw inspiration from the legendary *nearest-neighbor based machine learning* algorithms (Boiman et al., 2008; Wang et al., 2019; Sánchez et al., 1997; Chaudhuri, 1996; Zhang et al., 2006; Chao et al., 2013). The key insight in these approaches is that an instance's relationship to a particular label could be inferred from its relationship to the nearest neighbors (of that label).

Putting things together, we propose **Tab**ular data **P**re-**T**raining via **M**eta-representation (TABPTM), a general tabular model and pre-training approach over heterogeneous tabular datasets. TABPTM starts with representing a tabular data instance (*i.e.*, a row) *by the local context of its nearest neighbors in (the training set of) the dataset*. In this paper, we meta-represent each instance by a fixed-dimensional vector encoding the distances to a fixed number of nearest neighbors (ordered by distances) and their labels, while other methods may apply. Such a "meta-representation" standardizes heterogeneous

datasets, rendering instances into a *unifying* form of the same dimensionality and semantic meaning, and transforms heterogeneous prediction tasks into *homogeneous local prediction tasks*. TABPTM then trains a joint neural network on the meta-representations of a large number of datasets to learn to map the meta-representation — the neighborhood information — of each instance to the ground-truth label. Such an ability to *predict the label of a given instance in the local context* can then be transferred to new downstream tasks. Specifically, the pre-trained model by TABPTM can be directly applied to a downstream tabular dataset (represented by the meta-representation) without further fine-tuning, albeit that fine-tuning could further boost the performance. (See Figure 1 for a comparison of TABPTM with other existing strategies that train and deploy tabular models.)

We extensively validate TABPTM on 72 tabular datasets with various scenarios. The results demonstrate TABPTM's effectiveness in acquiring shareable knowledge over datasets to achieve promising generalizability on new classification and regression tasks. Our contributions are two-folded:

- We utilize meta-representations to reduce attribute heterogeneity and enable the pre-training of a general model over tabular datasets.
- Based on the meta-representations, the pre-trained TABPTM can directly generalize to new tabular datasets without further training, and achieve state-of-the-art accuracy in many datasets after fine-tuning, as evidenced by extensive experiments on both classification and regression tasks.

**Remark.**   Our TABPTM is conceptually related to TabPFN (Hollmann et al., 2023), which aims to learn a general in-context model for tabular classification datasets. Given a downstream dataset, TabPFN inputs the (whole) training set and each test instance into the pre-trained model, using the learned interactions between the test instances and the training ones to make predictions. Our TABPTM is particularly advantageous in three aspects. First, unlike TabPFN, which generates in-context tasks in every training step and feeds all the corresponding instances into the model at once, TABPTM pre-meta-represents each instance, enabling smaller batch sizes in training (hence more training-efficient). Second, a similar property applies to testing: TABPTM can pre-meta-represent every test instance and input only the meta-representations of the test instances into the model, hence more testing-efficient. Third, putting these two points together, Hollmann et al. (2023) claimed that their approach applies only to small datasets (with fewer than a thousand instances), while TABPTM can be trained on and applied to much larger tabular datasets.

## 2   RELATED WORK

This section outlines key methods related to our paper, with a more detailed discussion available in the appendix. Tabular data is one of the most common data forms in many fields (Richardson et al., 2007; Vanschoren et al., 2014; Hamidieh, 2018). Recent efforts have extended deep neural network successes to the tabular domain (Cheng et al., 2016; Wang et al., 2017; Gorishniy et al., 2021; Huang et al., 2020; Shwartz-Ziv & Armon, 2022; Grinsztajn et al., 2022), and explore the feasibility of general discriminative models that can adapt to a broad array of downstream datasets. The potential for in-distribution generalization of pre-trained tabular models has been demonstrated in contexts such as multi-task learning (Argyriou et al., 2006; Zhang & Yang, 2022; Rubachev et al., 2022; Luetto et al., 2023) and self-supervised learning (Ucar et al., 2021; Bahri et al., 2022), where data from various sources are in a consistent format. Some recent approaches utilize the deep neural network to pre-train a more generalizable tabular model, taking the difference in attributes and labels into account. The main difficulty lies in the ambiguity of shareable vocabulary and transferable knowledge across datasets. One representative kind of approach addresses this by harnessing the semantic meanings of column names (*i.e.*, attribute names) to transform instances into text, thereby leveraging large language models to enhance classification capabilities across diverse datasets (Liu et al., 2022a; Hegselmann et al., 2023; Zhang et al., 2023a; Wang et al., 2023). Concurrently, other researchers have focused on learning shared components, such as attribute-agnostic transformations, to provide effective initial weights for models adapting to new tasks (Iwata & Kumagai, 2020; Liu et al., 2022b; Zhang et al., 2023b; Shen et al., 2023; Zhu et al., 2023). Hollmann et al. (2023) addressed dataset heterogeneity by padding attributes across datasets into the same size and utilizing the contextual learning capabilities of transformers for classification tasks. Our proposed TABPTM standardizes heterogeneous datasets into a uniform format using meta-representation. This enables the effective pre-training of a model that can be directly applied to or minimally fine-tuned for downstream datasets without necessitating additional parameters.

## 3 Preliminary

**Learning with a single tabular dataset.** We denote a tabular classification dataset as $\mathcal{D} = \{(\boldsymbol{x}_i, y_i)\}_{i=1}^N$ which has $N$ examples (rows in the table) and $C$ classes. The label $y_i \in \{1, \ldots, C\}$ and its one-hot form is $\boldsymbol{y}_i \in \{0, 1\}^C$. The same discussion could be extended to the regression scenario where the label $y_i \in \mathbb{R}$. Each instance $\boldsymbol{x}_i \in \mathcal{X}$ is depicted by $d$ attributes (columns).[1] The goal is to estimate the posterior probability given an instance, *i.e.*, $\Pr(y_i \mid \boldsymbol{x}_i, \mathcal{D})$, often implemented by a model $f$ mapping from $\mathcal{X}$ to the label set $\mathcal{Y}$. The generalization ability of $f$ is measured by the prediction accuracy on an unseen instance sampled from the same distribution as $\mathcal{D}$.

**Pre-training across multiple tabular datasets.** Tabular datasets could be collected from various sources. Assume there are $T$ datasets $\mathbb{D} = \{\mathcal{D}_1, \ldots, \mathcal{D}_T\}$, in which the attribute and label spaces of the $t$-th dataset are denoted by $\mathcal{X}_t$ and $\mathcal{Y}_t$. Due to the inherent heterogeneity across datasets, we define the number of instances, attributes (*i.e.*, the dimension of an instance), and classes in dataset $\mathcal{D}_t$ as $N_t$, $d_t$, and $C_t$, respectively. Unlike multi-task or multi-view learning, which assumes one of $\mathcal{X}_t$ and $\mathcal{Y}_t$ is homogeneous (*i.e.*, identical) across all datasets, we consider *heterogeneous* datasets where the meanings of attributes and classes vary from one dataset to another. Regarding pre-training a model $f$ over $T$ heterogeneous datasets, we expect $f$ to deal with different attribute and label sets and generalize its discerning ability to an unseen downstream task $\mathcal{D}_u$. There are two primary challenges to pre-train $f$. The first is to determine the shareable "vocabulary" across datasets so that $f$ can learn from $\mathbb{D}$ and be applied to any heterogeneous dataset. The second is to identify and encode the "transferable knowledge" from $\mathbb{D}$ into $f$, enhancing the generalizability of $f$ such that it can improve an unseen downstream task.

There are two types of generalization. The first is the *direct* generalization, which means the learned $f$ is able to predict an unseen instance $\boldsymbol{x}_*^u$ in task $\mathcal{D}_u$ directly without additional training:

$$\hat{y}_*^u = f(\boldsymbol{x}_*^u \mid \mathcal{D}_u) , \tag{1}$$

where the label of $\boldsymbol{x}_*^u$ could be inferred via the shareable knowledge learned on the pre-training datasets. In addition, the learned $f$ can act as the initialization to be updated by several steps of gradient descent, minimizing the empirical risk over the target dataset $\mathcal{D}_u$. This *fine-tuning* strategy learns the task-specific property by fitting to $\mathcal{D}_u$. Due to the heterogeneity of datasets, additional parameters such as the feature tokenizer are introduced to adapt the model on downstream tasks (Zhu et al., 2023), which requires special tuning methods. Our goal is to develop a versatile general model $f$ that can be either directly applied to a target task or fine-tuned without introducing new parameters.

## 4 Method

Given the inherent heterogeneity in attribute and label spaces across various tabular datasets, the core idea behind TABPTM is to standardize these diverse datasets, enabling the application of a joint deep neural network. We begin by drawing inspiration from neighborhood-based methods and introduce the concept of meta-representation, which serves as the foundation for TABPTM's pre-training strategy. TABPTM applies to both classification and regression tasks, which is illustrated in Figure 2.

### 4.1 Motivation from Neighborhood Embedding

Accurate estimation of the posterior $\Pr(y_i \mid \boldsymbol{x}_i, \mathcal{D})$ is crucial for all classification and regression tasks (Murphy, 2022). Utilizing the balloon kernel density estimator (Terrell & Scott, 1992) with a uniform class prior, we can transform this posterior density into a neighborhood estimation problem, which calculates the probability via the relationship between $\boldsymbol{x}_i$ and its $K$ nearest neighbors in $\mathcal{D}$:

$$\Pr(y_i = c \mid \boldsymbol{x}_i, \mathcal{D}) = \frac{1}{K} \sum_{j \in \mathcal{N}_K(\boldsymbol{x}_i, \mathcal{D})} \mathbb{I}[y_j = c] . \tag{2}$$

$\mathbb{I}[\cdot]$ is the indicator function, and $\mathcal{N}(\boldsymbol{x}_i; \mathcal{D})$ denotes the set of $K$ nearest neighbors of $\boldsymbol{x}_i$ w.r.t. a specified distance $\text{dist}(\boldsymbol{x}_i, \boldsymbol{x}_j)$. Instead of considering all instances in $\mathcal{N}(\boldsymbol{x}_i; \mathcal{D})$ equally, a variant of

---

[1]We assume all attributes of an instance are numerical (continuous). If there exist categorical (discrete) attributes, we transform them into the one-hot forms in advance.

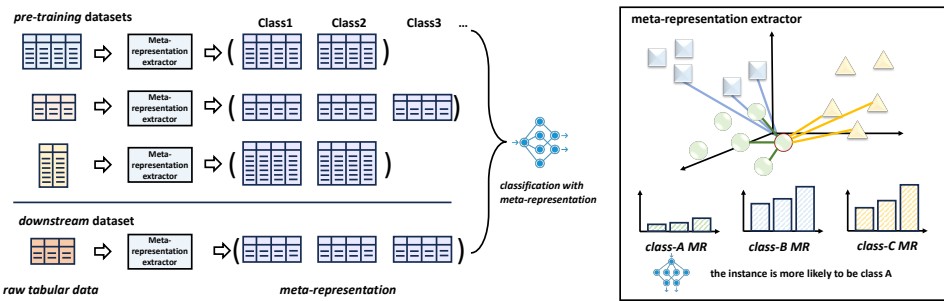

Figure 2: An illustration of the Meta-Representation (MR) for classification. MR transforms heterogeneous tabular datasets with different dimensions into a homogeneous form. A dataset has a set of $K$-dimension MRs, one for each class. The prediction scores for different classes could be obtained via MRs. We pre-train a joint model on MRs of different datasets and extend its generalization ability to downstream datasets. The right figure shows the MR of an instance, containing distances from an instance to the nearest neighbors of a certain class, characterizes the class membership patterns.

the estimator re-weights the influence of neighbors, allocating larger weight to closer neighbors and potentially enhancing prediction accuracy (Bishop, 2006; Rasmussen & Williams, 2006):

$$\Pr(y_i \mid \boldsymbol{x}_i, \mathcal{D}) = (\mathrm{softmax}\left([-\mathrm{dist}(\boldsymbol{x}_i, \boldsymbol{x}_1), \ldots, -\mathrm{dist}(\boldsymbol{x}_i, \boldsymbol{x}_K)]\right))^\top Y_K . \tag{3}$$

$Y_K \in \{0, 1\}^{K \times C}$ is the label matrix for $\mathcal{N}_K(\boldsymbol{x}_i; \mathcal{D})$, with each row corresponding to the one-hot label of $\boldsymbol{x}_i$. This formulation also extends to regression, and we will discuss details in Appendix B.

Equation 3 indicates that an instance's relationship to a particular label $c$ can be inferred from its relationship to the nearest neighbors associated with that label. This provides a *general* way to deal with heterogeneous datasets, as the relationships among instances serve as a shareable vocabulary across datasets, independent of dimensionality or semantic meaning. Thus, we can use these relationships to construct a meta-representation that encodes *transferable* knowledge.

### 4.2 META-REPRESENTATION OF AN INSTANCE

**Vanilla meta-representation.** We describe how to obtain the vanilla meta-representation for any instance in a tabular *classification* dataset $\mathcal{D}$ with $C$ classes, and later for the regression case. Based on the label of each instance, we partition the same-class instances in $\mathcal{D}$ into $C$ sets: $\mathcal{D}_{y=c} = \{(\boldsymbol{x}_i, y_i) \mid y_i = c\}, \forall c = 1, \ldots, C$. Given a class $c$, we calculate the distance between an instance $\boldsymbol{x}_i$ to instances in $\mathcal{D}_{y=c}$ (with $|\mathcal{D}_{y=c}|$ instances in total), and sort them in an *ascending* order:

$$\{\mathrm{dist}(\boldsymbol{x}_i, \boldsymbol{x}_1), \ldots, \mathrm{dist}(\boldsymbol{x}_i, \boldsymbol{x}_j), \ldots, \mathrm{dist}(\boldsymbol{x}_i, \boldsymbol{x}_{|\mathcal{D}_{y=c}|})\}$$

$$\mathrm{s.t.} \ \mathrm{dist}(\boldsymbol{x}_i, \boldsymbol{x}_1) \leq \ldots \leq \mathrm{dist}(\boldsymbol{x}_i, \boldsymbol{x}_j) \leq \ldots \leq \mathrm{dist}(\boldsymbol{x}_i, \boldsymbol{x}_{|\mathcal{D}_{y=c}|}) . \tag{4}$$

We then select the $K$ smallest distance values in the set, which constructs the local context with $K$ nearest neighbors for the instance $\boldsymbol{x}_i$. We define a mapping $\phi_c$ from $\boldsymbol{x}_i$ to its meta-representation $\phi_c(\boldsymbol{x}_i) \in \mathbb{R}^{2K}$ for the $c$-th class by:

$$\phi_c(\boldsymbol{x}_i) = [(\mathrm{dist}(\boldsymbol{x}_i, \boldsymbol{x}_1), \hat{y}_1), \ldots, (\mathrm{dist}(\boldsymbol{x}_i, \boldsymbol{x}_j), \hat{y}_j), \ldots (\mathrm{dist}(\boldsymbol{x}_i, \boldsymbol{x}_K), \hat{y}_K)] . \tag{5}$$

$\phi_c(\boldsymbol{x}_i)$ captures the neighborhood distribution of an instance, containing both the distance between an instance to the neighbors and the corresponding labels of the neighbors. $\hat{y}_j$ in Equation 5 provides auxiliary label information for $\boldsymbol{x}_i$, for example, the semantic vector of the label. In our implementation, we set $\hat{y}_j$ in a one-vs.-rest manner based on its true label $y_j$, which means $\hat{y}_j = 1$ if $y_j = c$, or $\hat{y}_j = -1$ otherwise. This sparse implementation indicates the relationship between $\boldsymbol{x}_j$ and class $c$. In our experiments, we will demonstrate if we use richer supervision from a stronger tabular model's prediction, the quality of the meta-representation as well as the discriminative ability of TABPTM can be further improved.

$\phi_c(\boldsymbol{x}_i)$ reveals the membership of $\boldsymbol{x}_i$ to a particular class based on its neighborhood context. If an instance resides within a high-density region of a class (akin to being near the class center), the majority of values in $\phi_c(\boldsymbol{x}_i)$ would typically be small, indicating proximity to neighboring instances of that class. Conversely, if only a few values in $\phi_c(\boldsymbol{x}_i)$ are small, while most are large, it indicates that the instance $\boldsymbol{x}_i$ unlikely belongs to class $c$. In sum, the heterogeneous tabular classification tasks

become homogeneous prediction tasks over the local context when we take advantage of $\phi_c(\boldsymbol{x}_i)$. No matter what value the original dimension $d$ of $\boldsymbol{x}_i$ is, $\phi_c(\boldsymbol{x}_i)$ has a fixed dimension with value $K$, standardizing the vectors and facilitating pre-training over heterogeneous tabular datasets. We set $\Phi(\boldsymbol{x}_i) = \{\phi_c(\boldsymbol{x}_i)\}_{c=1}^C$ as the meta-representation for the instance $\boldsymbol{x}_i$ given the dataset.

In the *regression* case, labels in $\mathcal{D}$ are scalars. We obtain the $K$ nearest neighbors from the whole dataset, and use the similar form as Equation 5, *i.e.*,

$$\Phi(\boldsymbol{x}_i) = \left[(\text{dist}(\boldsymbol{x}_i, \boldsymbol{x}_1), y_1), \ldots, (\text{dist}(\boldsymbol{x}_i, \boldsymbol{x}_K), y_K)\right] . \tag{6}$$

This $K$-dimensional meta-representation in the regression scenario depicts the relationship between $\boldsymbol{x}_i$ to the neighbors.

**Metric-based meta-representation.** The distance measure $\text{dist}(\cdot, \cdot)$ in Equation 4 plays an important role when making decisions via the relationship between a given instance and others (Schölkopf et al., 2001; Schölkopf & Smola, 2002). Yet, in the presence of high-dimensional features (large $d$), relying on all attributes becomes computationally challenging. Moreover, the distance might be unduly influenced by redundant or noisy attributes. To address this, we implement a distance metric over raw attributes, which ensures that our final meta-representation accurately captures both the properties of individual instances and the dataset.

The main challenge lies in designing an adaptive metric compatible with heterogeneous tasks. In this paper, we first formulate the distance measure in the following form (more distances are investigated in Appendix D):

$$\text{dist}(\boldsymbol{x}_i, \boldsymbol{x}_j) = \left(\sum_{l=1}^{d} w_l \cdot |\boldsymbol{x}_{il} - \boldsymbol{x}_{jl}|^p\right)^{\frac{1}{p}} , \tag{7}$$

where $\boldsymbol{x}_{il}$ denote the $l$-th dimension of $\boldsymbol{x}_i$. We set $p \in \{1, 2\}$ and $w_l > 0$ is a weight for each dimension. When $w_l = 1$, the distance in Equation 7 degenerates to Euclidean distance ($p = 2$) or Manhattan distance ($p = 1$). Given the training set $\mathcal{D} = \{\boldsymbol{X}, \boldsymbol{Y}\}$ of a dataset where $\boldsymbol{X}$ and $\boldsymbol{Y}$ denote the instance matrix and label vector, respectively, we derive feature weights from the mutual information shared between individual attributes and their labels

$$w_l = \text{normalize}\left(\text{MI}(\boldsymbol{X}_{:l}, \boldsymbol{Y})\right) . \tag{8}$$

$\boldsymbol{X}_{:l}$ is the $l$-th column of $\boldsymbol{X}$, *i.e.*, the vector containing values of the $l$-th attribute of all instances. $\text{MI}(\cdot, \cdot)$ calculates the mutual information between two sets, which measures the dependency between an attribute and the labels (Brown et al., 2012). The larger the mutual information, the more important an attribute is, so that we increase its weight in Equation 7. The $\text{normalize}(\cdot)$ normalizes input values by dividing their cumulative sum. The experiments validate that integrating this distance metric in meta-representation significantly enhances the model's generalization ability.

**Meta-representation in the few-shot scenario.** The previously discussed meta-representation assumes there are at least $K$ neighbors in the set $\mathcal{D}_{y=c}$. However, in some applications like few-shot classification or the existence of minority class, $\mathcal{D}_{y=c}$ might only contain a limited number of neighbors smaller than $K$. To address the data scarcity challenge, we pad the meta-representation with its last value (the largest distance) (Yang & Gopal, 2012).

## 4.3 A Pilot Study on Meta-Representation

Based on the definition of meta-representation, we use classification as an example to show its discriminative ability and demonstrate it is shareable across different datasets.

We consider two datasets, *i.e.*, "breast-cancer-wisc" (binary) and "dermatology" (multi-class). For each dataset, we calculate the meta-representation $\Phi(\boldsymbol{x}_i)$ for all classes in their datasets. We partition the meta-representation for the target class ($\phi_{y_i}(\boldsymbol{x}_i)$), and the ones for non-target classes ($\phi_{c \neq y_i}(\boldsymbol{x}_i)$) into two sets. We set $K = 8$. As shown in Figure 3, we use red/blue to denote the previous two types of meta-representation, respectively, and use different shapes to differentiate datasets.

We have several observations. First, the meta-representation w.r.t target and non-target classes are separated (as shown by the dotted line), which indicates it is possible to determine the target class of an instance by discerning the target meta-representation. Then, the meta-representations for different datasets are mixed, which means meta-representation is able to be a general strategy for

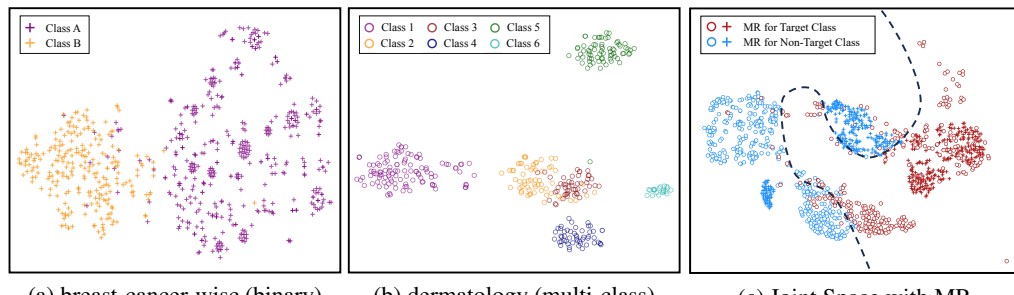

(a) breast-cancer-wisc (binary)     (b) dermatology (multi-class)     (c) Joint Space with MR.

Figure 3: Pilot study of Meta-Representation (MR) over two datasets "breast-cancer-wisc" (binary, denoted by "+") and "dermatology" (multi-class, denoted by "∘"). We use colors to distinguish between different classes. (a) and (b) display the unique characteristics of each dataset. In (c), we homogenize the datasets using MR, with red to indicate the MR for the target class ($\phi_{y_i}(\boldsymbol{x}_i)$) and blue for non-target classes ($\phi_{c \neq y_i}(\boldsymbol{x}_i)$, those not matching the true label). MR effectively integrates both datasets into a joint space where their classifications can be implemented by the dotted line.

extracting shareable knowledge across datasets. Moreover, different datasets have diverse patterns w.r.t. their meta-representations. For example, the target-class meta-representations denoted by "+" are clustered while those denoted by "∘" are not. Thus, joint training on multiple datasets to predict via meta-representations is necessary.

## 4.4 MAKING PREDICTIONS VIA META-REPRESENTATION

**Classification**. Given a dataset $\mathcal{D}$, we represent an instance $\boldsymbol{x}_i$ with $\Phi(\boldsymbol{x}_i) = \{\phi_c(\boldsymbol{x}_i)\}_{c=1}^C$. Based on the meta-representation, we need to obtain the prediction score for each class in a classification task. Define the score for each class as

$$[s(\boldsymbol{x}_i)_1, \ldots, s(\boldsymbol{x}_i)_C] = \mathbf{T}_\Theta \left( [\phi_1(\boldsymbol{x}_i), \ldots, \phi_C(\boldsymbol{x}_i)] \right) . \tag{9}$$

$\mathbf{T}_\Theta$ is a transformation that captures the class membership patterns from the meta-representation for each class and then outputs the corresponding class-wise classification scores based on the local neighborhood context. $\Theta$ denotes the learnable parameters in $\mathbf{T}$. In TABPTM, we implement $\mathbf{T}$ with Multi-Layer Perceptron (MLP), *i.e.*,

$$s(\boldsymbol{x}_i)_c = \mathbf{MLP}(\phi_c(\boldsymbol{x}_i)), \ \forall c = 1, \ldots, C . \tag{10}$$

The parameters of MLP are shared for all classes, whose detailed architecture is described in Appendix C. When multiple types of distances are used, we concatenate them together at first and then use Equation 10 to map the concatenated meta-representation vectors to a scalar.

Based on the scores, the predicted class for $\boldsymbol{x}_i$ is

$$\hat{y}_i = \arg\max_c \ \{s(\boldsymbol{x}_i)_1, \ldots, s(\boldsymbol{x}_i)_C\} . \tag{11}$$

The classification strategy based on meta-representation fits heterogeneous tasks with different attributes and class spaces. The meta-representation-based classification enables the usage of a joint model over heterogeneous tasks.

**Regression**. Meta-representation could be applied to the tabular regression tasks in a similar manner. Since the label is a continuous value, we map the meta-representation to a scalar

$$s(\boldsymbol{x}_i) = \mathbf{T}_\Theta \left( \Phi(\boldsymbol{x}_i) \right) . \tag{12}$$

$\mathbf{T}_\Theta$ captures the continuous values in the neighborhood and sets the prediction for $\boldsymbol{x}_i$ as a weighted combination of its neighbor's labels. We also implement $\mathbf{T}$ with MLP with learnable parameter $\Theta$.

## 4.5 PRE-TRAINING WITH META-REPRESENTATION

Based on the previous discussions, we pre-train a joint model, *i.e.*, the transformation $\mathbf{T}_\Theta$, over $\mathbb{D}$, whose parameters are shared across multiple seen tabular datasets.

$$\min_\Theta \sum_{t=1}^T \sum_{i=1}^{N_t} \ell \left( \mathbf{T}_\Theta(\Phi(\boldsymbol{x}_i^t)), \ y_i^t \right) . \tag{13}$$

Table 1: Average RMSE on 18 downstream regression datasets. The best results are in bold, and the second-best results are underlined. The value beside the dataset name denotes the scale of the results. For example, $\times 10$ means all results should be multiplied by 10. "OOM" means out-of-memory.

| $\downarrow$ | TABPTM | SVM | $K$NN | XGBoost | MLP | FT-T | TabR | SAINT | TANGOS | TabNet | PTaRL | XTab |
|---|---|---|---|---|---|---|---|---|---|---|---|---|
| SPP$_{\times 10}$ | .1648 | .2080 | .3240 | .1550 | .1960 | **.1430** | .1790 | .1970 | .2040 | .4670 | .3610 | .4740 |
| HPP$_{\times 10^7}$ | .1074 | .1070 | .1130 | .1060 | .1110 | **.1050** | .1100 | .1060 | .1130 | .1400 | .1100 | .1580 |
| STO | **.7355** | .9510 | .7950 | .8910 | .9410 | .9880 | .9660 | .7810 | .8030 | 1.060 | 1.180 | 4.920 |
| CDA$_{\times 10^3}$ | .4956 | .6980 | .5490 | **.4910** | .5910 | .6610 | .5970 | .5380 | .5890 | .6350 | .5560 | .7440 |
| PMS$_{\times 10^2}$ | .1534 | .1600 | .1600 | .1630 | **.1520** | .1630 | .1730 | .1720 | .1540 | .1670 | .1630 | .1720 |
| GCB | .4828 | .4880 | .5040 | **.4440** | .4870 | .4840 | .4840 | .4860 | .4840 | .4880 | .4840 | .5020 |
| ADO$_{\times 10^3}$ | **.7374** | .7960 | .7880 | .7750 | .7720 | .7710 | .7730 | .7730 | .7810 | .8200 | .7670 | 1.300 |
| CAC | **.1322** | .1350 | .1470 | .1430 | .1340 | .1350 | .1450 | OOM | .1390 | .1590 | .1380 | .1780 |
| AVE$_{\times 10^3}$ | 3.080 | 6.390 | 4.680 | **.6980** | 9.770 | 9.940 | 9.930 | 9.920 | 9.810 | 3.310 | 3.480 | 9.980 |
| DBT$_{\times 10^{-1}}$ | .6617 | 1.110 | .7070 | .7840 | .8530 | .7360 | **.6230** | .6390 | .6880 | 1.030 | .9600 | 1.450 |
| ASU$_{\times 10^{-1}}$ | .9036 | 1.860 | 1.080 | **.8420** | 1.010 | .9380 | .9040 | 1.010 | 1.070 | 1.110 | 1.290 | 3.020 |
| VNV$_{\times 10^{-1}}$ | 1.015 | 1.020 | 1.070 | 1.030 | **.9880** | 1.000 | 1.030 | 1.020 | 1.020 | 1.010 | .9910 | 1.010 |
| SFA$_{\times 10^2}$ | .1830 | .1890 | .2010 | **.1670** | .2020 | .1810 | .1840 | .2000 | .2000 | .2310 | .2190 | .1870 |
| PNH$_{\times 10}$ | .3318 | .3380 | .3860 | .3400 | .3320 | **.3270** | .3300 | .3340 | .3370 | .3330 | .3350 | .3830 |
| PUH$_{\times 10^{-2}}$ | .7804 | 2.560 | 2.780 | .9380 | 1.300 | .7760 | **.6070** | .8810 | .8920 | .9190 | 1.610 | 2.860 |
| SFU$_{\times 10^{-1}}$ | .2032 | .3480 | .2210 | .2540 | .2390 | .2550 | .2310 | **.1890** | .2220 | .2980 | .3080 | .3690 |
| SHP | .4318 | .4930 | .4840 | **.4290** | .4460 | .4390 | .4320 | .4530 | .4510 | .4340 | .4340 | .4400 |
| SAC$_{\times 10}$ | **.3983** | .4010 | .4170 | .4330 | .4470 | .4490 | .4510 | .4510 | .4510 | .4920 | .4320 | .4380 |
| rank | **2.444** | 7.833 | 7.667 | 4.500 | 5.833 | 4.833 | 5.667 | 5.588 | 6.333 | 8.556 | 6.556 | 10.278 |

The transformation $\mathbf{T}_\Theta$, pre-trained across $T$ datasets, links the meta-representation to the final score. The loss $\ell(\cdot, \cdot)$ measures the discrepancy between a prediction and the label, and we set it as cross-entropy for classification and mean square error for regression. Given a downstream dataset $\mathcal{D}_u$, we first obtain the meta-representation for each instance. Then the learned $\mathbf{T}_\Theta$ could be applied directly without further training, or acts as a good initialization for fine-tuning without additional parameters. Appendix C provides more details of TABPTM, including the pre-training and deployment workflows of TABPTM in Algorithm 1 and Algorithm 2, respectively.

In summary, we treat the relationships among instances as a form of shareable vocabulary. Meta-representation leverages these relationships between an instance and its nearest neighbors, transforming the tabular prediction task into inferring an instance's label based on its relationship to neighbors of that label. TABPTM not only standardizes heterogeneous datasets but also facilitates the learning of transferable knowledge.

## 5 EXPERIMENTS

We validate TABPTM on 72 open-source real-world tabular datasets from various fields (36 for classification and 36 for regression), including medical, software, and speech recognition. We analyze TABPTM as well as meta-representation in subsection 5.2 and Appendix E.

### 5.1 SETUPS

**Datasets**. Each 36 datasets for classification/regression are split into two parts: one is used as seen datasets for pre-training, and another part is used as downstream datasets. We show the results of one partition in the main paper and the results we exchange two partitions in Appendix E. For each dataset, we randomly sample 80% of them as the training set, and the remaining 20% instances are used for test. In the training set, we randomly hold out 20% of training instances as the validation set.

**Evaluation criteria**. After pre-training on all seen datasets, we evaluate the classification accuracy and root mean square error (RMSE) of the model on downstream datasets for classification and regression tasks, respectively. We use "TABPTM" to denote the method that we use the pre-trained weights as the initialization and *fine-tune* the model on a downstream dataset without introducing additional learnable parameters. The variant making predictions *directly* without additional training

Table 2: Average accuracy on 18 downstream classification datasets. The best results are shown in bold, and the second-best results are underlined.

| ↑ | TABPTM | SVM | XGBoost | MLP | FT-T | TabR | SAINT | TANGOS | TabNet | TabPFN | XTab | DEN |
|---|---|---|---|---|---|---|---|---|---|---|---|---|
| BAA | **59.40** | 53.10 | 55.40 | 54.10 | 53.80 | 55.60 | 55.50 | 54.70 | 54.60 | 54.60 | 49.30 | 45.90 |
| BCC | 85.30 | 86.50 | 86.80 | 85.70 | 86.50 | 85.80 | **87.10** | 86.40 | 86.00 | 78.90 | 85.00 | 80.10 |
| BCP | 82.00 | 75.00 | 84.70 | 79.00 | 83.20 | **87.30** | 83.00 | 76.30 | 82.30 | 74.00 | 77.70 | 75.00 |
| BCW | **97.50** | 97.10 | 97.10 | 96.60 | 95.80 | 97.10 | 95.90 | 88.40 | 96.90 | 95.10 | 97.30 | 97.10 |
| BLO | 78.00 | 74.70 | 76.90 | 77.70 | 77.90 | 77.80 | 75.40 | **78.20** | 77.20 | 76.40 | 75.70 | 71.20 |
| BRC | 67.10 | 55.20 | **69.00** | 62.60 | 65.90 | 66.80 | 67.50 | 67.00 | 67.00 | 66.70 | 65.60 | 63.40 |
| CDH | 75.20 | 75.10 | 75.10 | 75.20 | 75.30 | 74.20 | 75.00 | 75.20 | 75.00 | 75.00 | **86.70** | 70.20 |
| DER | 99.20 | 98.60 | 99.00 | 98.50 | **99.50** | 98.60 | 98.40 | 98.80 | 98.90 | 89.50 | 80.00 | 97.70 |
| ECH | 81.20 | 77.80 | 75.60 | 79.80 | 74.10 | 78.00 | 80.20 | 70.40 | 76.50 | 69.90 | **84.40** | 80.30 |
| ECS | 67.40 | **75.10** | 67.80 | 67.20 | 67.80 | 66.20 | 67.70 | 67.80 | 67.10 | 67.20 | 66.30 | 72.90 |
| FCC | 77.30 | 76.70 | **79.20** | 78.80 | 78.00 | 77.90 | 77.60 | 77.60 | 78.60 | 77.00 | 72.80 | 71.00 |
| HEC | 52.30 | **55.70** | 50.60 | 52.20 | 51.90 | 53.80 | 53.90 | 51.90 | 52.00 | 49.80 | 51.30 | 47.50 |
| KC2 | 83.10 | 81.90 | 79.60 | **84.50** | 79.10 | 82.00 | 77.80 | 81.60 | 83.10 | 79.70 | 84.40 | 82.60 |
| MHR | 68.70 | 67.00 | **81.90** | 73.20 | 69.60 | 73.80 | 78.10 | 74.10 | 71.40 | 58.80 | 54.30 | 56.80 |
| SAF | 81.90 | 83.50 | 83.20 | 81.80 | 80.60 | **84.60** | 83.60 | 82.90 | 80.30 | 79.70 | 68.80 | 78.30 |
| SEB | 92.80 | 92.80 | 92.70 | 92.90 | 92.90 | 92.80 | **93.00** | 92.90 | 92.80 | 92.90 | 92.80 | 92.10 |
| TSE | 48.30 | 49.20 | **51.90** | 49.70 | 51.20 | 49.20 | 50.20 | 50.40 | 49.30 | 49.20 | 46.70 | 31.50 |
| WAQ | **91.10** | 89.80 | 89.40 | 89.40 | 88.70 | 88.80 | 89.60 | 89.60 | 89.20 | 88.40 | 88.00 | 88.20 |
| rank | **4.556** | 6.333 | 4.611 | 5.722 | 5.667 | 5.444 | 4.889 | 5.278 | 6.111 | 9.056 | 8.278 | 9.500 |

will be analyzed in the appendix. The average accuracy over 10 random seeds is reported (Gorishniy et al., 2021). We also use the average rank over all datasets to compare the ability of different methods following Delgado et al. (2014); McElfresh et al. (2023).

**Comparison methods**. We compare TABPTM with three types of methods. First are the classical methods such as Support Vector Machine (SVM) and XGBoost (Chen & Guestrin, 2016). The second part contains deep tabular models, such as Multi-Layer Perceptron (MLP) (Kadra et al., 2021), FT-Transformer (FT-T) (Gorishniy et al., 2021), TabCaps (Chen et al., 2023a), and TabR Gorishniy et al. (2024). The third part involves methods that make predictions with a pre-trained model, such as XTab (Zhu et al., 2023), DEN (Liu et al., 2022b), and TabPFN (Hollmann et al., 2023).

**Implementation details**. We implement our model, *i.e.*, $\mathbf{T}_\Theta$ with a three-layer MLP. During the pre-training, we set the learning rate as 0.001 and randomly sample 1024 examples from a seen dataset in each iteration. For the first two groups of comparison methods, we tune their hyper-parameters and carry out early stopping on the corresponding validation set of a given dataset. For the last type of comparison methods and ours, we use a model's average accuracy/RMSE over all validation sets of the seen datasets to tune the hyper-parameters. For TABPTM, we fine-tune the pre-trained weights on a downstream dataset with a learning rate of 0.01 and 30 epochs. The model in the final epoch is utilized for evaluation. We set $K = 16$ for regression and $K = 128$ for classification.

## 5.2 GENERALIZATION ABILITY OF THE PRE-TRAINED MODEL

**Results on regression tasks**. The average RMSE results are listed in Table 1. After pre-training on 18 regression datasets, TABPTM not only outperforms representative deep tabular models like TabR but also outperforms the boosting method XGBoost. The deep tabular baseline MLP is trained over raw features for each downstream dataset separately. Although our TABPTM also adopts the MLP architecture, it is clear that the pre-trained model extracts the shareable knowledge across datasets and with better generalization ability. XTab learns a joint transformer module during pre-training, and the remaining parameters of a model are further fine-tuned for each downstream task. Our TABPTM outperforms XTab on most datasets without introducing additional learnable parameters. In summary, TABPTM provides an efficient and effective solution for tabular regression.

**Results on classification tasks**. The average comparison accuracy results are reported in Table 2. Methods like XGBoost and XTab get good results on certain datasets, but our TABPTM obtains good performance in general (with the best average rank). In summary, since the main computational burden of TABPTM comes from the nearest neighbor search, MLP inference, and fine-tuning with

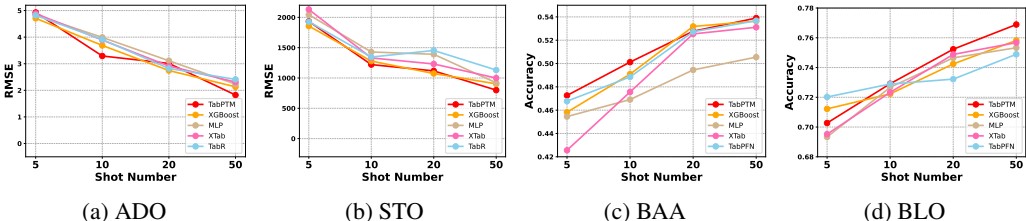

(a) ADO        (b) STO        (c) BAA        (d) BLO

Figure 4: Average RMSE or accuracy of few-shot regression over multiple trials. For each downstream dataset, $\{5, 10, 20, 50\}$ examples per class (shot) are randomly sampled as the training set.

Table 3: Comparison between variants of XGBoost and TABPTM on two regression datasets (STO and ADO with RMSE criterion) and two classification datasets (BAA and BLO with accuracy results). $\text{XGBoost}_{\text{MR}}$ denotes we train XGBoost on meta-representation. $\text{TABPTM}_{\text{S}}$ means we train the top-layer model in TABPTM directly without leveraging the pre-trained model.

|  | XGBoost | $\text{XGBoost}_{\text{MR}}$ | $\text{TABPTM}_{\text{S}}$ | TABPTM |
|---|---|---|---|---|
| STO ↓ | 0.8910 | 0.7809 | 0.7692 | 0.7355 |
| ADO ↓ | 775.0 | 749.1 | 758.3 | 737.4 |
| BAA ↑ | 55.40 | 57.91 | 56.82 | 59.40 |
| BLO ↑ | 76.90 | 77.70 | 76.40 | 78.00 |

a limited number of gradient descent, the overall inference time cost of TABPTM is very low. So our TABPTM makes *fast yet accurate* predictions, which could be a good choice when the model deployment efficiency is emphasized in some real-world applications.

### 5.3 ABLATION STUDIES

**Few-Shot Learning Ability of TABPTM**. We further investigate whether TABPTM also keeps its superiority when the training set is very small, on both regression and classification tasks. In this few-shot evaluation, we show the RMSE/accuracy change when the number of instances per class (shot) increases in $\{5, 10, 20, 40\}$. We mainly compare with XGBoost, the few-shot tabular model TabPFN (only works for classification tasks), and the pre-training tabular approach XTab. Due to the limited number of training instances, we use their default hyper-parameters. The few-shot results of four datasets are shown in Figure 4. Results indicate that TABPTM outperforms others in most cases, verifying its few-shot generalization ability.

**The Role of the Shared Top-Layer Model**. We investigate whether the MLP learned in the pre-training stage indeed encodes shareable knowledge and facilitates the downstream prediction tasks. We consider two comparison variants. First, we train XGBoost on the meta-representation of a given dataset, which is denoted by $\text{XGBoost}_{\text{MR}}$. Then, we also train the top-layer MLP in TABPTM directly without leveraging the pre-trained model, which is denoted by $\text{TABPTM}_{\text{S}}$. The results are listed in Table 3. We find $\text{XGBoost}_{\text{MR}}$ outperforms vanilla XGBoost, which means the proposed meta-representation helps both classification and regression. By comparing TABPTM and $\text{TABPTM}_{\text{S}}$, we find TABPTM achieves better performance, which indicates the pre-trained top-layer MLP indeed encodes shareable knowledge that helps the downstream tasks. The superiority of TABPTM over $\text{XGBoost}_{\text{MR}}$ further validates its strong generalization ability. The additional ablation studies discussed in subsection E.5 further validate that the pre-trained model successfully encodes shareable knowledge, significantly enhancing downstream tabular prediction capabilities.

## 6 CONCLUSION

Considering the large amount of heterogeneous tabular datasets in many machine learning fields, we explore a way to pre-train a general model and extend its classification or regression ability to downstream datasets. We address the primary challenge of disparate attribute and label spaces across datasets via meta-representation. Our pre-trained TabPTM can be directly applied to unseen datasets or be fine-tuned without modifying the architecture. The model achieves competitive performance in various scenarios, which validates its effectiveness.

## 7 REPRODUCIBILITY STATEMENT

We use CPU: Intel(R) Xeon(R) Gold 5218R CPU @ 2.10GHz and GPU: 4 × NVIDIA RTX 6000 Ada Generation in our experiments. The code is available at `https://anonymous.4open.science/r/TabPTM-code/`.

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

There are five parts in the appendix:

- Appendix A: Additional related methods;
- Appendix B: Motivation of meta-representation from nearest neighbor model;
- Appendix C: More details and discussions of our TABPTM approach;
- Appendix D: Details of the experimental setups;
- Appendix E: Analysis of meta representation and additional ablation studies on TABPTM.
- Appendix F: Experimental results of main tables including standard deviation;

## APPENDIX A    ADDITIONAL RELATED METHODS

**Learning with tabular data**. Tabular data is one of the most common data forms in many fields (Richardson et al., 2007; Vanschoren et al., 2014; Hamidieh, 2018), and a lot of classical machine learning methods have been developed for tabular data, such as XGBoost (Chen & Guestrin, 2016), LightGBM (Ke et al., 2017), and CatBoost (Prokhorenkova et al., 2018). Recently, researchers have tried to extend the success of deep neural networks such as multi-layer perceptron (Gorishniy et al., 2021), Transformer (Huang et al., 2020), and diffusion models (Kotelnikov et al., 2023) from visual and textual domains to the tabular fields (Borisov et al., 2022). Attribute embeddings (Song et al., 2019) and deep architectures have been designed (Guo et al., 2017; Katzir et al., 2021; Chen et al., 2023b) for tabular data, and some simple baselines can achieve competitive results as the classical methods after carefully tuned (Kadra et al., 2021; Jeffares et al., 2023). Deep tabular models have the flexibility for various scenarios and can be incorporated well with the classical methods (Cheng et al., 2016; Wang et al., 2017; Shwartz-Ziv & Armon, 2022; Grinsztajn et al., 2022).

**Reuse a heterogeneous tabular model**. Instead of training a tabular model from scratch on a new task, reusing a pre-trained model from a related task becomes a useful choice, especially when efficiency is emphasized (Tommasi et al., 2014; Kuzborskij & Orabona, 2017; Aghbalou & Staerman, 2023). In addition to the distribution shift between the pre-trained and the target tasks, the changes in their attribute spaces as well as the label spaces make the transfer of a tabular model challenging (Hou & Zhou, 2018; Ye et al., 2021). Tabular model reuse across heterogeneous datasets usually relies on some assumptions, *e.g.*, the existence of a set of overlapped attributes between two datasets (Hou et al., 2022; Levin et al., 2023; Onishi et al., 2023; Zhou et al., 2023b) and the column meanings (the textual names of all attributes) (Wang & Sun, 2022).

**Learning with multiple tabular datasets**. One more step to take advantage of the ability of deep neural networks on tabular fields is to pre-train a discriminative model on a large number of tabular datasets and extend its ability to downstream tasks. The in-distribution generalization ability of a pre-trained tabular model has been validated in multi-task learning (Argyriou et al., 2006; Zhang & Yang, 2022; Rubachev et al., 2022; Luetto et al., 2023) and self-supervised learning (Ucar et al., 2021; Bahri et al., 2022), where all datasets are collected in a homogeneous form. In multi-view learning, a model is required to capture the consistent nature among heterogeneous views, but those multiple views of an instance are paired and share the same label space (Xu et al., 2013). Some recent approaches utilize the deep neural network to pre-train a more generalizable tabular model, taking the difference in attributes and labels into account. One representative kind of approach assumes the existence of attribute names along with a dataset so that each instance could be transformed into a text, then a large language model could be applied to generalize the classification ability (Liu et al., 2022a; Hegselmann et al., 2023; Zhang et al., 2023a; Wang et al., 2023). Another thread of method learns shared components such as attribute-agnostic transformation across datasets, which provides a good model initialization for partial parameters given a downstream task (Iwata & Kumagai, 2020; Liu et al., 2022b; Zhang et al., 2023b; Shen et al., 2023; Zhu et al., 2023). We propose TabPTM to transform all datasets into a uniform form with meta-representation to enable the pre-training. Then, the pre-trained model could be applied directly to a downstream dataset, or fine-tuned without introducing additional parameters.

**Meta-Representation**. The notion of meta-representation has been previously leveraged in multi-label learning to express the relationship between an instance and a specific label, which facilitates decoupling the label correlations (Yang & Gopal, 2012; Zhang & Wu, 2015). The meta-representation is also used together with the raw tabular features to assist text classification (Canuto et al., 2014;

2018). In this paper, we provide a more general form of meta-representation, which contains both the distance values as well as the label information, working in both classification and regression tasks. Moreover, we employ meta-representation as a pivotal tool to construct a pre-trained model that can effectively operate across multiple heterogeneous tabular datasets. We also emphasize the metric-based variant and the strategies to deal with few-shot scenarios when a deep neural network is pre-trained over the meta-representations.

## APPENDIX B    META-REPRESENTATION FROM THE NEAREST NEIGHBOR PERSPECTIVE

As mentioned in Equation 2 in subsection 4.1, we can formulate the problem of estimating the posterior density $\Pr(y_i \mid \boldsymbol{x}_i, \mathcal{D})$ in the nearest neighbor form, and then represent the estimator with meta-representation.

Given an instance $\boldsymbol{x}_i$, $K$NN calculates the distance between $\boldsymbol{x}_i$ and other instances in $\mathcal{D}$, assume the $K$ nearest neighbors (those $K$ nearest instances to $\boldsymbol{x}_i$ based on a particular distance measure $\mathrm{dist}(\cdot, \cdot)$) are $\mathcal{N}(\boldsymbol{x}_i; \mathcal{D}) = \{(\boldsymbol{x}_1, y_1), \ldots, (\boldsymbol{x}_K, y_K)\}$. Then, the label $y_i$ of $\boldsymbol{x}_i$ is predicted based on those labels in the neighbor set $\mathcal{N}(\boldsymbol{x}_i; \mathcal{D})$. For classification task with $C$ classes, we assume the one-hot form of a certain label $y_k$ in $\mathcal{N}(\boldsymbol{x}_i; \mathcal{D})$ is $\boldsymbol{y}_k \in \{0, 1\}^C$, then we have

$$\hat{y}_i = \arg\max_c \left( \sum_{(\boldsymbol{x}_k, \boldsymbol{y}_k) \in \mathcal{N}(\boldsymbol{x}_i; \mathcal{D})} \boldsymbol{y}_k \right)_c , \tag{14}$$

where the confidence of $C$ classes is calculated by voting from the neighbors, and the label is predicted by the elements with the largest confidence. While for the regression case, $y_k \in \mathbb{R}$, we have

$$\hat{y}_i = \frac{1}{K} \sum_{(\boldsymbol{x}_k, y_k) \in \mathcal{N}(\boldsymbol{x}_i; \mathcal{D})} y_k . \tag{15}$$

The prediction is the average of the labels of the neighbors. Thus, in both classification and regression cases, the nearest neighbor model transforms the prediction task over the whole dataset $\mathcal{D}$ into a prediction task based on the local context $\mathcal{N}(\boldsymbol{x}_i; \mathcal{D})$. The neighborhoods of different instances reveal the property of an instance, and vary significantly.

Instead of considering all instances in the neighborhood equally, a variant of the nearest neighbor model re-weights the influence of the neighbors, *i.e.*, making the nearer neighbors have more weight in the prediction Bishop (2006); Rasmussen & Williams (2006); Vinyals et al. (2016). Given a vector whose elements are the normalized distance between $\boldsymbol{x}_i$ and its neighbors:

$$\hat{\phi}(\boldsymbol{x}_i) = \mathrm{softmax}\left([-\mathrm{dist}(\boldsymbol{x}_i, \boldsymbol{x}_1), \ldots, -\mathrm{dist}(\boldsymbol{x}_i, \boldsymbol{x}_K)]\right) , \tag{16}$$

with $\hat{\phi}(\boldsymbol{x}_i) \in \mathbb{R}_+^K$, where the softmax operator $\mathrm{softmax}(\cdot)$ makes the sum of weights equals one. Thus, the prediction for classification in Equation 14 could be updated by

$$\hat{y}_i = \arg\max_c \left( \hat{\phi}(\boldsymbol{x}_i)^\top Y_K \right)_c , \tag{17}$$

where we define $Y_K \in \{0, 1\}^{K \times C}$ as the set of one-hot labels in $\mathcal{N}(\boldsymbol{x}_i; \mathcal{D})$, and each row of $Y_K \in \{0, 1\}$ is the one-hot label of an instance. In addition, define the label vector $[y_1, \ldots, y_K] \in \mathbb{R}^K$, the weighted version of regression prediction in Equation 15 becomes

$$\hat{y}_i = \hat{\phi}(\boldsymbol{x}_i)^\top [y_1, \ldots, y_K] . \tag{18}$$

Based on the prediction of $K$NN in Equation 17 and Equation 18, the label of an instance is determined by both the transformed distance vector $\hat{\phi}(\boldsymbol{x}_i)$ to the neighbors and their corresponding labels. Therefore, we set our meta-presentation as the concatenation of both the distance values and labels, which captures instance and label distribution in a local context. We then learn a transformation $\mathbf{T}_\Theta$ in Equation 12 to map the meta-representation to the label and optimize the joint model over multiple heterogeneous tabular datasets.

The distance metric in $K$NN determines the set of neighbors and influences the discriminative ability Xing et al. (2002); Weinberger & Saul (2009). By learning a metric that pulls similar instances together and pushes dissimilar ones away from each other, the classification and regression ability of nearest neighbor models is improved Kulis (2013); Bellet et al. (2015). The learned distance metric is directly related to the number of attributes (dimension), and several special strategies, such as semantic mapping, are required to apply a learned metric from one dataset to another Zhang & Yeung (2010); Kulis et al. (2011); Yang et al. (2011). Instead of learning a separate distance metric for each tabular task, we use mutual information to select related features given a tabular task automatically in Equation 7, which acts as an adaptive metric in TABPTM. The learned weights of TABPTM are shared among different heterogeneous tasks.

## APPENDIX C  DETAILS AND DISCUSSIONS OF TABPTM

In this section, we describe the detailed architecture of TABPTM as well as the whole training flow. Finally, we discuss the relationship between TABPTM and some related methods.

### C.1  DETAILS OF THE SCORE TRANSFORMATION

In Equation 9, we utilize a transformation $\mathbf{T}_\Theta$ to map the meta-representation to the prediction score of an instance, for all $C$ classes or the regression score. The transformation could be implemented via various kinds of deep neural networks.

We describe the detailed architectures of the deep model following (Gorishniy et al., 2021). Multi-Layer Perceptron (MLP) contains several layers of non-linear blocks

$$\mathbf{MLP}(\boldsymbol{x}) = \text{Linear}(\text{MLPBlock}(\ldots(\text{MLPBlock}(\boldsymbol{x})))) \tag{19}$$

$$\text{MLPBlock}(\boldsymbol{x}) = \text{Dropout}(\text{ReLU}(\text{Linear}(\boldsymbol{x}))) . \tag{20}$$

The Linear block means a fully connected layer with linear projection. In the classification scenario, MLP maps the class-wise meta-representation $\phi_c(\boldsymbol{x}_i)$ to the prediction score $s(\boldsymbol{x}_i)_c$ (a scalar):

$$s(\boldsymbol{x}_i)_c = \mathbf{MLP}(\phi_c(\boldsymbol{x}_i)), \ \forall c = 1, \ldots, C . \tag{21}$$

Although this is a one-to-one mapping from the class-specific meta-representation to the confidence score, we validate its effectiveness in our experiments. In the regression scenario, a single MLP is applied to all the inputs.

The mapping $\mathbf{T}$ could also be implemented with Residual Network (ResNet) (He et al., 2016) and Transformer (Vaswani et al., 2017). For example, ResNet has the following architecture:

$$\mathbf{ResNet}(\boldsymbol{x}) = \text{Prediction}\left(\text{ResNetBlock}\left(\ldots(\text{ResNetBlock}\left(\text{Linear}\left(\boldsymbol{x}\right)\right)\right)\right)\right) \tag{22}$$

$$\text{ResNetBlock}(\boldsymbol{x}) = \boldsymbol{x} + \text{Dropout}(\text{Linear}(\text{Dropout}(\text{ReLU}(\text{Linear}(\text{BatchNorm}(\boldsymbol{x})))))) \tag{23}$$

$$\text{Prediction}(\boldsymbol{x}) = \text{Linear}(\text{ReLU}(\text{BatchNorm}(\boldsymbol{x}))) . \tag{24}$$

Different from MLP, ResNet has a residual link from its input to the output, and Batch Normalization (Ioffe & Szegedy, 2015) is introduced in the building block of ResNet.

We investigate several choices of $\mathbf{T}$ in our experiments. We find the simple MLP (*e.g.*, with three layers) is competitive in most cases. Applying more complicated ResNet and Transformer cannot obtain obvious improvements. Therefore, we set $\mathbf{T}$ with MLP, and better architectures can be explored in further research.

### C.2  THE PRE-TRAINING AND DOWNSTREAM APPLICATION WORKFLOW

TABPTM is a classification/regression model based on the meta-representation. The meta-representation transforms an instance into a $K$-dimensional form no matter how many dimensions it originally had, and the meta-representation extractor does not contain any learnable parameters.

The pre-training objective in Equation 13 is constructed based on the loss function of various pre-training tabular datasets (seen datasets). The indexes of nearest neighbors for each instance $\boldsymbol{x}_i$ are calculated at the beginning of the optimization and recorded for later use. We optimize Equation 13

---

**Algorithm 1** Pre-training on multiple heterogeneous tabular datasets.

---

**Require:** $T$ tabular training set $\mathbb{D} = \{\mathcal{D}_1, \ldots, \mathcal{D}_T\}$, the initialized model $\mathbf{T}_\Theta$
1: **for all** iteration = 1,... **do**
2:     Sample a dataset $\mathcal{D}_t$ from $\mathbb{D}$
3:     Sample a mini-batch with $B$ instances $\{\boldsymbol{x}_i^t, y_i^t\}_{i=1}^B$
4:     **for all** $(\boldsymbol{x}_i^t, y_i^t)$ **do**
5:         Get metric-based meta-representation $\Phi(\boldsymbol{x}_i^t)$ based on nearest neighbors in $\mathcal{D}_t$
6:         Obtain the prediction score $\mathbf{T}_\Theta(\Phi(\boldsymbol{x}_i^t))$
7:         **if** classification **then**
8:             Predict via $\hat{y}_i^t = \arg\max_c \mathbf{T}_\Theta(\Phi(\boldsymbol{x}_i^t)) = \{s(\boldsymbol{x}_i^t)_1, \ldots, s(\boldsymbol{x}_i^t)_{C_t}\}$
9:         **else**
10:            Predict via $\hat{y}_i^t = \mathbf{T}_\Theta(\Phi(\boldsymbol{x}_i^t))$ for regression
11:         **end if**
12:         Compute loss $\ell(\hat{y}_i^t, y_i^t)$
13:     **end for**
14:     Accumulate $B$ losses as Eq. 13
15:     Update $\Theta$ with SGD
16: **end for**
17: **Return:** Pre-trained model $\Theta$

---

in a stochastic manner. In particular, we randomly select a seen tabular dataset and randomly sample a mini-batch from the dataset in each iteration. For each sampled instance, we calculate the metric-based meta-representation based on its neighborhood. For classification, we make predictions with Equation 11, while for regression, we obtain the predicted label directly from the mapped results $\mathbf{T}_\Theta(\Phi(\boldsymbol{x}_i))$. The pre-training phase of TABPTM on multiple heterogeneous tabular datasets is summarized in Algorithm 1.

The learned TABPTM could be deployed with the following two options.

**Direct Generalization**. Since all tabular instances could be transformed into the homogeneous $\Phi(\boldsymbol{x}_i)$ regardless of its original dimension, the learned TABPTM can be applied to a downstream tabular dataset directly. We use classification on a downstream dataset with unseen classes and attributes as an example, and the results are also reported in our experiments. Given a new dataset $\mathcal{D}_u$ and a corresponding test instance $\boldsymbol{x}_*^u$, we calculate the meta-representation of $\boldsymbol{x}_*^u$ w.r.t. each one of the $C_u$ unseen classes. The distances between $\boldsymbol{x}_*^u$ to the nearest neighbors in $\mathcal{D}_u$ and the labels of those neighbors are utilized to obtain the meta-representation. We obtain $C_u$ meta-representations with $K$-dimension, one for each class. Then we apply the learned model $\mathbf{T}_\Theta$ over them to obtain the $C_u$-dimension class confidence vector without additional training. The instance is classified as one of the unseen classes by selecting the index with the maximum confidence. We summarize this workflow in Algorithm 2. The direct deployment requires the search of nearest neighbors in the downstream dataset, which is fast and could be accelerated by some off-the-shelf methods.

**Fine-Tuned Generealization**. Another choice is to use the pre-trained TABPTM as initialization and fine-tune the weights over the downstream task $\mathcal{D}_u$. In other words, the following objective is minimized with gradient descent for several steps

$$\min_\Theta \sum_{(\boldsymbol{x}_i^u, y_i^u) \sim \mathcal{D}_u} \ell(\mathbf{T}_\Theta(\boldsymbol{x}_i^u), y_i^u) . \tag{25}$$

The optimization starts from the learned weights $\Theta$ from the pre-training stage. Additional learnable parameters are usually added when fine-tuning a pre-trained tabular model Zhu et al. (2023), and the size of parameters is related to the dimension and class number of a downstream tabular data. In TABPTM, it is notable that no additional learnable parameters are required to be added in this deployment stage, which makes the fine-tuning efficient and avoids using special hyper-parameters for different sets of parameters.

---

**Algorithm 2** Apply the pre-trained model to the downstream tabular dataset.

---

**Require:** Tabular training set $\mathcal{D}_u$, test instance $\boldsymbol{x}_*^u$, the learned model $\mathbf{T}_\Theta$
 1: Compute the metric-based meta-representation $\Phi(\boldsymbol{x}_*^u)$ for $\boldsymbol{x}_*^u$
 2: **if** classification **then**
 3:    The meta-representation $\Phi(\boldsymbol{x}_*^u) = \{(\phi_c(\boldsymbol{x}_*^u))\}_{c=1}^{C_u}$
 4:    Obtain the prediction score $\{\mathbf{T}_\Theta(\phi_c(\boldsymbol{x}_*^u))\}_{c=1}^{C_u}$
 5:    Predict via $\hat{y}_*^u = \arg\max_c \{s(\boldsymbol{x}_*^u)_1, \ldots, s(\boldsymbol{x}_*^u)_{C_u}\}$
 6: **else**
 7:    Predict via $\hat{y}_*^u = \mathbf{T}_\Theta(\phi(\boldsymbol{x}_*^u))$
 8: **end if**
 9: **Return:** The predicted label of $\boldsymbol{x}_*^u$

---

Table 4: The detailed statistics of all tabular datasets. "Abbr." means the abbreviation of the name of the tabular dataset. The $C$ and $N$ denote the class number and the instance number of the datasets. There are two types of attributes with numerical and categorical values, and we denote their numbers as "Num." and "Cat.", respectively.

| Name | Abbr. | Task type | C | N | Num. | Cat. | Name | Abbr. | Task type | C | N | Num. | Cat. |
|---|---|---|---|---|---|---|---|---|---|---|---|---|---|
| Another-Dataset-on-used-Fiat-500-(1538-rows) | ADO | regression | 1 | 1538 | 6 | 0 | Bank_Customer_Churn_Dataset | BCC | binclass | 2 | 10000 | 6 | 4 |
| archive2 | ARC | regression | 1 | 1143 | 11 | 1 | CDC_Diabetes_Health_Indicators | CDH | binclass | 2 | 253680 | 7 | 14 |
| archive_r56_Maths | ARM | regression | 1 | 397 | 1 | 29 | E-CommereShippingData | ECS | binclass | 2 | 10999 | 6 | 4 |
| archive_r56_Portuguese | ARP | regression | 1 | 651 | 1 | 29 | Fitness_Club_c | FCC | binclass | 2 | 11000 | 10 | 5 |
| airfoil_self_noise | ASN | regression | 1 | 1503 | 5 | 0 | banknote_authentication | BAA | binclass | 2 | 1382 | 4 | 0 |
| analcatdata_supreme | ASU | regression | 1 | 4052 | 7 | 0 | kc2 | KC2 | binclass | 2 | 522 | 21 | 0 |
| auction_verification | AVE | regression | 1 | 2043 | 6 | 1 | maternal_health_risk | MHR | multiclass | 3 | 1014 | 6 | 0 |
| Bias_correction_r | BCR | regression | 1 | 7725 | 21 | 0 | seismic+bumps | SEB | binclass | 2 | 2584 | 14 | 5 |
| communities_and_crime | CAC | regression | 1 | 1994 | 102 | 0 | sports_articles_for_objectivity_analysis | SAF | binclass | 2 | 800 | 57 | 2 |
| combined_cycle_power_plant | CCP | regression | 1 | 9568 | 4 | 0 | turiye_student_evaluation | TSE | multiclass | 5 | 5820 | 30 | 2 |
| concrete_compressive_strength | CCS | regression | 1 | 1030 | 8 | 0 | water_quality | WAQ | binclass | 2 | 7996 | 20 | 0 |
| 1000-Cameras-Dataset | CDA | regression | 1 | 1038 | 10 | 0 | blood | BLO | binclass | 2 | 748 | 4 | 0 |
| Contaminant-detection | CDI | regression | 1 | 2400 | 30 | 0 | breast-cancer | BRC | binclass | 2 | 286 | 9 | 0 |
| dataset_sales | DAT | regression | 1 | 10738 | 10 | 0 | breast-cancer-wisc | BCW | binclass | 2 | 699 | 9 | 0 |
| debutanizer | DBT | regression | 1 | 2394 | 7 | 0 | breast-cancer-wisc-prog | BCP | binclass | 2 | 198 | 33 | 0 |
| 3D_Estimation_using_RSSI_of_WLAN_dataset | EUR | regression | 1 | 5760 | 6 | 0 | dermatology | DER | multiclass | 6 | 366 | 34 | 0 |
| Goodreads-Computer-Books | GCB | regression | 1 | 1234 | 5 | 0 | echocardiogram | ECH | binclass | 2 | 131 | 10 | 0 |
| housing_price_prediction | HPP | regression | 1 | 545 | 5 | 7 | heart-cleveland | HEC | multiclass | 5 | 303 | 13 | 0 |
| Is-this-a-good-customer | ITA | regression | 1 | 1723 | 9 | 4 | Basketball_c | BAS | binclass | 2 | 1340 | 11 | 0 |
| Parkinson_Multiple_Sound_Recording | PMS | regression | 1 | 1040 | 26 | 0 | diabetes | DIA | binclass | 2 | 202944 | 3 | 18 |
| puma8NH | PNH | regression | 1 | 8192 | 8 | 0 | mice_protein_expression | MIC | multi-class | 8 | 1080 | 77 | 0 |
| puma32H | PUH | regression | 1 | 8192 | 32 | 0 | Wilt | WIL | binclass | 2 | 4821 | 5 | 0 |
| Student_Alcohol_Consumption | SAC | regression | 1 | 395 | 13 | 17 | bank_marketing | BAN | binclass | 2 | 45211 | 7 | 7 |
| Shop_Customer_Data | SCA | regression | 1 | 2000 | 4 | 2 | statlog_german_credit_data | STA | binclass | 2 | 1000 | 7 | 13 |
| stock_fardamento02 | SFA | regression | 1 | 6277 | 5 | 1 | company_bankruptcy_prediction | COM | binclass | 2 | 6819 | 93 | 2 |
| sulfur | SFU | regression | 1 | 10081 | 6 | 0 | drug_consumption | DRU | multiclass | 7 | 1884 | 12 | 0 |
| Shipping | SHP | regression | 1 | 10999 | 5 | 4 | dry_bean_dataset | DRY | multiclass | 7 | 13611 | 16 | 0 |
| shrutime | SHR | regression | 1 | 10000 | 4 | 6 | internet_firewall | INT | multiclass | 4 | 65532 | 7 | 0 |
| satellite_image | SIM | regression | 1 | 6435 | 36 | 0 | heart-hungarian | HEA | binclass | 2 | 294 | 12 | 0 |
| Student_Performance_Portuguese | SPP | regression | 1 | 397 | 15 | 17 | heart-va | HEV | multiclass | 5 | 200 | 12 | 0 |
| stock | STO | regression | 1 | 950 | 9 | 0 | breast-cancer-wisc-diag | BRE | binclass | 2 | 569 | 30 | 0 |
| svmguide3 | SVM | regression | 1 | 1243 | 22 | 0 | mammographic | MAM | binclass | 2 | 961 | 5 | 0 |
| VulNoneVul | VNV | regression | 1 | 5692 | 16 | 0 | parkinsons | PAR | binclass | 2 | 195 | 22 | 0 |
| wine+quality | WQA | regression | 1 | 6497 | 11 | 0 | post-operative | POS | multiclass | 3 | 90 | 8 | 0 |
| Wine_Quality_white | WQW | regression | 1 | 4898 | 11 | 0 | primary-tumor | PRI | multiclass | 15 | 330 | 17 | 0 |
| Waterstress | WTS | regression | 1 | 1188 | 22 | 0 | spect | SPE | binclass | 2 | 265 | 22 | 0 |

# APPENDIX D  DETAILS OF EXPERIMENTAL SETUPS

## D.1  DATASETS

We experiment with 72 tabular datasets, which contain 36 datasets for classification and 36 datasets for regression. The statistics of all the datasets are listed in Table 4. We use $C$ and $N$ to denote the class number and the instance number of the datasets. For a regression task, we set $C = 1$. There are two types of attributes with numerical and categorical values, and we denote their numbers as "Num." and "Cat.", respectively. The datasets are collected from UCI Machine Learning Repository [2] or OpenML [3].

We use the same protocol for classification and regression. Given 36 datasets, we randomly split them into two sets, each with 18 datasets. The experiments in the main paper utilize the first 18 datasets to pre-train TABPTM, and set the remaining ones as downstream datasets. In other words, we pre-train the model using 18 heterogeneous tabular datasets, and evaluate its generalization ability on 18 datasets with different sizes. We also evaluate other configurations of the pre-training and

---

[2]https://archive.ics.uci.edu/
[3]https://www.openml.org/

downstream split. For example, we use the 18 datasets in the second part to pre-train the model and the first 18 ones as the downstream datasets. The additional results are reported in Table 7 and Table 8.

## D.2    ADDITIONAL IMPLEMENTATION DETAILS

We compare our TABPTM with different types of methods and describe the detailed way we tune their hyper-parameters in this subsection. We mainly follow the setups in (Gorishniy et al., 2021) to determine the hyper-parameters.

**Classical methods and deep tabular methods**. Both classical tabular methods (SVM, XGBoost) and standard deep methods (MLP) are trained for each dataset separately. We use the official hyper-parameter search spaces for deep tabular methods (FT-T, TabCaps, and TabR). We tune their hyper-parameters and carry out early stopping on the corresponding validation set of a given dataset. All hyper-parameters are selected by Optuna library[4] with Bayesian optimization over 30 trials Gorishniy et al. (2021). The best hyper-parameters are used and the average accuracy/RMSE over 10 different random seeds is calculated.

**Pre-training and fine-tuning approaches**. For TabPFN, we utilize the best official checkpoint, then we apply TabPFN on a downstream dataset with its in-context learning ability. For XTab, We reuse the checkpoint with the highest number of training epochs from the official implementation, then we perform evaluations on the target datasets using XTab's light fine-tuning approach. For DEN, we divide all pre-training datasets into binary and multiclass groups. Each group is then used to train models on their corresponding downstream unseen datasets. We set the learning rate as 0.001 and fine-tune the transform block on the downstream tasks.

**TABPTM**. We implement our TABPTM with a three-layer MLP. The combination of three distances, namely Euclidean distance, Manhattan distance, and Bray-Curtis distance, are utilized. The influence of distances are investigated in Appendix E. During the pre-training, we randomly sample 1024 examples from a seen dataset in each iteration. When we fine-tune TABPTM in Table 1 and Table 2, we set the learning rate as 0.01 and fine-tune the whole model for 30 epochs. The model in the last epoch is saved for evaluation.

## APPENDIX E    ADDITIONAL EXPERIMENTS AND ANALYSES

We analyze the properties of TABPTM from the following aspects.

### E.1    DIRECT GENERALIZATION OF TABPTM

The learned TABPTM has the ability to make predictions directly given a downstream dataset. The results of this variant, denoted by TABPTM$_D$ are included in Table 9 and Table 10. We find that the direct generalization of the pre-trained TABPTM is competitive in some cases (*e.g.*, on "ADO", "DER", and "ECH"), but fine-tuning TABPTM is still necessary on most datasets. One possible reason is that different classes in classification tasks are differentiated by their discrete labels, which makes cross-dataset generalization more difficult when compared with the continuous labels in regression. In addition, we only use sparse labels $\{1, -1\}$ in meta-representation for classification, which increases the difficulty of discovering the shared pattern from the local context.

### E.2    ANALYSIS OF META REPRESENTATION

**Richer supervision helps in classification**. As we mentioned in Equation 5, an auxiliary label $\hat{y}_j$ is appended after the distance value to provide the label information in the meta-representation. We set $\hat{y}_j$ following a one-vs.-rest manner in our implementation, which differentiates those instances in the neighborhood belonging to a certain class. We investigate whether richer supervision in $\hat{y}_j$ helps the classification task.

Given a binary downstream dataset $\mathcal{D}_u$, we first train XGBoost with default hyper-parameters (denoted as "teacher"), whose predictions (scalars) on the training set instances are used as a kind of richer supervision to set $\hat{y}_j$. We train TABPTM from scratch on the downstream dataset, denoted by

---

[4]https://optuna.org/

Table 5: Average accuracy on four classification datasets in Table 2. We use XGBoost with default hyper-parameters as the "teacher", whose predictions on an instance are appended to the meta-representation as enriched supervision. TABPTM variants trained on a given dataset from scratch with vanilla and enriched supervision are denoted as $\text{TABPTM}_\text{S}$ and $\text{TABPTM}_\text{rich}$.

| Dataset | TABPTM | XGBoost | Teacher | $\text{TABPTM}_\text{S}$ | $\text{TABPTM}_\text{rich}$ |
|---------|--------|---------|---------|-------------|----------------|
| ECS | 67.40 | 67.80 | 66.90 | 64.00 | 67.20 |
| BAA | 59.40 | 55.40 | 53.80 | 45.50 | 54.20 |
| SEB | 92.80 | 92.70 | 92.60 | 92.80 | 92.70 |
| SAF | 81.90 | 83.20 | 81.00 | 78.00 | 80.20 |

$\text{TABPTM}_\text{S}$. We also train TABPTM with the supervision-enriched meta-representation from scratch on the downstream dataset, denoted by $\text{TABPTM}_\text{rich}$. Finally, we include the results of the fine-tuned TABPTM pre-trained from various heterogeneous datasets. The results are reported in Table 5.

XGBoost with default hyper-parameters performs better than the vanilla trained $\text{TABPTM}_\text{S}$ in most cases, so we set it as the teacher. With supervision-enriched meta-representation, we find $\text{TABPTM}_\text{rich}$ outperforms $\text{TABPTM}_\text{S}$, which validates our assumptions that we can improve the discerning ability of the model by introducing richer supervision into the meta-representation. However, $\text{TABPTM}_\text{S}$ cannot achieve as good results as TABPTM, so by training a joint model on heterogeneous tabular datasets, it indeed learns shareable experience to achieve more discriminative models.

**Possible explanations of the better regression results**. The a bit improvement of regression results to the classification ones may result from the form of meta-representation and the sharing of prediction experience in TABPTM.

Regression tasks may benefit more from the local context. The core idea of meta-representation is to make predictions in a local context (based on the neighborhood of an instance). Then, the model's ability depends on the "locality" of the true hypothesis, *i.e.*, whether the decision function of an instance depends on its neighborhood or not. Since we may provide estimates of the regression function or conditional expectation by specifying the nature of the local neighborhood Ormoneit & Hastie (1999); Hastie et al. (2009); Kpotufe & Garg (2013), it is more probable to predict the continuous label of a center instance based on the weighted labels of its neighbors. Therefore, making predictions via meta-representation may help regression tasks.

More shareable knowledge in regression. As depicted in Figure 3, the strategy for inferring an instance's label from the distribution of distances to its neighbors is applicable across heterogeneous tabular tasks. For classification tasks, if an instance resides within a high-density region of a class (akin to being near the class center), the majority of values in the meta-representation would typically be small, indicating close proximity to neighboring instances of that class. Conversely, if only a few values in the meta-representation are small, while most are large, it indicates that the instance is likely located at the boundary among classes. Such a kind of discerning strategy is more shareable in regression tasks, which works in a way that determines the label via the weighted combination of its neighbors. Therefore, using a pre-trained model (the top-layer MLP in TABPTM) can be generalized to downstream tasks.

Rich label information in regression. In addition to the distance values between a given instance and its nearest neighbors, we append the labels of the neighbors into the meta-representation. The labels in regression tasks are continuous, and even if two instances are close to each other, they may have different label values, which introduces rich information. In contrast, since the labels in classification tasks are discrete, two close instances may have the same label and only differ in their distance values in meta-representation. In summary, the labels in regression tasks may introduce richer supervision. We validate this assumption in the previous experiments. We use the prediction of a well-trained XGBoost as the label of an instance in meta-representation, which transforms the labels in classification tasks into continuous ones. The experiments validate that the TABPTM equipped with such continuous labels outperforms the original one, which explains the different results in regression and classification to some extent.

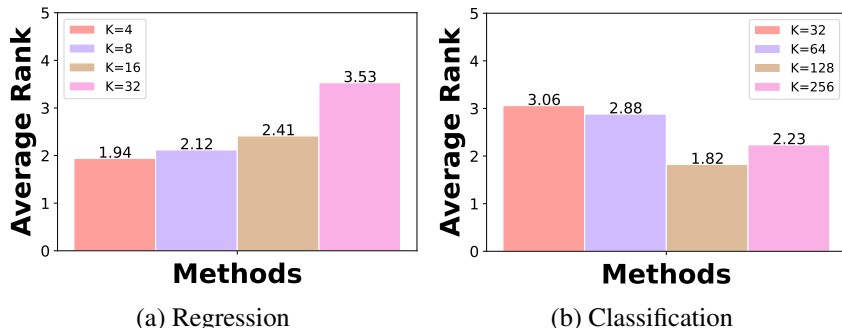

(a) Regression         (b) Classification

Figure 5: Average rank (the lower, the better) on 18 downstream datasets as in Table 1 and Table 2. Different dimension values $K$ of meta-representation are used to pre-train TABPTM.

**The influence of the metric on meta-representation**. We compare the results when we pre-train TABPTM over the vanilla meta-representation and its metric-based variant for regression tasks. Since different regression tasks have diverse label ranges, we compare the change of average rank with and without the metric. TABPTM gets an average rank of 1.941 over the 18 datasets in Table 1, compared with 12 methods in total. If the metric is not used, the average rank increases to 2.765 (the lower, the better). The results clearly indicate that the distance metric filters out redundant and noisy attributes, which is necessary to improve the generalization ability of TABPTM.

### E.3 THE INFLUENCE OF THE DIMENSION OF THE META-REPRESENTATION

In Equation 5, we consider the nearest $K$ neighbors in the training set of each class in classification and $K$ neighbors in the whole training set for regression. Then the size of the meta-representation is related to $K$. We set $K = 8$ for regression and $K = 128$ for classification by default in previous experiments. We pre-train TABPTM over meta-representations with different dimensions $K$ in Figure 5, where we show the change of average rank on 18 downstream datasets when comparing 4 values of $K$. We find different types of downstream datasets may prefer various dimension values.

### E.4 THE INFLUENCE OF THE DISTANCES IN META-REPRESENTATION

In Equation 5, a certain distance is utilized to obtain the nearest neighbors in the training set to construct the meta-representation. The indexes of instances, as well as the distance values, depend on the choice of distances. We consider several distance measures (equipped with the adaptive metric in Equation 7. The average rank on 18 downstream datasets as in Table 1 and Table 2 based on different distances as well as their combinations are investigated. "Man" denotes Manhattan distance, "Euc" denotes the Euclidean distance, "Bra" denotes Bray-Curtis distance, "CAN" denotes Canberra distance, "COS" denotes cosine distance, and "CHE" denotes Chebyshev distance. "MEB" denotes the combination of Manhattan, Euclidean, and Bray-Curtis distances. The results in Figure 6 show the change of average rank on 18 downstream datasets when comparing seven distance choices. The Manhattan distance works the best for regression, and the combination of three distances is the best choice for classification. We set the latter one as the default distance choice in TABPTM.

### E.5 THE INFLUENCE OF PRE-TRAINING SIZE

To investigate the impact of varying pre-training data size on the performance of TABPTM, we split the 18 pre-training datasets into two groups: one containing 6 datasets (HEA, HEV, BRE, MAM, PAR, POS) and another containing 12 datasets, which included the initial 6 plus PRI, SPE, DIA, MIC, WIL, and BAN. All other experimental settings remained consistent with those in Table 2. We also compared the performance of models trained directly on individual downstream datasets using the same meta-representation. Hyper-parameters were optimized using the Optuna library, with 30 trials and a maximum of 200 training epochs. This approach is referred to as "TABPTM$_S$".

We randomly selected six datasets as downstream tasks, and their accuracy, along with the average performance rank across the four configurations over 18 downstream datasets, is presented in Table 6.

Table 6: The change of classification accuracy when we increase the size of the pre-training datasets. In addition to three sizes of the pre-training datasets, we also list the results training TABPTM directly on a downstream dataset, corresponding to TABPTM$_S$. We list the accuracy of six randomly selected datasets and the average rank of four variants over the 18 downstream datasets. By enlarging the pre-training size, we find TABPTM has enhanced performance for TABPTM.

| Dataset | 6 datasets | 12 datasets | 18 datasets | TABPTM$_S$ |
|---------|------------|-------------|-------------|------------|
| BAA | 51.10 | 52.50 | **59.40** | 55.40 |
| BCP | 71.80 | 73.80 | **82.00** | 77.00 |
| BLO | 71.80 | 75.10 | **78.00** | 76.90 |
| ECH | 72.20 | 73.30 | **81.20** | 72.60 |
| ECS | 65.70 | 66.10 | **67.40** | 67.30 |
| KC2 | 79.00 | 78.70 | **83.10** | 78.70 |
| avg. rank | 3.667 | 2.833 | 1.000 | 2.333 |

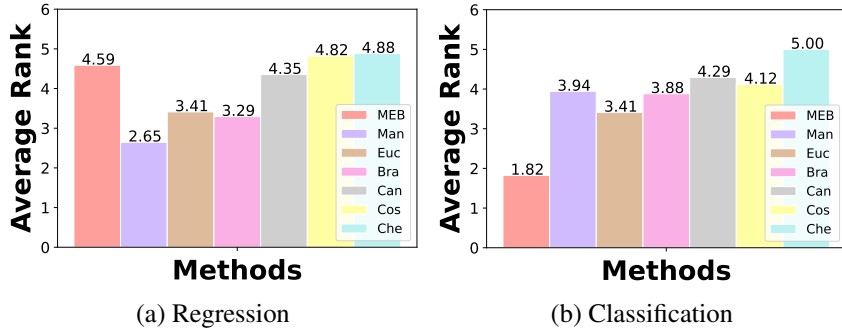

(a) Regression                                  (b) Classification

Figure 6: Average rank (the lower, the better) on 18 downstream datasets as in Table 1 and Table 2. Different distances or their combinations are used to obtain the meta-representation. "Man" denotes Manhattan distance, "Euc" denotes the Euclidean distance, "Bra" denotes Bray-Curtis distance, "CAN" denotes Canberra distance, "COS" denotes cosine distance, and "CHE" denotes Chebyshev distance. "MEB" denotes the combination of Manhattan, Euclidean, and Bray-Curtis distances.

The results indicate that pre-training on a larger and more diverse set of datasets enhances TABPTM's ability to learn transferable knowledge for tabular predictions. Even with only 30 epochs of fine-tuning, TABPTM significantly outperformed models that were fully trained for 200 epochs on downstream datasets.

### E.6 RESULTS WITH OTHER PRE-TRAINING DATASETS

Recall that there are 36 tabular datasets for both classification and regression. We randomly select 18 of them as the pre-training datasets and 18 of them as the downstream ones. We exchange the pre-training and downstream datasets in this subsection. The results are shown in Table 7 and Table 8. The results validate the regression and classification ability of TABPTM variants.

## APPENDIX F   WHOLE EXPERIMENTAL RESULTS

The full results including average RMSE/accuracy and standard deviation of Table 1 and Table 2 are listed in Table 9 and Table 10, respectively. The standard deviation of TABPTM also comes from the estimation of mutual information when constructing the metric-based meta-representation.

Table 7: Average RMSE on 18 downstream datasets (the lower, the better). We use *another set of 18 datasets as the pre-training datasets* w.r.t. Table 1. The best results are shown in bold, and the second-best results are underlined. The value beside the dataset name denotes the scale of the results.

| | TABPTM | SVM | $K$NN | XGBoost | MLP | FT-T | TabR | SAINT | TANGOS | TabNet | PTaRL | XTab |
|---|---|---|---|---|---|---|---|---|---|---|---|---|
| ARM$_{\times 10}$ | .4414 | **.4000** | .4480 | .4290 | .4300 | .4390 | .4650 | .4360 | .4440 | .4910 | .4420 | .4480 |
| ARP$_{\times 10}$ | .3038 | .2930 | .3130 | .2980 | .2930 | **.2860** | .3120 | .2870 | .2960 | .3450 | .3040 | .3530 |
| CCS$_{\times 10}$ | .8217 | .6990 | .9110 | **.5470** | 1.610 | 1.630 | 1.590 | 1.630 | 1.620 | .7400 | .7070 | 1.720 |
| ARC$_{\times 10^3}$ | .3733 | .3790 | .3710 | .3670 | **.3530** | .4010 | .3880 | .3950 | .3680 | .4220 | .3700 | .4790 |
| WTS | .4649 | .4970 | .4680 | .4750 | **.4430** | .4470 | .4710 | .4690 | **.4430** | .4750 | .4500 | .5030 |
| SVM | .7189 | .7610 | .7760 | .7220 | **.6990** | .7200 | .7240 | .8200 | **.6990** | .7970 | .7030 | .8610 |
| ASN$_{\times 10}$ | .2321 | .3200 | .2790 | .1830 | .2670 | .1830 | .1510 | **.1460** | .2040 | .2600 | .3230 | .6420 |
| ITA | .3152 | .3320 | .3420 | **.3110** | .3130 | .3200 | .3160 | .3160 | .3180 | .3230 | .3200 | .3210 |
| SCA$_{\times 10^2}$ | .2854 | .2960 | .3140 | .2870 | .2860 | .2860 | .2870 | .2870 | .2860 | .2910 | **.2850** | .2890 |
| CDI | .2842 | .3070 | .3290 | .2910 | .2480 | .2690 | .2680 | .2800 | **.2230** | .3660 | .2810 | .4430 |
| WQW | .6879 | .6940 | **.6430** | .6670 | .6880 | .7060 | .6860 | .7080 | .6850 | .7450 | .6980 | .7930 |
| EUR$_{\times 10^{-2}}$ | .8943 | 43.70 | 6.760 | **.1380** | 26.40 | 8.040 | .7840 | 4.290 | 11.40 | 17.80 | 33.90 | 53.30 |
| SIM | .6974 | .8010 | .7190 | .7570 | .7440 | .7700 | .7700 | **.6660** | .6830 | .8280 | .7640 | 1.140 |
| WQA | .7172 | .7290 | .7020 | **.7010** | .7200 | .7240 | .7080 | .7420 | .7230 | .7510 | .7280 | .7930 |
| BCR | .8353 | 1.070 | 1.170 | .9280 | .8560 | .8200 | **.5610** | .7580 | .7790 | 1.180 | 1.080 | 1.730 |
| CCP$_{\times 10}$ | .3606 | .4110 | .3850 | **.3450** | 1.680 | 1.690 | 1.600 | 1.700 | 1.700 | .4140 | .4250 | 1.690 |
| SHR | .3243 | .3420 | .3480 | .3210 | .3220 | **.3170** | **.3170** | .3260 | .3240 | .3260 | .3190 | .3400 |
| DAT$_{\times 10}$ | .3993 | .4410 | .4590 | .4020 | **.3960** | .4200 | .4200 | .4180 | **.3960** | .4020 | .3990 | .4300 |
| avg. rank | 4.667 | 8.000 | 7.611 | 3.944 | 4.278 | 5.722 | 5.500 | 6.222 | 4.500 | 9.167 | 6.000 | 11.111 |

Table 8: Average accuracy on 18 downstream datasets (the higher, the better). We use *another set of 18 datasets as the pre-training datasets* w.r.t. Table 2. The best results are shown in bold, and the second-best results are underlined. "OOM" means out-of-memory.

| dataset | TABPTM | SVM | XGBoost | MLP | FT-T | TabR | SAINT | TANGOS | TabNet | TabPFN | XTab | DEN |
|---|---|---|---|---|---|---|---|---|---|---|---|---|
| BAN | 86.20 | 88.00 | 88.00 | 88.00 | 88.00 | 88.00 | 88.00 | **96.20** | 88.00 | 88.00 | 88.00 | 84.80 |
| BAS | 67.30 | **70.40** | 68.00 | 65.80 | 69.60 | 68.40 | 70.20 | 69.80 | 65.70 | 62.20 | 66.40 | 66.50 |
| BRE | 97.50 | 96.50 | 97.20 | 96.30 | 95.60 | **98.20** | 96.10 | 96.20 | 96.80 | 94.10 | 93.00 | 95.70 |
| COM | 96.50 | 96.50 | **96.60** | 96.20 | 96.30 | 96.40 | 96.20 | OOM | 96.30 | 96.30 | 96.40 | 96.10 |
| DIA | 76.20 | **77.60** | 75.50 | 74.50 | 77.20 | 75.90 | 76.00 | 75.90 | 73.90 | 69.60 | 70.10 | 67.90 |
| DRU | 40.20 | **41.40** | 40.30 | 39.90 | 40.80 | **41.40** | 39.50 | 40.60 | 40.20 | 40.30 | 40.40 | 40.10 |
| DRY | 92.90 | 92.90 | **93.10** | **93.10** | 93.00 | 92.70 | **93.10** | **93.10** | **93.10** | 92.50 | 92.30 | 91.80 |
| HEA | 81.70 | **84.70** | 77.60 | 82.10 | 80.60 | 80.90 | 83.30 | 77.40 | 81.90 | 68.00 | 79.30 | 78.30 |
| HEV | 34.50 | 30.00 | **38.70** | 30.00 | 27.20 | 30.70 | 24.50 | 28.50 | 29.50 | 24.30 | 33.00 | 32.30 |
| INT | 91.50 | 74.90 | **93.50** | 79.80 | 79.80 | 80.10 | 91.00 | 80.30 | 79.80 | 61.50 | 56.40 | 63.80 |
| MAM | 83.40 | 82.90 | **84.60** | 81.30 | 82.70 | **84.60** | 83.20 | 84.10 | 82.00 | 81.40 | 80.30 | 78.10 |
| MIC | 98.90 | 99.10 | 96.90 | 99.10 | 98.10 | 98.90 | **99.50** | 99.10 | 98.70 | 85.90 | 43.10 | 83.00 |
| PAR | 94.50 | 92.30 | 94.40 | 90.30 | 90.80 | **94.90** | 93.20 | 89.90 | 90.40 | 86.80 | 87.70 | 87.50 |
| POS | **84.80** | 55.60 | 83.30 | 78.90 | 59.30 | 83.30 | 70.40 | 71.50 | 75.20 | 43.30 | 74.40 | 73.40 |
| PRI | 50.90 | 48.50 | **54.60** | 47.10 | 44.10 | 51.20 | 54.40 | 48.00 | 47.30 | 34.60 | 30.00 | 22.20 |
| SPE | 72.60 | **73.60** | 71.40 | 68.70 | 70.80 | 71.30 | 71.60 | 69.40 | 68.10 | 59.60 | 70.70 | 68.90 |
| STA | 70.40 | **70.50** | 70.40 | 68.00 | 70.40 | **70.50** | 69.80 | 69.60 | 69.60 | 69.60 | 65.30 | 66.90 |
| WIL | 88.40 | 84.50 | 77.40 | 87.60 | **89.30** | 89.00 | OOM | 87.90 | 88.40 | 82.00 | 62.90 | 71.20 |
| avg. rank | 4.056 | 4.056 | 3.889 | 6.667 | 5.889 | 3.500 | 4.941 | 5.471 | 6.333 | 9.611 | 8.667 | 10.000 |

Table 9: The whole results of Table 1: RMSE with standard deviation on 18 downstream datasets (the lower, the better). The best results are shown in bold, and the second-best results are underlined. The value beside the dataset name denotes the scale of the results. TABPTM$_D$ predicts directly without additional training, while TABPTM is fine-tuned on the downstream dataset. "OOM" means out-of-memory.

| | TABPTM$_D$ | TABPTM | SVM | XGBoost | MLP | FT-T | TabR | SAINT | TANGOS | TabNet | TabPFN | XTab | DEN |
|---|---|---|---|---|---|---|---|---|---|---|---|---|---|
| SPP$_{\times 10}$ | .2525 ± .00 | .1648 ± .001 | .2079 ± .00 | .3239 ± .00 | .1551 ± .007 | .1963 ± .013 | **.1430** ± .009 | .1786 ± .005 | .1972 ± .012 | .2038 ± .005 | .4671 ± .068 | .3615 ± .135 | .4736 ± .018 |
| HPP$_{\times 10^7}$ | .1210 ± .001 | .1074 ± .002 | .1075 ± .00 | .1132 ± .00 | .1064 ± .004 | .1107 ± .002 | **.1054** ± .001 | .1103 ± .002 | .1064 ± .001 | .1134 ± .002 | .1400 ± .016 | .1104 ± .00 | .1585 ± .012 |
| STO | 2.974 ± .001 | **.7355** ± .002 | .9513 ± .00 | .7948 ± .00 | .8907 ± .047 | .9408 ± .014 | .9883 ± .020 | .9659 ± .161 | .7809 ± .015 | .8032 ± .014 | 1.057 ± .063 | 1.184 ± .015 | 4.922 ± .352 |
| CDA$_{\times 10^3}$ | .5567 ± .014 | .4956 ± .015 | .6976 ± .00 | .5489 ± .00 | **.4914** ± .053 | .5908 ± .125 | .6613 ± .058 | .5966 ± .076 | .5381 ± .022 | .5893 ± .051 | .6351 ± .019 | .5561 ± .051 | .7441 |
| PMS$_{\times 10^2}$ | .1601 ± .001 | .1534 ± .001 | .1603 ± .00 | .1603 ± .00 | .1629 ± .002 | **.1524** ± .001 | .1633 ± .007 | .1727 ± .006 | .1720 ± .009 | .1543 ± .004 | .1673 ± .005 | .1630 ± .002 | .1717 ± .002 |
| GCB | .5087 ± .002 | .4828 ± .001 | .4884 ± .00 | .5038 ± .00 | **.4437** ± .016 | .4866 ± .001 | .4838 ± .00 | .4844 ± .001 | .4855 ± .001 | .4843 ± .001 | .4876 ± .007 | .4841 ± .00 | .5015 ± .016 |
| ADO$_{\times 10^3}$ | **.7184** ± .002 | .7374 ± .002 | .7956 ± .00 | .7880 ± .00 | .7752 ± .010 | .7725 ± .004 | .7710 ± .007 | .7726 ± .001 | .7731 ± .003 | .7810 ± .008 | .8197 ± .025 | .7668 ± .006 | 1.295 ± .294 |
| CAC | .1341 ± .00 | **.1322** ± .001 | .1348 ± .00 | .1467 ± .00 | .1433 ± .002 | .1337 ± .001 | .1350 ± .001 | .1452 ± .006 | OOM | .1393 ± .002 | .1591 ± .010 | .1384 ± .001 | .1781 ± .019 |
| AVE$_{\times 10^3}$ | 3.816 ± .007 | 3.080 ± .007 | 6.394 ± .00 | 4.678 ± .00 | **.6983** ± .124 | 9.765 ± .027 | 9.942 ± .146 | 9.933 ± .594 | 9.920 ± .070 | 9.811 ± .036 | 3.314 ± .236 | 3.484 ± .134 | 9.982 ± .184 |
| DBT$_{\times 10^{-1}}$ | .9609 ± 34.5 | .6617 ± 34.6 | 1.107 ± .00 | .7065 ± .00 | .7841 ± .029 | .8534 ± .005 | .7356 ± .018 | **.6233** ± .044 | .6385 ± .017 | .6884 ± .016 | 1.034 ± .037 | .9602 ± .010 | 1.446 ± .035 |
| ASU$_{\times 10^{-1}}$ | 2.452 ± .012 | .9036 ± .032 | 1.862 ± .00 | 1.078 ± .00 | **.8418** ± .038 | 1.013 ± .025 | .9379 ± .015 | .9037 ± .029 | 1.010 ± .042 | 1.071 ± .046 | 1.109 ± .085 | 1.293 ± .040 | 3.016 ± .905 |
| VNV$_{\times 10^{-1}}$ | 1.071 ± .016 | 1.015 ± .016 | 1.020 ± .00 | 1.071 ± .00 | 1.032 ± .012 | **.9879** ± .003 | 1.004 ± .010 | 1.031 ± .032 | 1.020 ± .008 | 1.019 ± .016 | 1.007 ± .011 | .9910 ± .002 | 1.007 ± .004 |
| SFA$_{\times 10^2}$ | .2505 ± .013 | .1830 ± .013 | .1888 ± .00 | .2012 ± .00 | **.1672** ± .016 | .2024 ± .005 | .1806 ± .00 | .1843 ± .005 | .2003 ± .015 | .2001 ± .002 | .2312 ± .027 | .2192 ± .002 | .1872 ± .003 |
| PNH$_{\times 10}$ | .3893 ± .00 | .3318 ± .00 | .3375 ± .00 | .3862 ± .00 | .3404 ± .002 | .3324 ± .001 | **.3265** ± .00 | .3296 ± .00 | .3344 ± .003 | .3366 ± .001 | .3326 ± .004 | .3346 ± .001 | .3832 ± .009 |
| PUH$_{\times 10^{-2}}$ | 1.515 ± 266. | .7804 ± 266. | 2.557 ± .00 | 2.776 ± .00 | .9376 ± .070 | 1.299 ± .658 | .7762 ± .003 | **.6072** ± .008 | .8815 ± .020 | .8925 ± .012 | .9189 ± .068 | 1.612 ± .031 | 2.855 ± .100 |
| SFU$_{\times 10^{-1}}$ | .3174 ± 21.5 | .2032 ± 21.5 | .3479 ± .00 | .2206 ± .00 | .2545 ± .021 | .2395 ± .007 | .2555 ± .065 | .2307 ± .009 | **.1891** ± .032 | .2218 ± .033 | .2976 ± .019 | .3084 ± .003 | .3686 ± .006 |
| SHP | .4391 ± .007 | .4318 ± .001 | .4930 ± .00 | .4841 ± .00 | **.4292** ± .001 | .4464 ± .001 | .4393 ± .014 | .4318 ± .005 | OOM | .4514 ± .007 | .4341 ± .003 | .4341 ± .001 | .4402 ± .011 |
| SAC$_{\times 10}$ | .4007 ± .010 | **.3983** ± .009 | .4015 ± .00 | .4169 ± .00 | .4329 ± .014 | .4472 ± .002 | .4494 ± .002 | .4511 ± .005 | .4508 ± .001 | .4508 ± .004 | .4923 ± .028 | .4324 ± .016 | .4379 ± .009 |
| avg. rank | 8.500 | **2.444** | 8.611 | 8.056 | 4.778 | 6.167 | 5.389 | 6.278 | 5.875 | 6.889 | 9.000 | 7.111 | 11.111 |

Table 10: The whole results of Table 2: Average accuracy with standard deviation on 18 downstream datasets. The best results are shown in bold, and the second-best results are underlined. TABPTM$_D$ predicts directly without additional training, while TABPTM is fine-tuned on the downstream dataset.

| dataset | TABPTM$_D$ | TABPTM | SVM | XGBoost | MLP | FT-T | TabR | SAINT | TANGOS | TabNet | TabPFN | XTab | DEN |
|---|---|---|---|---|---|---|---|---|---|---|---|---|---|
| BCC | 85.90 ± 0.30 | 85.30 ± 0.80 | 86.50 ± 0.01 | 86.80 ± 0.28 | 85.70 ± 0.01 | 86.50 ± 0.17 | 85.80 ± 0.26 | **87.10** ± 0.34 | 86.40 ± 0.30 | 86.00 ± 1.50 | 78.90 ± 1.60 | 85.00 ± 0.60 | 80.10 ± 2.30 |
| CDH | 72.60 ± 0.80 | 75.20 ± 0.00 | 75.10 ± 0.01 | 75.10 ± 0.09 | 75.20 ± 0.10 | 75.30 ± 0.12 | 74.20 ± 0.14 | 75.00 ± 0.14 | 75.20 ± 0.13 | 75.00 ± 0.13 | 75.00 ± 0.26 | **86.70** ± 0.61 | 70.20 ± 1.20 |
| ECS | 67.40 ± 1.30 | 67.40 ± 0.80 | **75.10** ± 0.01 | 67.80 ± 0.29 | 67.20 ± 0.23 | 67.80 ± 0.43 | 66.20 ± 0.52 | 67.70 ± 0.34 | 67.80 ± 0.45 | 67.10 ± 0.46 | 67.20 ± 0.70 | 66.30 ± 1.30 | 72.90 ± 0.68 |
| FCC | 76.30 ± 0.75 | 77.30 ± 1.40 | 76.70 ± 0.01 | **79.20** ± 1.20 | 78.80 ± 2.00 | 78.00 ± 1.70 | 77.90 ± 0.74 | 77.60 ± 1.60 | 77.60 ± 2.10 | 78.60 ± 1.50 | 77.00 ± 2.50 | 72.80 ± 1.80 | 71.00 ± 1.00 |
| BAA | 48.40 ± 1.20 | **59.40** ± 1.40 | 53.10 ± 0.01 | 55.40 ± 0.73 | 54.10 ± 1.40 | 53.80 ± 1.20 | 55.60 ± 0.01 | 55.50 ± 1.20 | 54.70 ± 1.50 | 54.60 ± 1.70 | 54.60 ± 1.30 | 49.30 ± 1.30 | 45.90 ± 2.00 |
| KC2 | 84.10 ± 1.50 | 83.10 ± 1.60 | 81.90 ± 0.01 | 79.60 ± 1.40 | **84.50** ± 1.70 | 79.10 ± 2.40 | 82.00 ± 0.65 | 77.80 ± 2.20 | 81.60 ± 1.00 | 83.10 ± 2.30 | 79.70 ± 2.30 | 84.40 ± 2.70 | 82.60 ± 2.60 |
| MHR | 64.40 ± 1.30 | 68.70 ± 1.10 | 67.00 ± 0.01 | **81.90** ± 0.63 | 73.20 ± 1.20 | 69.60 ± 1.40 | 73.80 ± 0.72 | 78.10 ± 2.20 | 74.10 ± 1.80 | 71.40 ± 1.70 | 58.80 ± 5.00 | 54.30 ± 1.60 | 56.80 ± 1.50 |
| SEB | 92.60 ± 1.80 | 92.80 ± 2.80 | 92.80 ± 0.01 | 92.70 ± 0.01 | 92.90 ± 1.00 | 92.90 ± 1.20 | 92.80 ± 0.05 | **93.00** ± 0.21 | 92.90 ± 0.17 | 92.80 ± 0.13 | 92.90 ± 0.13 | 92.80 ± 0.26 | 92.10 ± 1.00 |
| SAF | 81.40 ± 0.98 | 81.90 ± 1.30 | 83.50 ± 0.01 | 83.20 ± 0.98 | 81.80 ± 1.10 | 80.60 ± 1.50 | **84.60** ± 0.73 | 83.60 ± 1.10 | 82.90 ± 0.96 | 80.30 ± 1.60 | 79.70 ± 1.30 | 68.80 ± 2.00 | 78.30 ± 0.03 |
| TSE | 44.50 ± 0.60 | 48.30 ± 0.80 | 49.20 ± 0.01 | **51.90** ± 0.15 | 49.70 ± 0.80 | 51.20 ± 0.79 | 49.20 ± 0.55 | 50.20 ± 0.58 | 50.40 ± 0.64 | 49.30 ± 0.79 | 49.20 ± 1.10 | 46.70 ± 0.51 | 31.50 ± 1.40 |
| WAQ | 88.90 ± 0.70 | **91.10** ± 1.10 | 89.80 ± 0.01 | 89.40 ± 0.20 | 89.40 ± 0.20 | 88.70 ± 0.41 | 88.80 ± 0.18 | 89.60 ± 0.19 | 89.60 ± 0.37 | 89.20 ± 0.27 | 88.40 ± 0.64 | 88.00 ± 2.20 | 88.20 ± 2.10 |
| BLO | 73.60 ± 0.45 | 78.00 ± 1.90 | 74.70 ± 0.01 | 76.90 ± 1.90 | 77.70 ± 1.90 | 77.90 ± 0.41 | 77.80 ± 0.38 | 75.40 ± 2.50 | **78.20** ± 1.00 | 77.20 ± 0.13 | 76.40 ± 2.60 | 75.70 ± 1.20 | 71.20 ± 0.20 |
| BRC | 66.00 ± 0.85 | 67.10 ± 1.00 | 55.20 ± 0.01 | **69.00** ± 0.42 | 62.60 ± 1.80 | 65.90 ± 2.90 | 66.80 ± 1.60 | 67.50 ± 1.40 | 67.00 ± 1.70 | 67.00 ± 1.70 | 66.70 ± 1.60 | 65.60 ± 0.34 | 63.40 ± 1.70 |
| BCW | 96.60 ± 1.30 | **97.50** ± 1.70 | 97.10 ± 0.01 | 97.10 ± 1.30 | 96.60 ± 0.20 | 95.80 ± 1.10 | 97.10 ± 1.30 | 95.90 ± 0.46 | 88.40 ± 2.30 | 96.90 ± 0.53 | 95.10 ± 1.80 | 97.30 ± 2.40 | 97.10 ± 1.70 |
| BCP | 82.00 ± 1.70 | 82.00 ± 2.00 | 75.00 ± 0.01 | 84.70 ± 2.70 | 79.00 ± 1.40 | 83.20 ± 2.00 | **87.30** ± 1.00 | 83.00 ± 1.70 | 76.30 ± 0.38 | 82.30 ± 0.12 | 74.00 ± 1.20 | 77.70 ± 1.80 | 75.00 ± 2.00 |
| DER | **99.80** ± 0.30 | 99.20 ± 0.50 | 98.60 ± 0.01 | 99.00 ± 0.80 | 98.50 ± 0.30 | 99.50 ± 0.39 | 98.60 ± 0.30 | 98.40 ± 0.90 | 98.80 ± 0.10 | 98.90 ± 0.34 | 89.50 ± 0.00 | 80.00 ± 0.50 | 97.70 ± 0.50 |
| ECH | **84.70** ± 1.90 | 81.20 ± 1.80 | 77.80 ± 0.01 | 75.60 ± 1.60 | 79.80 ± 0.33 | 74.10 ± 1.60 | 78.00 ± 2.50 | 80.20 ± 1.60 | 70.40 ± 1.90 | 76.50 ± 1.60 | 69.90 ± 1.60 | 84.40 ± 1.50 | 80.30 ± 1.80 |
| HEC | 52.70 ± 1.00 | 52.30 ± 2.50 | **55.70** ± 0.01 | 50.60 ± 2.30 | 52.20 ± 2.60 | 51.90 ± 1.80 | 53.80 ± 1.90 | 53.90 ± 2.20 | 51.90 ± 1.60 | 52.00 ± 1.20 | 49.80 ± 1.60 | 51.30 ± 1.10 | 47.50 ± 2.90 |
| avg. rank | 7.944 | **4.833** | 6.833 | 5.056 | 6.389 | 6.167 | 6.056 | 5.167 | 5.778 | 6.722 | 9.944 | 8.889 | 10.556 |

