# Supplementary Material

This material provides additional tables and figures that support the rebuttal of our paper. Below is a brief description of each item:

1. **Table1** extends our experiments by incorporating an established Tabular Benchmark.

2. **Figure1** visualizes the raw and transformed data distributions using t-SNE.

3. **Figure2** explores the impact of class imbalance on Meta-representations.

| Datasets | Dummy | KNN | SVM | XGB | CatB | RF | MLP | ResNet | Node | SwitchTab | TabNet | Tangos | DANets | FTT | AutoInt | DCNv2 | SNN | TabTrans | PTaRL | GrowNet | TabR | Excel | ModernNCA | TuneTables | TabPFN | LoCalPFN | TabPTM |
|---|---|---|---|---|---|---|---|---|---|---|---|---|---|---|---|---|---|---|---|---|---|---|---|---|---|---|---|
| FOREX_audchf-day-High | .5068 | .5940 | .7520 | .6627 | .6398 | .6554 | .6901 | .7464 | .6523 | .5232 | .5135 | .5010 | .5370 | .5179 | .6567 | .7480 | .7262 | .7424 | .6191 | .6754 | .7520 | .5297 | .7544 | **.7570** | .7559 | .7540 | .6095 |
| taiwanese_bankruptcy_prediction | .9677 | .9714 | .9677 | .9702 | **.9718** | .9717 | .9685 | .9667 | .9678 | .9522 | .9681 | .9689 | .9686 | .9674 | .9674 | .9662 | .9664 | .9686 | .9670 | .9660 | .9678 | .9669 | .9674 | .9680 | .9692 | .9696 | .9675 |
| rl | .5000 | .6549 | .6308 | .7993 | .7858 | .7295 | .6562 | .6592 | .6475 | .6227 | .6281 | .6596 | .6665 | .7196 | .6822 | .6488 | .6490 | .6282 | .6429 | .6258 | **.8779** | .7168 | .8375 | .6160 | .6738 | .6085 | .7857 |
| pc3 | .8978 | .8914 | .8871 | .8856 | .8880 | .8941 | .8820 | .8867 | .8978 | .8835 | .8884 | .8920 | .8907 | .8978 | .8952 | .8867 | .8929 | .8976 | .8909 | .8963 | .8846 | .8969 | .8892 | **.8980** | .8950 | .8965 | .8978 |
| qsar | .6635 | .8057 | .8464 | .8673 | .8613 | .8708 | .8654 | .8894 | .6768 | .8657 | .7912 | .8970 | .8547 | .8537 | .8641 | .8720 | .8761 | .8720 | .8569 | .8569 | .7953 | .8635 | .9000 | .9087 | **.9115** | .8768 | .8768 |
| eye_movements_bin | .5000 | .5802 | .5558 | .6325 | .6152 | .6057 | .5708 | .5754 | .5611 | .5582 | .5645 | .5760 | .5745 | .5945 | .5831 | .5741 | .5764 | .5673 | .5745 | .5579 | .6595 | .6028 | **.9088** | .5490 | .5784 | .5711 | .6735 |
| BNG(breast-w) | .6560 | .9837 | .9781 | .9876 | **.9878** | .9852 | .9846 | .9856 | .9818 | .9834 | .9838 | .9840 | .9860 | .9848 | .9841 | .9848 | .9594 | .9840 | .9839 | .9868 | .9856 | .9859 | .9801 | .9827 | .9842 | .9842 | .9825 |
| FOREX_cadjpy-hour-High | .5179 | .5461 | .7099 | .6121 | .6259 | .5718 | .7061 | .7096 | .5338 | .5334 | .6243 | .5885 | .6871 | .6880 | .6971 | .7045 | .6990 | .7034 | .7046 | .6586 | .6432 | .6644 | .7118 | .6370 | .6290 | .6196 | .6886 |
| dis | .9841 | .9841 | .9841 | .9876 | .9866 | .9841 | .9852 | .9841 | .9841 | .9832 | .9837 | .9849 | .9835 | .9876 | .9848 | .9839 | .9850 | .9841 | .9841 | .9840 | .9876 | .9858 | .9868 | .9840 | .9841 | .9841 | .9868 |
| sylvine | .4995 | .8820 | .9161 | .9454 | .9524 | .9316 | .9309 | .9329 | .9154 | .9112 | .9231 | .9287 | .9329 | .9471 | .9270 | .9357 | .8994 | .9350 | .9315 | .9161 | .9638 | .9330 | .9588 | .9220 | .9320 | .9342 | .9600 |
| BNG(tic-tac-toe) | .6529 | .7831 | .7280 | .8121 | .8133 | .8014 | .8153 | .8118 | .8117 | .7703 | .7974 | .7992 | .8127 | .8131 | .8100 | .8153 | .8150 | .8139 | .8112 | .8112 | .8114 | .8080 | .8108 | .8054 | .7599 | .7841 | .8169 |
| online_shoppers | .8451 | .8459 | .8779 | .9051 | .9063 | .8633 | .8803 | .8871 | .8875 | .8905 | .8908 | .8958 | .8864 | .9019 | .8976 | .9032 | .8969 | .8868 | .8989 | .8829 | .8965 | .8998 | .9053 | .8910 | .8969 | .8543 | .9025 |
| Cardiovascular-Disease-dataset | .5003 | .6737 | .6878 | **.7343** | .7340 | .7323 | .7335 | .7307 | .7320 | .7284 | .7147 | .7298 | .7315 | .7322 | .7326 | .7321 | .7328 | .6799 | .7321 | .6677 | .7308 | .7317 | .7320 | .7220 | .7254 | .7245 | .7319 |
| credit | .5001 | .6787 | .7098 | **.7817** | .7796 | .7795 | .7580 | .7540 | .6225 | .7319 | .7469 | .7546 | .7554 | .7588 | .7561 | .7556 | .7528 | .7467 | .7550 | .7443 | .7755 | .7298 | .7440 | .7693 | .7567 | .7567 | .7625 |
| FOREX_audsgd-hour-High | .5148 | .5254 | .7041 | .6023 | .6158 | .5653 | .6985 | .7092 | .5225 | .5146 | .6413 | .5270 | .6562 | .6184 | .6884 | .6909 | .6899 | .6699 | .6512 | .6336 | .7080 | .6146 | .7109 | .5150 | .5813 | .5660 | .7019 |
| waveform-5000 | .3380 | .8421 | .8628 | .8541 | .8596 | .8623 | .8510 | .8623 | .8625 | **.8637** | .8267 | .8613 | .8602 | .8611 | .8545 | .8588 | .8532 | .8554 | .8119 | .4077 | .8604 | .8609 | .8624 | .8570 | .8591 | .8524 | .8768 |
| jungle_chess_2pcs_raw_endgame_complete | .5146 | .8814 | .6801 | .8610 | .8647 | .8262 | .9527 | .8696 | .8551 | .7515 | .8784 | .8722 | .9211 | .9761 | .9552 | .9674 | .9057 | .7144 | .9517 | .6480 | .9898 | .9734 | **.9950** | .7990 | .8210 | .8563 | .9166 |
| BNG(cmc) | .4262 | .5574 | .5239 | .5860 | **.5893** | .5730 | .5790 | .5791 | .5881 | .5624 | .5772 | .5819 | .5795 | .5850 | .5829 | .5838 | .5780 | .5273 | .5475 | .5281 | .5823 | .5801 | .5682 | .5399 | .5558 | .5597 | .5462 |
| page-blocks | .8977 | .9616 | .9565 | **.9745** | .9667 | .9703 | .9669 | .9668 | .9295 | .8715 | .9527 | .9679 | .9694 | .9671 | .9711 | .9663 | .9661 | .9640 | .9661 | .9638 | .9677 | .9632 | .9668 | .9550 | .9727 | .9713 | .9680 |
| segment | .1429 | .8896 | .8297 | .9149 | .9202 | .9196 | .9186 | .7957 | .8420 | .8792 | .9130 | .9185 | .9137 | .9039 | .9188 | .8906 | .9139 | .9052 | .2404 | .9190 | .9059 | **.9277** | .8920 | .9234 | .9163 | .9163 | .9221 |
| website_phishing | .5166 | .8672 | .8672 | .8986 | .9043 | .8942 | .8731 | .8873 | .8962 | .8578 | .8391 | .8728 | .8895 | .9014 | .8677 | **.9087** | .8681 | .8972 | .8145 | .8745 | .8950 | .8723 | .8753 | .8728 | .8994 | .8893 | .8937 |
| baseball | .9067 | .9254 | **.9552** | .9398 | .9388 | .9460 | .9254 | .9313 | .9164 | .9239 | .9231 | .9328 | .9289 | .9376 | .9276 | .9358 | .9386 | .9017 | .9306 | .9415 | .9473 | .9333 | .9428 | .9370 | .9333 | .9353 | .9291 |
| pendigits | .1041 | .9909 | .9879 | .9414 | .9913 | .9879 | **.9950** | .9948 | .9807 | .9854 | .9874 | .9947 | .9929 | .9935 | .9935 | .9942 | .9916 | .9924 | .9927 | .2276 | .9943 | .9937 | .9940 | .9941 | .9888 | .9904 | .9932 |
| Gender_Gap_in_Spanish_WP | .5884 | .5726 | .5888 | .5987 | .5926 | .5959 | **.6009** | .5961 | .5917 | .5868 | .5874 | .5956 | .5900 | .5986 | .5951 | .5895 | .5957 | .5809 | .5966 | .5884 | .5914 | .5941 | .5925 | .5729 | .5900 | .5902 | .5716 |
| wine-quality-white | .4490 | .6337 | .5306 | .6242 | .6315 | .5779 | .5779 | .5430 | .5235 | .4956 | .5752 | .5498 | .5517 | .6093 | .5586 | .5884 | .5308 | .5531 | .3165 | .6265 | .5327 | .6327 | .6287 | .5795 | .5980 | .5980 | **.6408** |
| satimage | .2379 | .9075 | .8367 | .9142 | .9135 | .9086 | .8993 | .9044 | .8612 | .8654 | .8790 | .8923 | .8979 | .8959 | .8970 | .9045 | .9041 | .8631 | .8953 | .8981 | .9131 | .8947 | .9104 | .9127 | .9077 | **.9168** | .9098 |
| mfeat-fourier | .1000 | .8650 | .8367 | .8572 | **.8718** | .8602 | .8445 | .8597 | .8330 | .8340 | .7742 | .8483 | .8407 | .8523 | .8390 | .8467 | .8388 | .8188 | .8457 | .8425 | .8653 | .8555 | .8648 | .8573 | .8363 | .8570 | .8625 |
| VulNoneVul×10⁻¹ | 1.021 | .9969 | 1.059 | 1.010 | 1.044 | 1.023 | 1.011 | 1.022 | 1.011 | 1.025 | 1.012 | 1.023 | 1.026 | 1.023 | 1.020 | 1.041 | 1.046 | 1.025 | **.9904** | 1.015 | 1.024 | 1.014 | 1.036 | | | | 1.015 |
| CookbookReviews×10 | .1536 | .1491 | .1532 | **.1483** | .1486 | .1488 | .1504 | .1520 | .1497 | .1567 | .1500 | .1498 | .1594 | .1520 | .1544 | .1511 | .1621 | .1546 | .1496 | .1592 | .1549 | .1562 | .1487 | | | | .1498 |
| kin8nm×10⁻¹ | 2.631 | 1.205 | 2.054 | 1.322 | .9017 | 1.492 | .6877 | .6699 | .9350 | 2.620 | .8301 | .7158 | 4.072 | .6736 | .6986 | .6677 | .7429 | 2.619 | .6781 | 1.111 | .6639 | .6924 | .6968 | | | | 1.784 |
| Ailerons×10⁻³ | .3993 | .2047 | .1696 | .1527 | **.1465** | .1559 | .1563 | .1572 | .1519 | .3832 | .1641 | .1564 | .3841 | .1554 | .1553 | .2719 | .1627 | .3981 | .1554 | .2377 | .1522 | .1528 | .1576 | | | | .1512 |
| Superconductivty×10 | 3.427 | 1.078 | 1.838 | .9587 | **.9911** | 1.057 | 1.044 | 1.050 | 1.269 | 3.361 | 1.276 | 1.088 | 3.713 | 1.065 | 1.080 | 1.031 | 1.127 | 3.417 | 1.079 | 1.478 | 1.032 | 1.061 | 1.038 | | | | 1.046 |
| IEEE80211aa-GATS×10⁻¹ | 3.131 | .8366 | .5894 | .4063 | .3483 | .4478 | .2844 | .2651 | .5025 | 3.200 | .5711 | .2896 | 3.396 | .2959 | .3136 | .2605 | .3459 | 3.254 | .3248 | .7772 | **.1991** | .3168 | .2629 | | | | .6934 |
| house_16H_reg×10⁵ | .5300 | .3623 | .4819 | .3136 | **.2994** | .3278 | .3175 | .3137 | .3483 | .5397 | .3533 | .3187 | .5458 | .3067 | .3123 | .3133 | .3280 | .5404 | .3158 | .5156 | .3054 | .3223 | .3102 | | | | .3276 |
| mv×10⁻² | 1045. | 150.2 | 542.3 | 11.94 | 8.548 | 8.319 | 3.949 | 13.10 | **.9159** | 1037. | 20.76 | 23.81 | 1078. | 3.095 | 4.977 | 1.624 | 7.286 | 1061. | 9.692 | 71.90 | 2.776 | 3.633 | 2.738 | | | | 3.441 |
| compass_reg | .5000 | .4146 | .4606 | .3924 | .3903 | .4249 | .4795 | .4519 | .4447 | .5252 | .4469 | .4332 | 1.065 | .4468 | .4563 | .4714 | .4491 | .5002 | .4326 | .5126 | .3554 | .4108 | **.3118** | | | | .4357 |
| law-school-admission-bianry×10⁻⁴ | 4643. | 2308. | 2816. | 328.3 | **.1620** | .2950 | 116.1 | 61.31 | 6.391 | 4668. | 115.7 | 41.43 | 7089. | 2.446 | 28.70 | 6.369 | 6.497 | 5142. | 20.51 | 2101. | 1.929 | 8.179 | 183.0 | | | | 8.706 |
| KDD | 1.000 | .8591 | .8153 | **.7066** | .7070 | .7168 | .9204 | .8083 | .7371 | 1.036 | .7339 | .8121 | 1.023 | .7158 | .7812 | .8344 | .8100 | 1.044 | .7902 | 1.003 | .8765 | .7193 | .7224 | | | | .7330 |
| Large-scale_Wave_Energy_Farm_Sydney_49×10⁴ | 7.019 | 1.176 | 1.465 | .4923 | .4498 | .6289 | .5998 | .5980 | .5431 | 7.202 | 1.059 | .5898 | 9.119 | .4370 | .5620 | .7411 | .7012 | 6.984 | .5841 | 1.295 | **.3894** | .4144 | 4.740 | | | | 4.183 |
| healthcare_insurance_expenses×10⁴ | 1.250 | .5525 | .6534 | .4665 | .4736 | **.4522** | .5049 | .4790 | .4677 | 1.283 | .5636 | .4869 | 1.673 | .4616 | .4721 | .4874 | .5219 | 1.248 | .4763 | 1.116 | .4624 | .5255 | .4699 | | | | .5296 |
| communities_and_crime | .2213 | .1348 | .1336 | .1325 | .1336 | .1337 | .1338 | .1425 | .1363 | .2294 | .1530 | .1355 | .3889 | .1369 | .1386 | .1361 | .1366 | .2218 | .1414 | .1449 | .1437 | .1327 | .1390 | | | | **.1322** |
| bank8FM×10⁻¹ | 1.551 | .4932 | .4088 | .3009 | .2878 | .3195 | .2973 | .2889 | | 1.549 | .3130 | .2942 | 1.588 | .2863 | .2909 | .2876 | .2911 | 1.544 | **.2848** | .3879 | .2925 | .2856 | .2921 | | | | .3240 |
| fried×10 | .4958 | .1854 | .2643 | .1079 | .1010 | .1395 | .1022 | .1029 | **.1007** | .4945 | .1032 | .1041 | .4949 | .1009 | .1018 | .1010 | .1048 | .4935 | .1026 | .1144 | .1017 | **.1006** | .1011 | | | | .1326 |
| archive×10³ | .4612 | .3642 | .4113 | .3443 | **.3336** | .3466 | .3629 | .3936 | .3448 | .4598 | .4307 | .3739 | .4612 | .3907 | .3730 | .3986 | .3598 | .4653 | .3611 | .4067 | .3875 | .3812 | .3944 | | | | .3733 |
| avg. rank | 23.227 | 16.455 | 18.591 | 8.045 | 7.000 | 10.727 | 10.977 | 11.159 | 13.814 | 21.932 | 18.273 | 12.864 | 17.568 | 8.591 | 11.341 | 10.727 | 13.091 | 18.318 | 12.364 | 19.023 | 7.545 | 10.909 | 8.205 | 15.889 | 12.370 | 13.370 | 9.705 |

Table 1: We extend our experiments by incorporating the recently introduced Tabular Benchmark. Specifically, we utilize the official "Tiny Benchmark subset for Rank Consistent Evaluation" from TabBench, which comprises 44 datasets (27 for classification and 17 for regression). These datasets demonstrate consistency in method ranking across a larger pool of 300 datasets. While other methods require 100 trials of hyperparameter tuning, TabPTM achieves competitive results through lightweight fine-tuning alone. (Note that TabPFN, TuneTables, and LoCalPFN primarily focus on classification tasks, and when ranking regression datasets, they report the median score. **TabPTM outperforms the KNN-based improved version of PFN in ranking.**)

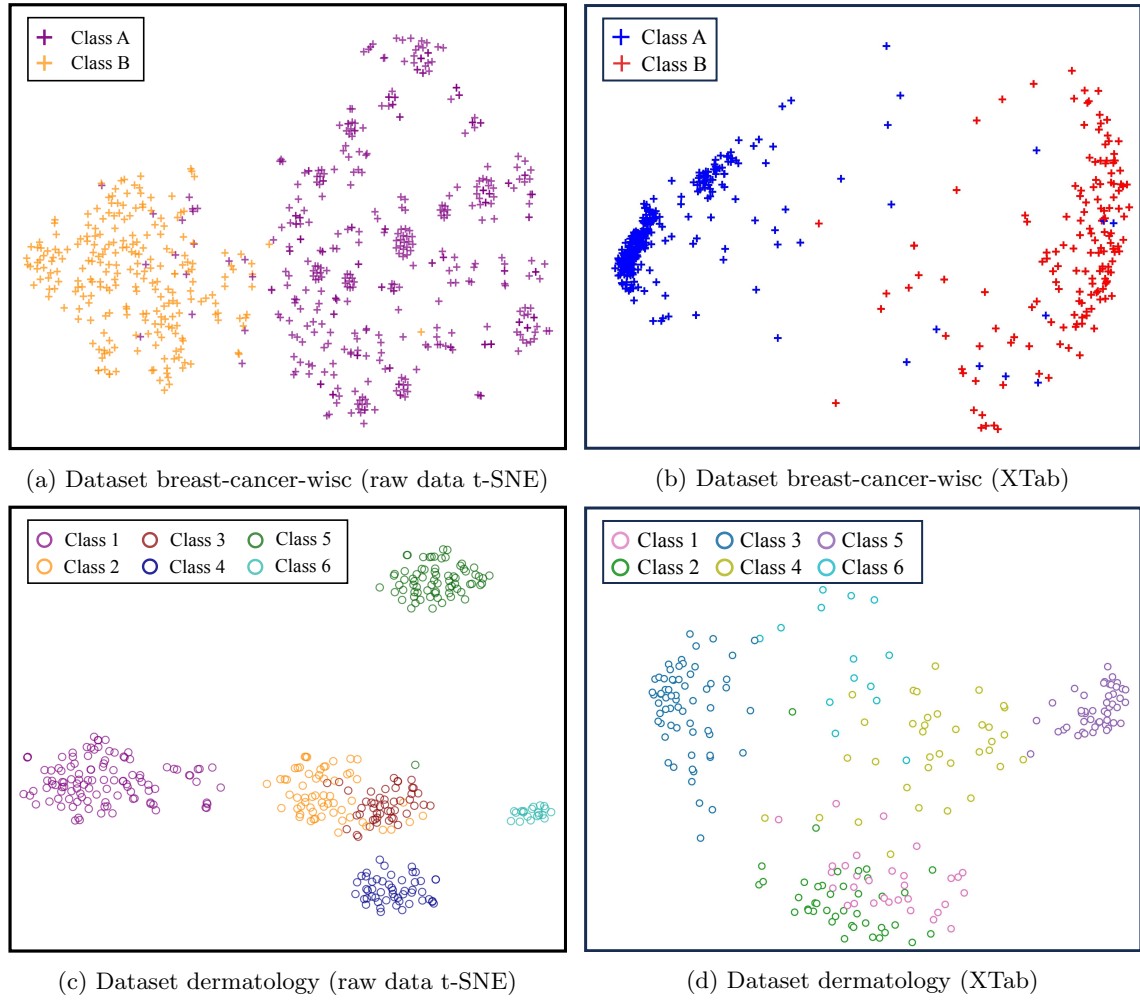

(a) Dataset breast-cancer-wisc (raw data t-SNE)

(b) Dataset breast-cancer-wisc (XTab)

(c) Dataset dermatology (raw data t-SNE)

(d) Dataset dermatology (XTab)

Figure 1: In subfigures (a) and (c), we visualized the distribution of the raw data using t-SNE. To extend visualizations to other methods, we specifically examined XTab, another pre-trained method for tabular data. We applied average pooling to the transformer's output to obtain a condensed 32-dimensional representation for each sample. These processed representations, which are input into the classification heads, are shown in subfigures (b) and (d). However, the representations from XTab did not show a significant improvement over the raw data. This suggests that much of the representational learning in XTab may rely heavily on the model's top layers (the classification module), rather than developing robust intermediate representations. In contrast, as shown in Figure 3c of the main paper, the meta-representations generated by TabPTM successfully capture relational patterns within the dataset.

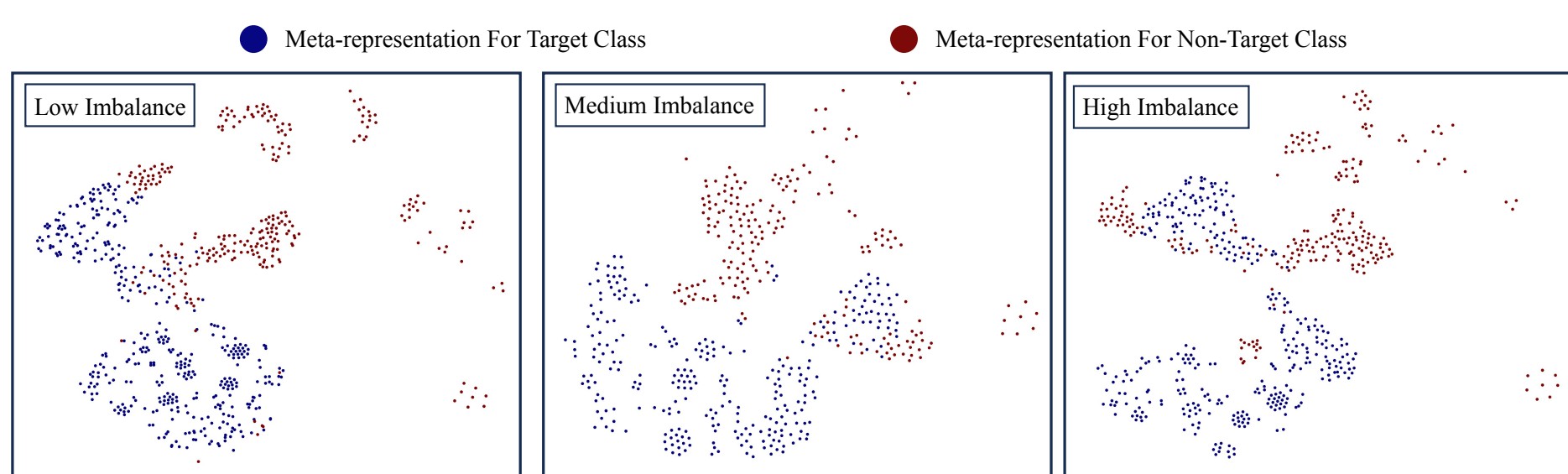

Figure 2: To explore the effect of class imbalance on Meta-representation, we utilized the "breast-cancer-wis" binary classification dataset. We manipulated the dataset to reflect three levels of class imbalance: Low (majority:minority = 0.6), Medium (majority:minority = 0.4), and High (majority:minority = 0.2). The Meta-representations for both the Target and Non-Target classes were visualized under these varying conditions. Despite the imbalance, Meta-representations maintained a good level of discriminative ability. As the imbalance intensified, the discriminative effectiveness for challenging-to-classify samples decreased. This was visually apparent as the red (Target) and blue (Non-Target) points became more intermixed, indicating that the neighboring space, dominated by the majority class, diminishes distinctiveness between classes. The model's exposure to diverse distributions from large-scale datasets helps it capture essential patterns even under imbalanced conditions. Thus, while class imbalance poses challenges, TabPTM's performance remains comparatively stable.