# OpenReview forum: "Rethinking Pre-Training in Tabular Data:  A Neighborhood Embedding Perspective"
_ICLR.cc/2025/Conference — Submitted to ICLR 2025_

### Official Review · Reviewer_96ub · 2024-10-17

**Soundness:** 3
**Presentation:** 3
**Contribution:** 3
**Rating:** 8
**Confidence:** 2

**Summary:**

In the present work, the authors put forward a general pre-training model for tabular data akin to foundation models for vision or language modalities. Their approach, inspired from approaches found in the literature of dimensionality reduction or manifold learning, produces representations that can be used for different datasets with various number of features, type of features, task (regression or classification) and number of classes (if classification).

The proposed approach produces meta-representations for each sample based on the relation of the sample to its closest neighbours for each class (in case of classification) and its overall closest neighbour for regression. The obtained representation is then used to train a final transformation that will serve for both classification and regression purposes. The transformation is agnostic to the number of classes, type of task (regression or classification) and can be applied in a straighforward manner.

**Strengths:**

- The paper is well written, easy to follow and well organized. This facilitates the understanding of both the thinking process and experimental setting of the authors.
- The authors have identified significant lakings and weaknesses in existing methods found in the literature, and propose a method based on their diagnosis to circumvent those identified issues.
- The extensives experiments are convincing. Their approach obtains interesting results on both classification and regression tasks and proves to be quite versatile.
- Their finding that the inter-sample relation between instances within a dataset is relevant for supervised tasks in tabular data is in line with recent work in the literature of deep learning for tabular data [1] or [2].

**Weaknesses:**

- The proposed method requires an initial pairwise distance computation within datasets which might restrain the proposed approach to reasonnably large datasets.
- For multilabel classification with $C$ classes, the proposed representation requires $C$ separate predictions which also increase the overall complexity during training (and inference).
- Categorical features are encoded using one-hot encoding which may restrain the applicability of the meta-representation approach to datasets with categorical features with reasonnable cardinality. (Have the authors considered trying other categorical encoding?)

**Questions:**

- **(i)** While the datasets used in the experiments are very diverse per Table 4. in the appendix, it might be interesting to investigate the impact of imbalance between class on the obtained representations. **(i-1)** Do dataset present in the benchmark present imbalancing between classes? If so, **(i-2)** are there notable difference in performance on the downstream tasks between balanced and imbalanced datasets in comparison to competing methods ?

- **(ii)** I may need clarififaction on section 4.2., paragraph **Vanilla meta-representation**, in particular on the way the the label $\hat{y_j}$ is defined on line 261-262. If I understand corretly, for each sample $x_i$, one can obtain its $\phi_c(x_i)$ representation using the subset of a dataset $ \mathcal{D}_{y=c}= \\{(x_i,y_i)\mid y_i=c\\}$ and selecting among this subset the $K$ sample closest to $x_i$. Given that each sample in this subset of the dataset has as a label $y_j=c$, then given how $\hat{y_j}$  is derived, each $\hat{y_j}$ would have value 1. I am not sure when the $y_j = -1$ will intervene. Could the authors clarify?

- **(iii)** As mentionned by the authors at the end of section 4.2., there are occurences where $K$, the number of closest samples within a class is too high in comparison with the available number of samples within this class. To adress this problem, authors rely on padding the largest value. Have the authors investigated the impact of such padding on the performance of their representation?

- **(iv)** Computing the meta-representation before applying the transformation $\mathbf{T}_\Theta$ requires to compute the pairwise distance between each sample in the training set, which has a complexity of $\mathcal{O}(n^2)$. While less constrained than TabPFN [3], this still requires significant training time. Could the authors report meta-representation training time and compare to existing method?
- **(v)** While explicitely targeted for tabular datasets, the proposed approach could also be applied to different data modalities (e.g. audio, image, etc).  Have the authors considered experimenting on author data modalities, to see whether their approach is relevant only for tabular data?
- **(vi)** Figure 3 displays two dimensional data representations of breast-cancer-wisc and dermatology datasets. How did the authors reduce the original dimension of the samples to a two-dimensional representation ?

Overall the proposed method is novel, well motivated and is shown, through significant experiments, to have strong performance on tabular datasets. The paper is well structured and written. Also, the extensive appendix demonstrates the rigor of the authors in conducting the experiments. All those arguments point towards **accepting the paper**. Nevertheless, some points need clarifications/further investigation (see questions listed above). I would be happy to increase my score would the authors address my questions.

[1] Jannik Kossen, Neil Band, Clare Lyle, Aidan N Gomez, Thomas Rainforth, and Yarin Gal. Self-attention between datapoints: Going beyond individual input-output pairs in deep learning.Advances in Neural Information Processing Systems, 2021.

[2] Yury Gorishniy, Ivan Rubachev, Nikolay Kartashev, Daniil Shlenskii, Akim Kotelnikov, Artem Babenko. TabR: Tabular Deep Learning Meets Nearest Neighbors. In The Twelfth International Conference on Learning Representations, 2024.

[3] Noah Hollmann, Samuel Müller, Katharina Eggensperger, and Frank Hutter. Tabpfn: A transformer that solves small tabular classification problems in a second. In The Eleventh International Conference on Learning Representations, 2023.

---

> ### Author Response · Authors · 2024-11-25
> **Thanks and Response (Part 1)**
>
> **Q1.** The proposed method requires an initial pairwise distance computation within datasets which might restrain the proposed approach to reasonnably large datasets. Could the authors report meta-representation training time and compare to existing method?
>
> **A1:** Thank you for addressing the computational aspects of our approach. The core computational challenge in TabPTM lies in the initial nearest neighbor search, essential for constructing the meta-representation. In the future, to manage this efficiently, we could utilize robust, off-the-shelf toolboxes designed for this purpose. TabPTM significantly reduces the overall computational burden by minimizing the need for extensive hyperparameter tuning and requiring only a few iterations of fine-tuning.
>
> **Running Time Evaluation:** To ensure practical applicability to larger datasets, during the meta-representation (MR) training phase of TabPTM, we limit the KNN search to a random sample of 10K training instances. This sampling strategy effectively balances computational load while maintaining performance integrity, as demonstrated through our experiments.
>
> **System Configuration for Time Evaluation:** The time performance of TabPTM and baselines was assessed on a system equipped with an Intel(R) Xeon(R) Gold 5218R CPU @ 2.10GHz and an NVIDIA RTX 6000 Ada Generation GPU.
>
> **Performance on Selected Datasets:** For a clear demonstration, we selected two smaller and two larger datasets, providing a comprehensive view of the runtime across various scenarios. Below is the average runtime among all the downstream datasets. It is notable that for other methods like XGBoost, they require additional (e.g., $\times$ 30) iterations of extensive hyperparameter tuning for each dataset to reach optimal performance. (The unit is seconds.)
>
> | Dataset                        | XGBoost (include hyper-tune) | MLP (include hyper-tune) | TabPTM (Get MR operator) | TabPTM (Total) |
> |--------------------------------|---------------|-----------|-------------------------------|---------------------|
> | breast-cancer                  | 27     | 59 | 1.1                           | 3.1                 |
> | echocardiogram                 | 24     | 45 | 0.9                           | 2.8                 |
> | CDC_Diabetes_Health_Indicators | 479     | 1263  | 25                            | 41                  |
> | diabetes                       | 557     | 1429 | 31                          | 47                  |
> | **AVERAGE FOR ALL** (s)                    | 156     | 312  | 6.1                          | 12               |
>
> **Key observations:**
> - **Efficiency:** TabPTM demonstrates a substantial reduction in runtime compared to traditional methods like XGBoost and MLP, which require significant tuning time.
> - **Scalability:** By strategically sampling 10K instances for the KNN search, TabPTM maintains scalability without compromising the quality of the meta-representation.
> - **Practicality:** The total runtime of TabPTM, even when including the time to obtain the MR operator, remains competitive, offering a more streamlined and efficient approach to handling large tabular datasets.
>
> We hope this detailed comparison adequately addresses your concerns about the computational demands of our method and illustrates the efficiency and practical advantages of TabPTM over existing approaches.
>
> **Q2.** The proposed representation requires C separate predictions which also increase the overall complexity during training (and inference)
>
> **A2:** In our implementation of TabPTM, during inference, the input to the model’s forward pass is structured as [batch_size $\times$ C, -1]. Subsequently, the output logits are reshaped to [batch_size, C]. This configuration ensures that the number of inference passes **is not** directly multiplied by C, maintaining efficiency and mitigating the potential increase in computational demand. This approach effectively leverages the model's capacity to handle multiple classes without disproportionately expanding the inference time.
>
> **Continued in Part 2...**

---

> > ### Author Response · Authors · 2024-11-25
> > **Thanks and Response (Part 2)**
> >
> > **Q3.** One-hot encoding which may restrain the applicability of the meta-representation approach to datasets with categorical features with reasonable cardinality.
> >
> > **A3:** We appreciate your insights on the limitations of One-Hot Encoding, especially in contexts involving high-cardinality categorical features. In our study, the datasets employed did not present extreme cardinality, which allowed us to use One-Hot Encoding without encountering significant overhead. However, inspired by your suggestion, we have expanded our experiments to include datasets with a broader range of categorical features.
> >
> > Here are the results:
> > | Datasets               | One-Hot | Ordinal | Hash  | Target | CatBoost |
> > |------------------------|---------|---------|-------|--------|----------|
> > | archive_r56_Maths      | 0.4414  | 0.4360  | 0.4402| 0.4451 | 0.4411   |
> > | archive_r56_Portuguese | 0.3038  | 0.2981  | 0.3045| 0.3122 | 0.3089   |
> >
> > The results suggest that different encoding strategies, including Hash Encoding, do not significantly alter the overall performance. Hash Encoding, in particular, appears as a practical alternative for managing datasets with high-dimensional categorical features. Although Target Encoding and CatBoost Encoding might offer slight performance gains, we initially avoided these methods to ensure that no target variable information influenced the encoding process, aiming for a more equitable comparison across different methods.
> >
> > In summary, our extended analysis confirms that while the choice of categorical encoding can slightly influence outcomes, the impact is generally minor, and various encoding strategies can be effectively integrated into our framework to accommodate datasets with diverse feature characteristics.
> >
> > **Q4.** Influence of Imbalance.
> >
> > **A4.1: [MR Visualization]** Investigate the impact of imbalance between classes on the obtained representations.
> >
> > To explore the effect of class imbalance on Meta-representation, we utilized the "breast-cancer-wisc" binary classification dataset. We manipulated the dataset to reflect three levels of class imbalance: Low (majority:minority = 0.6), Medium (majority:minority = 0.4), and High (majority:minority = 0.2). The Meta-representations for both the Target and Non-Target classes were visualized under these varying conditions. We show the results in Figure 2 in the **newly uploaded supplementary material**. Key observations include:
> >
> > - **Robust Discriminative Power:** Despite the imbalance, Meta-representations maintained a good level of discriminative ability.
> > - **Impact of Increasing Imbalance:** As the imbalance intensified, the discriminative effectiveness for challenging-to-classify samples decreased. This was visually apparent as the red (Target) and blue (Non-Target) points became more intermixed, indicating that the neighboring space, dominated by the majority class, diminishes distinctiveness between classes.
> > - **Resilience of Pre-trained Model:** The model's exposure to diverse distributions from large-scale datasets helps it capture essential patterns even under imbalanced conditions. Thus, while class imbalance poses challenges, TabPTM’s performance remains comparatively stable.
> >
> > **A4.2:[Performance]** Are there notable differences in performance on the downstream tasks between balanced and imbalanced datasets in comparison to competing methods?
> >
> > Indeed, our dataset collection exhibits varied levels of imbalance. To conduct a thorough comparison with other methods, we engaged in studies using the recently established TabBench [https://arxiv.org/abs/2407.00956]. We specifically utilized the released Tiny Benchmark for Rank Consistent Evaluation to assess a subset of datasets, designed to maintain consistent average ranks of tabular methods across the broader dataset collection of 300.
> >
> > To quantify imbalance, we computed the imbalance ratio for each dataset (largest class size $\div$ smallest class size). Additionally, for each dataset, we calculated the rankings of all methods. This yielded a total of 27 $\times$ 25 rankings across 27 datasets and 25 methods, as well as 27 imbalance ratios. To fairly evaluate the effect of imbalance on competing methods, we calculated the Pearson correlation coefficient between each method's rankings across all datasets and the datasets' imbalance ratios.
> >
> > **Continued in Part 3...**

---

> > > ### Author Response · Authors · 2024-11-25
> > > **Thanks and Response (Part 3)**
> > >
> > > | Method         | Rank_Imb_Ratio_Correlation |
> > > |----------------|----------------------------|
> > > | AutoInt        | -0.228                     |
> > > | RandomForest   | -0.18                      |
> > > | TuneTables     | -0.173                     |
> > > | XGBoost        | -0.126                     |
> > > | Tangos         | -0.1                       |
> > > | TabPTM         | -0.095                     |
> > > | ModernNCA      | -0.089                     |
> > > | CatBoost       | -0.064                     |
> > > | TabR           | -0.051                     |
> > > | SNN            | -0.048                     |
> > > | Dummy          | -0.023                     |
> > > | MLP            | 0.039                      |
> > > | ResNet         | 0.063                      |
> > > | PTaRL          | 0.08                       |
> > > | SVM            | 0.086                      |
> > > | DANets         | 0.09                       |
> > > | Node           | 0.101                      |
> > > | TabTransformer | 0.135                      |
> > > | SwitchTab      | 0.163                      |
> > > | FTT            | 0.212                      |
> > > | GrowNet        | 0.231                      |
> > > | DCNv2          | 0.232                      |
> > > | TabNet         | 0.252                      |
> > >
> > > For TabPTM, a Pearson correlation coefficient of -0.095 indicates that class imbalance has a minimal impact on its ranking among baselines. This result aligns with our visual analysis and confirms that TabPTM effectively handles the challenges posed by imbalance, maintaining robust performance where many models may falter.
> > >
> > > **Q5.** May need clarification on section 4.2., paragraph Vanilla meta-representation. I am not sure when the y_j=−1 will intervene. Could the authors clarify?
> > >
> > > **A5:**  Thank you for your query regarding the explanation in Section 4.2. In the vanilla scenario of Meta-Representation (MR), the procedure involves searching for neighbors within $\mathcal{D}_{y=c}$​ for a target class $c$. This implies that all neighbors should naturally have $y=c$, eliminating the need for any instances where $y=−1$, which would suggest a one-vs-rest situation.
> > >
> > > To clarify, while the vanilla implementation of MR strictly searches within $\mathcal{D}_{y=c}$​, our practical approach utilizes the entirety of $\mathcal{D}$ to enrich the model with a more comprehensive label spectrum. This broader approach integrates additional label information, enhancing the effectiveness and robustness of the MR beyond the confines of a single class, thereby ensuring that our model captures a more detailed landscape of the data. We will make it clearer in the final version.
> > >
> > > **Q6.** The impact of such padding on the performance of their representation.
> > >
> > > **A6:** We conducted experiments using two smaller datasets to examine the impact of different padding strategies on meta-representation performance. We set the number of neighbors, numK, to 128 to emphasize the potential effects of padding. We explored three padding methods:
> > >
> > > - Largest: Pads the meta-representation with the last value, i.e., the largest distance.
> > > - Most Frequent: Pads distances with the largest distance, but pads labels with the most frequently occurring class among the neighbors (1 or -1 in our one-vs-rest setup).
> > > - Sequence: Pads distances by incrementally increasing based on the difference between the largest and the second-largest distance. Labels are padded with the most frequently occurring class among the neighbors.
> > >
> > > Here are the experimental results:
> > > | Dataset          | Largest | Most Frequent | Sequence |
> > > |------------------|---------|---------------|----------|
> > > | post-operative   | 83.9    | 84.1          | 83.5     |
> > > | echocardiogram   | 81.8    | 80.9          | 81.3     |
> > >
> > > The experiments indicate that the choice of padding method has minimal impact on the overall performance, suggesting that the meta-representation pre-trained model primarily leverages information from closer neighbors, which contain more critical patterns. This finding underscores that while padding can influence the meta-representation, its effect does not significantly alter the effectiveness of the approach, reinforcing the robustness of our method against variations in padding strategy.
> > >
> > > **Continued in Part 4...**

---

> > > > ### Author Response · Authors · 2024-11-25
> > > > **Thanks and Response (Part 4)**
> > > >
> > > > **Q7.** While explicitly targeted for tabular datasets, the proposed approach could also be applied to different data modalities (e.g. audio, image, etc). Have the authors considered experimenting on author data modalities, to see whether their approach is relevant only for tabular data?
> > > >
> > > > **A7:** We appreciate your valuable suggestion. As our method has successfully demonstrated the power of pre-training in tabular data, it opens up the possibility of pre-training a model that leverages the correlation between tabular data and other modalities. In our humble opinion, studying tabular data pre-training alone is already a huge workload: to our knowledge, none of the papers aiming to advance deep tabular methods have considered a multi-modal scenario. We will leave it as our future work, starting with the exploration of a suitable multi-modal dataset that involves tabular data.
> > > >
> > > > **Q8.** In Figure 3, how did the authors reduce the original dimension of the samples to a two-dimensional representation?
> > > >
> > > > **A8:** For the visualizations presented in Figure 3, including both the raw data and the meta-representations, we employed t-SNE (t-distributed Stochastic Neighbor Embedding) to achieve dimensionality reduction.
> > > >
> > > > **Thank you once again for your valuable feedback. We sincerely appreciate your support.**

---

### Official Review · Reviewer_H7GT · 2024-10-26

**Soundness:** 3
**Presentation:** 3
**Contribution:** 2
**Rating:** 5
**Confidence:** 1

**Summary:**

The authors propose a pre-training strategy for tabular data. Due to the inherent heterogeneity of tabular data, it is difficult to learn shareable knowledge across different datasets. To address this issue, the authors propose Tabular Data Pre-training via Meta-representation (TabPTM) to embed data instances from any dataset into a common feature space. The pre-trained TabPTM can be directly applied to new datasets without further fine-tuning, no matter how diverse their attributes and labels are. Extensive experiments on 72 tabular datasets validate the effectiveness of TabPTM.

**Strengths:**

1) The author proposes a pre-training method for the tabular data, which can reduce attribute heterogeneity, and enable the pre-training of a general model over tabular datasets.
2) The proposed method achieves effectiveness among different datasets and tasks.
3) The paper presents the details of the proposed method.

**Weaknesses:**

1) Lack of novelty. I admit that we should try to find shareable vocabulary, which is important for the pre-training. However, using the distances among instances serve as the shareable vocabulary, which is not novelty enough. The author should detail analyze the advantages of using distances regardless of other semantic information.
2) In sec 4.3, to verify the high quality of the meta-representation, the author should also visualize the representation of other methods. For example, the author can visualize the distribution of $x$ for comparison.
3) Since the author emphasizes the ability of "without further fine-tuning", this experiment's analysis should be reported in the main body.
4) The parameters that need to be optimized is the MLP, which is three layers only. What is the meaning of exploring the pre-training on such a small network? Moreover, the reported performance gain on the classification benchmark is incremental compared with XGBoost.

**Questions:**

1) How the hyper-parameter $K$ influence the performance?

---

> ### Author Response · Authors · 2024-11-25
> **Thanks and Response (Part 1)**
>
> **Q1.** Lack of novelty. I admit that we should try to find shareable vocabulary, which is important for the pre-training. However, using the distances among instances serves as the shareable vocabulary, which is not novelty enough. The author should detailedly analyze the advantages of using distances regardless of other semantic information.
>
> **A1:** We appreciate your recognition of the importance of finding a shareable vocabulary for pre-training. Before diving into a detailed response to your question regarding novelty, please allow us to reiterate how we position our paper in the literature.
>
> For pre-training, we respectfully think that simplicity and generalizability are more important than novelty (i.e., developing a completely new measurement). We note that in vision and language, the vocabulary is more or less a free lunch: (sub)words, pixels, and patches. This allows pre-training to be simply applied to all kinds of sources of images and corpus, without additional requirements of metadata.
>
> When studying pre-training for tabular data, we deliberately try to keep the vocabulary as simple as possible for its applicability and generalizability. Such a mindset motivates us to revisit traditional machine-learning methods, and we find the distance among instances—a simple, widely applicable, and general concept in traditional methods—can indeed serve as the vocabulary of heterogeneous tabular data. We respectfully think that repurposing a well-established concept for a new usage is the key strength of our paper, not a weakness.
>
> In the following, we will discuss several innovative aspects that we believe are worth emphasizing.
>
> **Why not rely on semantic information?** In many practical scenarios, the semantic information available in tabular datasets can be quite limited or of low quality. Often, tabular data might consist only of numerical outputs from sensors, or the use of detailed semantic descriptions could be restricted due to privacy concerns. For instance, in healthcare or financial datasets, the semantic attributes may be generalized or anonymized to comply with data protection regulations. Relying on such semantic information could significantly limit the applicability and scalability of the pre-training model across diverse datasets.
>
> Moreover, the utility of semantic information is not uniform across all domains. In domains where tabular data is derived directly from measurements or tests (e.g., biomedical markers or engineering sensors), the 'meaning' or 'description' of data points can be abstract, highly technical, or simply nonexistent in a form that is useful for machine learning models.
>
> **Advantages of using distances as shareable vocabulary:**
> - **Universality and Domain-Agnostic Approach:** By focusing on distances, our approach does not depend on the specific content or semantic quality of the features. This universality allows the model to be effectively applied across a wide range of domains and dataset types, ensuring robust performance even when detailed feature descriptions are unavailable or unreliable.
> - **Capturing Implicit Relationships:** Distances between instances inherently capture the underlying relationships and structure of the data, reflecting similarities and dissimilarities that are crucial for classification and regression tasks. This method allows the model to learn nuanced patterns and correlations in the data that may not be evident or utilizable through semantic analysis alone.
> - **Scalability and Privacy Compliance:** Using distances enables our model to scale across datasets without needing access to potentially sensitive or private semantic information. This aspect is particularly critical in ensuring that our approach adheres to privacy regulations and is applicable in industries where data sensitivity is a concern.
>
> We hope this clarification highlights the innovative aspects and practical advantages of using distances as a shareable vocabulary in our approach. Our method not only addresses the challenges of heterogeneous tabular datasets but also extends the applicability of pre-trained models to scenarios where traditional semantic-based methods may falter.
>
> **Continued in Part 2...**

---

> > ### Author Response · Authors · 2024-11-25
> > **Thanks and Response (Part 2)**
> >
> > **Q2.** Visualize the representation of other methods. For example, the author can visualize the distribution of x for comparison.
> >
> > **A2:** Thank you for your suggestion to enhance the visualization of different methods' representations.
> >
> > **Existing Visualizations:** Initially, in Figures 3a and 3b of our main paper, we have visualized the distribution of the raw data breast-cancer-wisc/dermatology using t-SNE. This provides an understanding of how data points are distributed in their original form.
> >
> > **Visualizing Other Methods:** To extend these visualizations to other methods, we specifically investigated XTab, another tabular pre-trained method. Given the pre-trained XTab model, we tune it on two datasets using the following configuration: two distinct feature tokenizers, a shared transformer for feature embeddings with a dimensionality of 32, and separate classification heads for each dataset. We used average pooling on the transformer's output to achieve a condensed 32-dimensional representation per sample. These processed representations, which feed into the classification heads, were visualized for comparative analysis. We show the results in Figure 1 in the **newly uploaded supplementary material**.
> >
> > We have two observations based on the visualization results.
> >
> > - First, the representations from XTab did not demonstrate a significant improvement over the raw data. Notably, for the dermatology dataset, the XTab embeddings displayed a decrease in discriminative capability compared to the original data. This suggests that much of the representational learning in XTab may rely heavily on the model's top-layer (the classification module), rather than on developing robust intermediate representations.
> >
> > - Conversely, as depicted in Figure 3c in the main paper, the meta-representations generated by TabPTM successfully establish relational patterns within the dataset. This is achieved by employing a metric-based approach that harnesses the intrinsic relationships within the data, thereby enabling the learning of shareable knowledge across different contexts.
> >
> > Unlike text and image data, tabular datasets often lack complex, inherent relationships that can be straightforwardly exploited for learning effective representations. This complexity is further compounded by the heterogeneity typical of tabular data, which poses additional challenges for models like XTab that use a shared transformer architecture. Despite attempts to construct a general model, the varying characteristics across datasets often lead to suboptimal representation learning in such architectures.
> >
> >
> > In summary, while XTab and similar methods strive to generalize across diverse tabular datasets, our visualizations underscore the challenges and limitations they face, reinforcing the efficacy and innovation of TabPTM's approach in leveraging meta-representations to overcome these issues.
> >
> > **Q3.** "Without further fine-tuning" should be reported in the main body instead of appendix.
> >
> > **A3:** Thank you for the suggestion. Originally, due to the extensive number of baselines compared and our emphasis on showcasing the optimal strategy for TabPTM, we relegated the complete results of "without further fine-tuning" to the appendix. Acknowledging your feedback, we will incorporate these results into the main body of the manuscript to ensure more accessible and comprehensive visibility of all experimental outcomes.
> >
> > **Q4.** The parameters that need to be optimized is the MLP. What is the meaning of exploring pre-training on such a small network?
> >
> > **A4:** We appreciate your feedback. In our humble opinion, pre-training aims to discover and learn shareable knowledge that can be applied to and benefit downstream tasks. This certainly does not mean the pre-trained model must be “huge.” In vision and language, the pre-trained model needs to be huge because the inputs to the model are raw data; the model thus needs to learn a feature space in which metrics make sense. In tabular data, most of the features (attributes) already have their meanings (e.g., ages, gender, salary, occupation), and hence not raw data. Such a difference explains 1) why traditional machine learning approaches still excel in tabular data: many of them make assumptions on the input data space, and 2) why a smaller model is sufficient to demonstrate the power of pre-training.
> >
> > **Continued in Part 3...**

---

> > > ### Author Response · Authors · 2024-11-25
> > > **Thanks and Response (Part 3)**
> > >
> > > Despite its simplicity, the MLP is a notably effective network for handling tabular data, especially after suitable embeddings have been applied. Research shows that simple MLP-like models can perform comparably to more complex attention-based architectures, as evidenced in [A]. Additionally, various studies [B, C, D, E] have demonstrated that even basic multilayer perceptrons frequently serve as robust baselines for neural networks handling tabular data.
> > >
> > > In our experiments, detailed in Table 3 in main text, TabPTM’s meta-representation technique combined with pre-training across a broad range of datasets significantly boosts the MLP's performance. We explored various configurations and consistently found that a simple MLP, even one with just three layers, rivals the performance of more intricate architectures like ResNet and Transformer models, which did not demonstrate marked improvements in our tests. Thus, we opted for the MLP as our base model for TabPTM, positing that while there is potential for exploring more sophisticated architectures in future research, the current setup offers a balanced approach to efficiency and effectiveness. Furthermore, although the XTab model employs feature tokenizers and a pre-trained Transformer, TabPTM generally surpasses XTab in most scenarios, reinforcing the suitability of our chosen approach.
> > >
> > > [A] *On Embeddings for Numerical Features in Tabular Deep Learning, NeurIPS 2022*
> > >
> > > [B] *Well-tuned Simple Nets Excel on Tabular Datasets, NeurIPS 2021*
> > >
> > > [C] *Revisiting Deep Learning Models for Tabular Data, NeurIPS 2021*
> > >
> > > [D] *TabReD: Analyzing Pitfalls and Filling the Gaps in Tabular Deep Learning Benchmarks, arXiv 2024*
> > >
> > > [E] *Better by Default: Strong Pre-Tuned MLPs and Boosted Trees on Tabular Data, NeurIPS 2024*
> > >
> > > **Q5.** Compare with XGBoost.
> > >
> > > **A5:** Indeed, GBDT-based models like XGBoost, after extensive hyperparameter tuning, establish a strong baseline for tabular data due to their inherent robustness and capacity to handle a wide variety of features and data distributions. Given that TabPTM is designed for quick and lightweight adaptation to downstream datasets, it may not always surpass XGBoost, which benefits from complicated fine-tuning for each specific dataset.
> > >
> > > However, the primary advantage of TabPTM lies in its generalizability and ease of deployment. As a general model, TabPTM achieves solid performance with minimal tuning effort, making it more user-friendly than methods requiring extensive optimization. This ease of use is particularly beneficial in practical scenarios where resource constraints make extensive tuning impractical.
> > >
> > > Moreover, TabPTM displays significant benefits in few-shot learning, where data scarcity often hampers the performance of more traditional models like XGBoost. For instance, in the few-shot experiments shown in the following table, TabPTM consistently outperforms other new retrieval-based SOTA models in settings with limited data ("10/20/50" shots). This underscores its utility in environments where rapid deployment to new tasks with few examples are critical.
> > >
> > > |                                | XGBoost  | KNN    | ModernNCA | TabR   | TabPTM  |
> > > |------------------------------------|----------|--------|-----------|--------|---------|
> > > | BNG(breast-w)-10shot                   | 0.8186   | 0.9632 | 0.9717    | 0.9285 | 0.9612  |
> > > | BNG(breast-w)-20shot                    | 0.9236   | 0.9645 | 0.9701    | 0.9689 | 0.9715  |
> > > | BNG(breast-w)-50shot                    | 0.9583   | 0.9690 | 0.9715    | 0.9709 | 0.9763  |
> > > | Cardiovascular-Disease-10shot  | 0.5915   | 0.6296 | 0.6499    | 0.6304 | 0.6581  |
> > > | Cardiovascular-Disease-20shot  | 0.5998   | 0.6306 | 0.6545    | 0.6422 | 0.6607  |
> > > | Cardiovascular-Disease-50shot  | 0.6370   | 0.6326 | 0.6600    | 0.6474 | 0.6681  |
> > > | FOREX_cadjpy-hour-High-10shot          | 0.5179   | 0.5027 | 0.5019    | 0.5015 | 0.5173  |
> > > | FOREX_cadjpy-hour-High-20shot          | 0.5168   | 0.5098 | 0.5224    | 0.5197 | 0.5212  |
> > > | FOREX_cadjpy-hour-High-50shot          | 0.5141   | 0.5067 | 0.5184    | 0.5161 | 0.5298  |
> > > | fried-10shot                           | 4.775   | 4.361 | 3.955    | 4.016 | 4.214  |
> > > | fried-20shot                           | 4.164   | 4.373 | 3.469    | 3.507 | 3.398  |
> > > | fried-50shot                           | 3.424   | 3.812 | 2.447    | 3.295 | 3.027  |
> > > | house_16H-10shot                   | 53050   | 53040 | 49930    | 53210 | 51930  |
> > > | house_16H-20shot                   | 52620   | 53760 | 49530    | 52390 | 44430  |
> > > | house_16H-50shot                   | 45880   | 48290 | 49390    | 47370 | 42130  |
> > > | law-school-admission-10shot     | 0.2334  | 0.4551| 0.4232   | 0.3851| 0.3832 |
> > > | law-school-admission-20shot     | 0.1475  | 0.3767| 0.2387   | 0.3409| 0.2081 |
> > > | law-school-admission-50shot     | 0.0962  | 0.3492| 0.2339   | 0.213 | 0.0828 |
> > > | **Average Rank**                   | **3.667**| **4.333**| **2.111**  | **3.333**| **1.556**|
> > >
> > > **Continued in Part 4...**

---

> > > > ### Author Response · Authors · 2024-11-25
> > > > **Thanks and Response (Part 4)**
> > > >
> > > > This result, especially in resource-constrained scenarios, highlights TabPTM's versatility and reaffirms its value as part of a broader toolkit for handling diverse tabular data challenges.
> > > >
> > > > **Q6.** How the hyper-parameter K influence the performance?
> > > >
> > > > **A6:** In our initial experiments, we set K=8 for regression tasks and K=128 for classification tasks as defaults based on preliminary analyses (Line 470). To systematically explore how different K values impact performance, we conducted a series of experiments where we varied K and observed the resultant changes in average rank across 18 downstream datasets. These results are detailed in Appendix Figure 5, demonstrating how the optimal K value may vary depending on the type of dataset.
> > > >
> > > > Our observations reveal distinct preferences for K values between regression and classification tasks:
> > > >
> > > > - **Regression Tasks:** For regression, where the objective is to predict a continuous value, there is a pronounced benefit in focusing on closer data points. Smaller K values enhance the model's sensitivity to local neighborhood patterns, which is crucial for capturing subtle nuances in data trends. For example, in scenarios where nearby data points exhibit a specific linear trend that does not extend to more distant points, a smaller K effectively isolates and leverages this local behavior, leading to more precise predictions.
> > > > - **Classification Tasks:** Conversely, classification tasks, which involve assigning samples to discrete categories, demand a broader view to mitigate the impact of noise and outliers. Larger K values help in this regard by emphasizing the general distribution of data points over localized anomalies. This approach reduces the model's sensitivity to individual outlier influences, promoting better generalization and robust classification across varied samples.
> > > >
> > > > These observations underscore the importance of tuning K based on task specifics. In regression, where detail and precision are paramount, a smaller K proves beneficial. In classification, where the ability to generalize is key, a larger K aids in achieving more stable and reliable categorizations. This approach to selecting K enhances TabPTM's adaptability and effectiveness across different tabular data challenges.
> > > >
> > > > **We deeply value your feedback and appreciate the time you took to review our work.**

---

> > > > > ### Comment · Reviewer_H7GT · 2024-11-28
> > > > >
> > > > > Thanks for your response.
> > > > >
> > > > > I still have some concerns. In your response, you mentioned that "the semantic information available in tabular datasets can be quite limited or of low quality," which explains why we cannot rely directly on semantic information. However, does using distances eliminate the influence of low-quality data? Furthermore, measuring similarity based on distances between sample pairs may lack sufficient technical novelty for me. I think more evidence should be provided to verify your point.

---

> > > > > > ### Author Response · Authors · 2024-12-01
> > > > > > **Thanks and further clarify (Part 1)**
> > > > > >
> > > > > > Thank you for your comments and the opportunity to clarify further.
> > > > > >
> > > > > > We'd like to reiterate that "the semantic information available in tabular datasets can be quite limited or of low quality" refers to the inherent challenges in accessing semantic information of attributes in tabular datasets, and those semantic descriptions may with low quality or be ambiguous in some cases. Here are some example datasets from OpenML:
> > > > > >
> > > > > >  - **Limited Semantic Utility**: In datasets like DNA dataset [https://www.openml.org/search?type=data&status=active&id=40670&sort=runs], although the 180 features in DNA dataset have clearly defined semantic labels, each feature is a binary indicator representing a nucleotide symbol that has been encoded. In this case, while the feature semantics are clear and complete, they provide minimal assistance in the tabular prediction task, and cannot be directly used as shared knowledge to aid prediction.
> > > > > >
> > > > > >  - **Confidentiality and Anonymization**: We take Ada dataset [https://www.openml.org/search?type=data&status=active&id=41156&sort=runs] as an example, whose attribute details are concealed for privacy or security reasons, making semantic information inaccessible.
> > > > > >
> > > > > > These examples highlight why we need to consider a method avoids reliance on semantic information.
> > > > > >
> > > > > > **Continued in Part 2...**

---

> > > > > > > ### Author Response · Authors · 2024-12-01
> > > > > > > **Thanks and further clarify (Part 2)**
> > > > > > >
> > > > > > > Measuring similarity based on distances have been applied in classical machine learning tasks such as dimensionality reduction and manifold learning, but we explore the way to pre-train on tabular dataset by leveraing the relationships among instances within a dataset, which effectively serve as the shareable vocabulary across different tabular datasets, regardless of their dimensionality and semantic meanings. The experiments validate the ability of pre-trained TabPTM when compared with other general tabular models.
> > > > > > >
> > > > > > > By leveraging distance-based meta-representation, TabPTM effectively standardizes heterogeneous datasets into a homogeneous form, making it robust to variations or inaccuracies in attribute semantics. To further demonstrate TabPTM's robustness, we conducted experiments on three datasets from the benchmark: BNG(breast-w) (dominated by XGBoost), pendigits (dominated by MLP), and eye_movements_bin (dominated by TabPTM). We generated two types of low-quality datasets:
> > > > > > >  - **Additive Noise**: Noise was added to numeric features with proportions of 0.01 (low), 0.05 (medium), and 0.1 (high), keeping the test set unchanged.
> > > > > > >  - **Concatenated Noise**: Random features were added, with new features comprising 0.1 (low), 0.3 (medium), and 0.5 (high) of the original feature count.
> > > > > > >
> > > > > > > | Datasets                              | XGBoost | MLP    | KNN    | TabPTM |
> > > > > > > |---------------------------------------|---------|--------|--------|--------|
> > > > > > > | BNG(breast-w) [XGBoost dominate]      | 0.9876  | 0.9846 | 0.9837 | 0.9825 |
> > > > > > > | BNG(breast-w)_add_noise_low           | 0.9850  | 0.9836 | 0.9818 | 0.9830 |
> > > > > > > | BNG(breast-w)_add_noise_medium        | 0.9867  | 0.9837 | 0.9817 | 0.9829 |
> > > > > > > | BNG(breast-w)_add_noise_high          | 0.9856  | 0.9836 | 0.9823 | 0.9825 |
> > > > > > > | BNG(breast-w)_new_noise_low           | 0.9865  | 0.9839 | 0.9821 | 0.9827 |
> > > > > > > | BNG(breast-w)_new_noise_medium        | 0.9873  | 0.9827 | 0.9806 | 0.9825 |
> > > > > > > | BNG(breast-w)_new_noise_high          | 0.9868  | 0.9835 | 0.9774 | 0.9830 |
> > > > > > > | pendigits [MLP dominate]              | 0.9914  | 0.9951 | 0.9909 | 0.9932 |
> > > > > > > | pendigits_add_noise_low               | 0.9914  | 0.9941 | 0.9886 | 0.9936 |
> > > > > > > | pendigits_add_noise_medium            | 0.9900  | 0.9941 | 0.9886 | 0.9936 |
> > > > > > > | pendigits_add_noise_high              | 0.9891  | 0.9941 | 0.9886 | 0.9941 |
> > > > > > > | pendigits_new_noise_low               | 0.9895  | 0.9859 | 0.9859 | 0.9918 |
> > > > > > > | pendigits_new_noise_medium            | 0.9882  | 0.9714 | 0.9714 | 0.9923 |
> > > > > > > | pendigits_new_noise_high              | 0.9886  | 0.9563 | 0.9563 | 0.9936 |
> > > > > > > | eye_movements_bin [TabPTM dominate]   | 0.6325  | 0.5708 | 0.5802 | 0.6736 |
> > > > > > > | eye_movements_bin_add_noise_low       | 0.5703  | 0.5729 | 0.5729 | 0.6376 |
> > > > > > > | eye_movements_bin_add_noise_medium    | 0.5749  | 0.5453 | 0.5453 | 0.6148 |
> > > > > > > | eye_movements_bin_add_noise_high      | 0.5887  | 0.5486 | 0.5486 | 0.5994 |
> > > > > > > | eye_movements_bin_new_noise_low       | 0.5972  | 0.5473 | 0.5473 | 0.6570 |
> > > > > > > | eye_movements_bin_new_noise_medium    | 0.5650  | 0.5407 | 0.5407 | 0.6675 |
> > > > > > > | eye_movements_bin_new_noise_high      | 0.5683  | 0.5289 | 0.5289 | 0.6761 |
> > > > > > >
> > > > > > > We have several observations. Regardless of which model dominates the dataset, TabPTM demonstrates strong robustness to noise, maintaining consistent performance. However, other comparison methods may be negatively influenced by the noisy features. This robustness of TabPTM may stem from pre-trained knowledge and the incorporation of mutual information and various metrics when constructing meta-representations.
> > > > > > >
> > > > > > > In summary, TabPTM avoids reliance on semantic information, effectively leverages instance relationships, and demonstrates resilience to noisy data, making it a reliable choice for tabular prediction tasks.
> > > > > > >
> > > > > > > Thanks for your valuable feedback. If you have any other concerns, we would be happy to discuss them promptly.

---

> > > > > > ### Author Response · Authors · 2024-12-02
> > > > > >
> > > > > > Dear reviewer H7GT,
> > > > > >
> > > > > > We appreciate your valuable comments. We have provided a detailed rebuttal, and we hope that you have a chance to read through it. It is worth noting that the other two reviewers have increased their scores by two after reading our rebuttal. We thus respectfully believe that our rebuttal has addressed most if not all of your concerns. If you have additional concerns, we would hope to hear from you soon, so that we can prepare for a response. Thank you.

---

> > > > > > > ### Comment · Reviewer_H7GT · 2024-12-03
> > > > > > >
> > > > > > > Thank you for your response. I truly appreciate the author's efforts and the insights provided. However, after further reflection, I still feel that the approach of measuring distance may not be sufficiently novel in my view. With this in mind, I have decided to maintain my score but adjust my confidence level from 2 to 1.

---

> > > > > > > > ### Author Response · Authors · 2024-12-04
> > > > > > > >
> > > > > > > > Thank you for your feedback.
> > > > > > > >
> > > > > > > > In terms of novelty, we respectfully think its definition is not limited to creating something new, such as new model architectures or objective functions. In our humble opinion, demonstrating that “well-established distance metrics can be repurposed to unify heterogeneous tabular datasets” is novel and significant. We humbly believe it has a profound implication for pre-training in tabular datasets. Specifically, future work may build upon ours to investigate more effective pairwise measures between data instances to further facilitate pre-training.

---

### Official Review · Reviewer_7pPr · 2024-11-04

**Soundness:** 2
**Presentation:** 3
**Contribution:** 2
**Rating:** 5
**Confidence:** 4

**Summary:**

The paper proposes to perform supervised learning on tabular data with a shared model that takes a proposed "meta-representation" as input instead of original sample features.

Samples are represented as distances to prototypes (K the closest samples of particular classes in terms of weighted p-norm distance).

The proposed approach called TabPTM is compared to classical ML models (GBDTs, KNN, SVM) and some contemporary tabular DL models on a set of 36 datasets showing improved performance in the particular setup.

**Strengths:**

- The paper is well written, it clearly describes the method (visualizations also help)
- The goal of learning a shared model for a set of heterogeneous tabular problems is hard and intriguing. This may open up performance and usability gains to tabular models if done right

**Weaknesses:**

The experimental setup is the core weakness of the paper. I expand on main points for improvement below:

- The process behind dataset selection for evals should be more detailed. This is one of the major problems in new method papers in tabular ML: method papers introduce new arbitrary benchmarks with no clear dataset selection criteria making it very hard to assess and compare method performance (many questions regarding dataset quality, baseline performance arise and should be answered when introducing a benchmark)
- The above issue may be minimized by using an established benchmark that is proven in some way (e.g. the benchmark from Grinsztajn et. al. [https://arxiv.org/abs/2207.08815]) -- this also helps ensure comparisons against strong baselines (as established benchmarks often already have results for properly tuned baselines and SoTA algorithms -- see e.g TabR or ModernNCA [https://arxiv.org/abs/2307.14338]).
- Many very relevant baselines are missing (methods that are also heavily relying on the nearest neighbors of the sample)
  - Modern tabular DL versions of the K-NN algorithm that are the current SoTA in the field (ModernNCA [https://arxiv.org/abs/2407.03257], TabR [https://arxiv.org/abs/2307.14338])
  - K-NN based improvements for TabPFN. Very similar ideas stemming from https://arxiv.org/abs/2305.11097 Localized PFN e.g.: https://arxiv.org/abs/2405.16156  https://arxiv.org/abs/2311.10609 https://arxiv.org/abs/2402.11137
  - Properly tuned standard algorithms like XGBoost and MLP. Experimental setup description in the appendix mentions 30 trial hyperparameter budget -- in my experience, methods such as XGBoost and baseline neural networks are considerably undertuned with this budget. You should consider tuning baselines more thoroughly for a fairer comparison. (adopting an established benchmark may help mitigate the costs)
- Ablations should be done on more than two datasets. Tabular datasets are highly diverse, the effect is best measured on a multitude of datasets, improvement on two dataset (that are seemingly arbitrarily chosen) may be not representative (I've seen many ideas and methods shine on a few datasets but fail to generalize and be useful on average)
- Overall, the necessity and usefulness of a shared model is very debatable. The results in table 9 show that fine-tuning is necessary (e.g. no zero-shot generalization is happening), strong nearest-neighbors based baselines are missing, classical baselines are seemingly undertuned and the dataset selection is not justified

**Questions:**

- How does TabPTM compare to existing strong nearest-neighbors based baselines on an established tabular benchmark?
- What is the most compelling evidence **for** the shared model pretraining?

---

> ### Author Response · Authors · 2024-11-25
> **Thanks and Response (Part 1)**
>
> We appreciate your detailed feedback. We are happy that the reviewer recognized the strengths of the paper, suggesting that we are solving an important, valuable, but hard problem. We understand that the main concern leading to a “3” rating is the experimental setup, and we provide more discussions and results as follows.
>
> **Q1.** The process behind dataset selection for evals should be more detailed. … The above issue may be minimized by using an established benchmark … How does TabPTM compare to existing strong nearest-neighbors-based baselines on an established tabular benchmark? More baselines (TabR, ModernNCA, K-NN-based improvements for TabPFN)?
>
> **A1.1: Benchmark selection**
>
> Thank you for your constructive feedback. We agree that many existing tabular works used different (and often not many) datasets, making it hard to compare different methods and assess whether the proposed methods are generalizable enough.
>
> To this end, we chose 72 datasets from UCI and OpenML, the two largest open sources of tabular datasets. In particular, we chose 36 classification and 36 regression datasets. This amount surpasses the scope of the 45 datasets utilized in [https://arxiv.org/abs/2207.08815].
> (We will include more details in the main paper, cf. Lines 417 - 427.)
>
> To further address your concern, we plan to expand our experiments to include the recently established Tabular Benchmark (TabBench) [https://arxiv.org/abs/2407.00956], which comprises 300 datasets. Due to the limited rebuttal period, we first use TabBench’s official ''Tiny Benchmark subset for Rank Consistent Evaluatio'', consisting of 44 datasets that exhibit consistency in method ranking across a larger (300) dataset pool.
>
> **A1.2: Extended Experiment Results with Additional Baselines**
>
> We found that the 44 datasets in TabBench's official Tiny Benchmark do not overlap with the 36 pre-training datasets used in Table 2 and Table 8 of the main text. This allows us to explore a direct transfer of the two pre-trained models in the original TabPTM framework (one trained with 18 classification datasets and one trained with 18 regression datasets) **without additional hyperparameter tuning** and assess them exclusively on the Tiny TabBench subset **with a fixed fine-tuning hyperparameter configuration** (Lines 464-470).
>
> Besides including further datasets, we also follow your suggestion to expand the baselines, including TabR, ModernNCA, TuneTables [https://arxiv.org/pdf/2402.11137], and  LoCalPFN [https://arxiv.org/abs/2406.05207]. For these additional baselines, we strictly follow the training protocol of TabBench. Specifically, the full-shot settings involve hyperparameter optimization via Optuna over 100 trials for each dataset. That is, each pair of dataset and method has its dedicated hyperparameter configuration, sharply contrasting TabPTM’s fine-tuning setting.
>
> (Thank you for providing additional references, especially those aiming to improve TabPFN. We will cite them and discuss them in more detail in our final version. We note that among the K-NN-based baselines, TuneTables was the only method with sufficient reproducibility for inclusion in our study. For TuneTables, we strictly followed its official implementations and used the provided checkpoint prior_diff_real_checkpoint_n_0_epoch_42.cpkt for TuneTables. For LoCalPFN, we set the maximum number of neighbours as K=1000.
> Besides, TabR is also compared in our main text.)
>
> ***Full-shot Performance:***
>
> As shown in Table 1 in the **newly uploaded supplementary material**, TabPTM showed competitive performance in our benchmark comparisons, securing the sixth-highest average rank among 26 meticulously fine-tuned baselines—including XGBoost, CatBoost, FTT, TabR, ModernNCA, TuneTables, and LoCalPFN.
> **It is worth reiterating that these TabPTM results were obtained without additional hyperparameter tuning for each dataset—we simply fine-tuned the pre-trained models in the original submission with a fixed fine-tuning hyperparameter configuration (Lines 464-470).** This demonstrates the excellent general capability of the pre-trained models by TabPTM, which facilitates robust performance with minimal fine-tuning, offering an easier application compared to extensively tuned existing methods.
>
> Compared to K-NN-based improvements for TabPFN, such as TuneTables, TabPTM archives the highest rank. Compared to TabR and ModernNCA, while our TabPTM is ranked slightly behind, it excels on 17/15 out of the 44 datasets, respectively (our TabPTM also ranks better than LocalPFN on 17 of 27  classification datasets since LocalPFN only works on classification tasks), showing its complementary strength.
>
> **Continued in Part 2...**

---

> > ### Author Response · Authors · 2024-11-25
> > **Thanks and Response (Part 2)**
> >
> > ***Few-shot Performance:***
> >
> > We note that, while we did not emphasize this experiment in the main paper, few-shot settings are where pre-training is desperately needed. The recent breakthrough of few-shot learning in vision and language largely results from pre-training.
> > We selected the three largest classification and regression datasets from TabBench for this analysis, so that we can investigate different (sub-sampled) training portions while obtaining results on large enough test portions. TabPTM outperformed other retrieval-based methods in most instances, confirming its superior few-shot generalization capabilities, particularly in data-scarce environments.
> >
> > | Datasets                           | XGBoost  | KNN    | ModernNCA | TabR   | TabPTM  |
> > |------------------------------------|----------|--------|-----------|--------|---------|
> > | BNG(breast-w)-10shot                   | 0.8186   | 0.9632 | 0.9717    | 0.9285 | 0.9612  |
> > | BNG(breast-w)-20shot                    | 0.9236   | 0.9645 | 0.9701    | 0.9689 | 0.9715  |
> > | BNG(breast-w)-50shot                    | 0.9583   | 0.9690 | 0.9715    | 0.9709 | 0.9763  |
> > | Cardiovascular-Disease-dataset-10shot  | 0.5915   | 0.6296 | 0.6499    | 0.6304 | 0.6581  |
> > | Cardiovascular-Disease-dataset-20shot  | 0.5998   | 0.6306 | 0.6545    | 0.6422 | 0.6607  |
> > | Cardiovascular-Disease-dataset-50shot  | 0.6370   | 0.6326 | 0.6600    | 0.6474 | 0.6681  |
> > | FOREX_cadjpy-hour-High-10shot          | 0.5179   | 0.5027 | 0.5019    | 0.5015 | 0.5173  |
> > | FOREX_cadjpy-hour-High-20shot          | 0.5168   | 0.5098 | 0.5224    | 0.5197 | 0.5212  |
> > | FOREX_cadjpy-hour-High-50shot          | 0.5141   | 0.5067 | 0.5184    | 0.5161 | 0.5298  |
> > | fried-10shot                           | 4.775   | 4.361 | 3.955    | 4.016 | 4.214  |
> > | fried-20shot                           | 4.164   | 4.373 | 3.469    | 3.507 | 3.398  |
> > | fried-50shot                           | 3.424   | 3.812 | 2.447    | 3.295 | 3.027  |
> > | house_16H_reg-10shot                   | 53050   | 53040 | 49930    | 53210 | 51930  |
> > | house_16H_reg-20shot                   | 52620   | 53760 | 49530    | 52390 | 44430  |
> > | house_16H_reg-50shot                   | 45880   | 48290 | 49390    | 47370 | 42130  |
> > | law-school-admission-binary-10shot     | 0.2334  | 0.4551| 0.4232   | 0.3851| 0.3832 |
> > | law-school-admission-binary-20shot     | 0.1475  | 0.3767| 0.2387   | 0.3409| 0.2081 |
> > | law-school-admission-binary-50shot     | 0.0962  | 0.3492| 0.2339   | 0.213 | 0.0828 |
> > | **Average Rank**                   | **3.667**| **4.333**| **2.111**  | **3.333**| **1.556**|
> >
> >
> > **Q2.** Ablations should be done on more than two datasets. Tabular datasets are highly diverse, the effect is best measured on a multitude of datasets. What is the most compelling evidence for the shared model pretraining?
> >
> > **A2:** We fully agree with the reviewer that conducting ablation studies on a larger number of datasets is crucial when resources allow. We note that the ablation study presented in the main text (e.g., Table 3) was performed on four datasets. To expand on this, we have conducted additional key experiments from Table 3 on six of the largest datasets selected from TabBench.
> > The most compelling evidence for the effectiveness of shared model pretraining comes from demonstrating its benefits over training separate models on the same meta-representations. To investigate this, we performed the following two compared methods:
> > - Standard TabPTM: TabPTM is trained from scratch on the downstream datasets without pretraining.
> > - XGBoost + MR: XGBoost is trained from scratch on the meta-representations generated from the downstream datasets.
> >
> >
> >
> > **Continued in Part 3...**

---

> > > ### Author Response · Authors · 2024-11-25
> > > **Thanks and Response (Part 3)**
> > >
> > > Below are the results for six datasets:
> > > | Dataset                          | XGBoost + MR | Standard TabPTM | TabPTM    |
> > > |----------------------------------|--------------|-----------------|-----------|
> > > | BNG (breast-w)                   | 0.9811       | 0.9801          | **0.9824**|
> > > | Cardiovascular-Disease-dataset   | 0.7307       | 0.7297          | **0.7359**|
> > > | FOREX_cadjpy-hour-High          | 0.6799       | 0.6722          | **0.6985**|
> > > | fried                            | **1.1928**   | 1.2423          | 1.3263    |
> > > | house_16H_reg                    | 35751        | 35157           | **32760** |
> > > | law-school-admission-binary     | 0.1268       | 0.0129          | **0.0087**|
> > >
> > > The results indicate that the pre-trained TabPTM achieves the best performance in most cases, highlighting the effectiveness of knowledge sharing across tabular datasets. The advantage of XGBoost + MR on the ''fried" dataset may stem from the relatively consistent MR distribution between the training and test sets. In this case, we only need XGBoost to learn fixed rules rather than relying on pre-trained knowledge. Importantly, when we expanded the number of datasets in the ablation study from 4 to 10, the conclusion that pretraining provides benefits remained consistent. We will include the experiments on more datasets in the final version to validate the effectiveness of the pre-trained component in our model.
> > >
> > > **Q3.** classical baselines are seemingly undertuned.
> > >
> > > **A3:** We note that we strictly follow the existing protocol (e.g., the one proposed in FT-T) to tune the classical baselines. We thus think our experimental setup does not under-tune nor over-tune these baselines. We will be happy to discuss this question further if the reviewer can provide more details.
> > >
> > > We greatly value the suggestions you have provided. In Q1/A1, we conducted a comparison based on TabBench. All methods that required hyperparameter tuning were run for 100 iterations. Our approach still outperforms many of the other methods.
> > >
> > > **Thanks again for your constructive feedback, which has played a crucial role in enhancing our research.**

---

> > > > ### Comment · Reviewer_7pPr · 2024-11-29
> > > > **Response**
> > > >
> > > > Thanks for such an extensive response!
> > > >
> > > > I am pleased to see more results and comparisons with relevant baselines.
> > > >
> > > > My concerns were partially addressed by the authors response. But I still have questions and concerns regarding the paper:
> > > > 1. I am not convinced that the stated core contributions and the view of the model as a general model for all datasets holds strongly enough. My last concern stated above: "*overall, the necessity and usefulness of a shared model is very debatable*" still holds true, especially with new DL+kNN baselines in Modern-NCA and kNN+tabpfn demonstrating similar and better performance:
> > > >     - Looking at few-shot performance. Focusing on the 10-shot performance, for example, I see that ModernNCA is very close to the pretrained TapPTM and even winning.
> > > >     - The average rank compared to existing strong baselines on the new benchmark indicates that TabPTM is no better than other KNN+DL solutions. Additionally, comparisons with TabPFN varians should be done on classification datasets (I suspect performance of TabPFN+kNN there would be >= one of TabPTM)
> > > >     - I see that TabPTM is not tuned as the other baselines, but this does not tell me much. It should either be compared to default variants of the baselines explicitly or tuned to obtain maximum performance. Currently it's not an "apples to apples" comparison
> > > > 2. Submission PDF is not changed and the extensive new results in the rebuttal would require a significant rewriting and restructuring. Some examples:
> > > >     - New results should be clearly represented and discussion in the main text
> > > >     - Focus on Few-Shot performance
> > > >
> > > > Overall, I think the additions during rebuttals improved the paper. But there are still questions and concerns. I increased my score to 5 - weak reject to indicate the improvements in methodology and baselines. The paper should discuss related kNN+DL approaches more carefully. The current positioning as a a general universal tabular model is a bit misleading.

---

> > > > > ### Author Response · Authors · 2024-12-01
> > > > > **Thanks and further discussion**
> > > > >
> > > > > Thank you for your valuable comments and for raising the score to indicate the improvements in methodology and baselines. We aim to address your concerns comprehensively below.
> > > > >
> > > > > **1. The Necessity and Usefulness of a Shared Model**
> > > > >
> > > > > We acknowledge that the role of a shared model for tabular datasets is still an area of active exploration. The shared model demonstrates promising performance in some tasks, but local (non-shared) models also achieve better performance in some cases, especially when datasets are large, and computational resources are ample. We think that investigating shared models holds significant potential and may provide insights to the tabular machine learning field due to several reasons:
> > > > >  - **Few-Shot Learning**: Shared models like TabPTM demonstrate competitive performance in few-shot scenarios, where standard models struggle due to limited training data. TabPTM leverages pre-trained knowledge across datasets to boost its performance effectively.
> > > > >  - **Time and Computational Constraints**: In real-world scenarios, time and computational resources are often limited. TabPTM’s lightweight fine-tuning with default hyperparameters outperforms standard methods that require extensive hyperparameter tuning and optimization. This is particularly useful in environments where rapid deployment is essential. The detailed results are listed below in the next part.
> > > > >  - **Robustness to Noise**: The shared model such as TabPTM is able to handle noisy data (e.g., additive or concatenated noise) better than other methods. Its pre-trained knowledge prevents significant degradation in performance, as demonstrated in the latest response to Reviewer H7GT.
> > > > >
> > > > > We will make our claim clearer and discuss the advantages as well as limitations of shared model in the final version of the paper.
> > > > >
> > > > > **2. Fairer Comparison with KNN+DL Solutions**
> > > > >
> > > > > To address concerns about the comparison of TabPTM with baseline methods, we conducted an "apples-to-apples" comparison using default hyperparameters and a maximum of 30 training epochs for all methods, matching TabPTM’s setup. Due to time constraints during the rebuttal phase, we selected first six datasets from TabBench and evaluated top-ranked models in our previous experiments, including MLP, DCNv2, FTT, and TabR. The results are summarized below:
> > > > >
> > > > > | Datasets                         | MLP    | DCNv2  | FTT    | TabR   | TabPTM |
> > > > > |----------------------------------|--------|--------|--------|--------|--------|
> > > > > | FOREX_audchf-day-High            | 0.4699 | 0.5545 | 0.5125 | 0.6278 | 0.6095 |
> > > > > | taiwanese_bankruptcy_prediction  | 0.8069 | 0.8006 | 0.8959 | 0.9101 | 0.9675 |
> > > > > | rl                               | 0.7258 | 0.7020 | 0.7196 | 0.7563 | 0.7857 |
> > > > > | pc3                              | 0.7646 | 0.7671 | 0.7804 | 0.8109 | 0.8978 |
> > > > > | qsar                             | 0.9027 | 0.8760 | 0.8724 | 0.8517 | 0.8768 |
> > > > > | eye_movements_bin                | 0.6097 | 0.6164 | 0.6209 | 0.6523 | 0.6735 |
> > > > > | **AVG RANK**                     | **3.833** | **4** | **3.5** | **2.333** | **1.333** |
> > > > >
> > > > > we find that TabPTM show its effectiveess, which indicate that the shared model may help in the resource limited scenario. We will add more comparison results in the final version of the paper.
> > > > >
> > > > > Additionally, we ranked KNN+DL-related methods across 27 classification datasets in our last response, as the reviewer suggested, and found that TabPTM outperformed other TabPFN+kNN solutions:
> > > > >
> > > > > | Model    | AVG RANK |
> > > > > |----------|----------|
> > > > > | TuneTables | 2.963  |
> > > > > | TabPFN   | 2.407  |
> > > > > | LoCalPFN | 2.481  |
> > > > > | TabPTM   | 2.111  |
> > > > >
> > > > > These results highlight TabPTM’s effectiveness in resource-constrained scenarios and its competitive performance compared to state-of-the-art KNN+DL solutions. We will expand these comparisons in the final version of the paper.
> > > > >
> > > > >
> > > > > **3. Updates to the Submission**
> > > > >
> > > > > While the current stage does not allow for updates to the submission PDF, we commit to incorporating all new results in the final version of the paper. These updates will include:
> > > > >  - Additional experiments with more datasets and baseline models.
> > > > >  - Expanded results focusing on few-shot performance.
> > > > >  - Comprehensive discussions of related KNN+DL approaches.
> > > > >
> > > > > We appreciate the reviewer’s constructive feedback and hope that our responses address the remaining concerns.

---

> > > > > ### Author Response · Authors · 2024-12-02
> > > > >
> > > > > Dear Reviewer 7pPr,
> > > > >
> > > > > We thank you for reading our rebuttal and increasing your score (3 to 5). This is encouraging, indicating that our rebuttal has addressed your major concerns. We have further responded to your latest comments. We hope that you can take a look at your earliest convenience and let us know if you have additional concerns so that we can prepare for a response. Thank you.

---

> > > > > > ### Comment · Reviewer_7pPr · 2024-12-03
> > > > > >
> > > > > > Thanks for the response and clarifications!
> > > > > >
> > > > > > I do believe that additional baselines and datasets improve the paper (that's why I raised the soundness and overall scores).
> > > > > >
> > > > > > However, the two concerns I outlined in the previous response remain. I believe the paper should either *have compelling evidence for the pretraining a shared model* (concrete suggestions below) or should be *significantly* revised to position the method more carefully. As a nearest neighbors based prediction approach with potential for training on multiple datasets. As currently it reads as if TabPTM is a superior general pre-training approach.
> > > > > >
> > > > > > Currently:
> > > > > > - Results on an additonal benchmark are not superior to relevant k-NN based baselines. (Limiting models to 30 epoch training is not an ideal way to ensure fair comparisons, in the note above I've meant the hyperparameter tuning only with no other restrictions on baselines)
> > > > > > - Few shot results are very close, to k-NN and ModernNCA (even though the authors claim that's the methods advantage)
> > > > > > - There should be more analysis on the role of pretraining vs meta representation (e.g. like results for the XGBoost+MR compared to the default XGBoost). This should be done on all datasets, with different pretraining regimes (size and composition of the pre-training datasets, training longer on target datasets with regularization -- pretraining might be viewed as a better initialization that ensures faster convergence, this does not necessarily mean that shared knowledge is encoded in the MLP predictor), furthermore MR could be applied to different parametric models beyond XGBoost and MLP
> > > > > >
> > > > > > Thus, I do not change the score.

---

> > > > > > > ### Author Response · Authors · 2024-12-04
> > > > > > > **Thanks and further rebuttal (Part1)**
> > > > > > >
> > > > > > > Dear Reviewer 7pPr,
> > > > > > >
> > > > > > > We are glad to know that our additional clarifications, baselines, and datasets address your major concerns.
> > > > > > >
> > > > > > > We respond to the remaining concerns as follows.
> > > > > > >
> > > > > > > **Q1.** I believe the paper should either have compelling evidence for the pretraining a shared model (concrete suggestions below) or should be significantly revised to position the method more carefully.
> > > > > > >
> > > > > > > **A1.**
> > > > > > >
> > > > > > > We humbly believe that our results have shown compelling evidence for pre-training, especially under the few-shot downstream settings (the results in our first response). We note that few-shot downstream settings are where pre-training should thrive, as demonstrated in other domains such as computer vision and natural language processing. We also respectfully think, as pre-training has demonstrated remarkable impacts on other domains, exploring its potential in tabular data is a valuable direction.
> > > > > > >
> > > > > > > **Q2.** As currently it reads as if TabPTM is a superior general pre-training approach.
> > > > > > >
> > > > > > >
> > > > > > > **A2.**
> > > > > > >
> > > > > > > We apologize if our writing has created such an impression. We certainly do not claim that TabPTM is the best and only approach to pre-training on heterogeneous tabular data. What we argue in the paper is that pre-training on heterogeneous tabular data is inherently much more challenging than other domains as tabular data lack common vocabularies. We view our approach as an effective way to address it, and future methods may build upon ours to further strengthen it.
> > > > > > >
> > > > > > >
> > > > > > > **Q3.** Limiting models to 30 epoch training is not an ideal way to ensure fair comparisons.
> > > > > > >
> > > > > > >
> > > > > > > **A3.**
> > > > > > >
> > > > > > > We apologize if we misunderstood your request. We want to emphasize it is inherently hard to fairly compare a pre-training-and-fine-tuning approach (ours) to a training-from-scratch approach (others). The former leveraged additional data, already suggesting the comparison cannot be fair. In our humble opinion, a crucial advantage of a pre-training-and-fine-tuning approach is its relative simplicity in the downstream tasks, and we humbly think our results have demonstrated it. Without significantly tuning the hyper-parameter in the downstream tasks but just using a default one, we can already achieve on-par performances with other existing methods. We appreciate your suggestion to tune our approach and others using the same strategy, but we humbly think such results would deviate from our overall goal of creating a general pre-trained model.
> > > > > > >
> > > > > > >
> > > > > > > We note that XTab and TabPFN are also a pre-training-and-fine-tuning approach and our approach notably outperforms them (as shown in the uploaded supplementary pdf and the table in our main paper).
> > > > > > >
> > > > > > > **Q4.** Few shot results are very close, to k-NN and ModernNCA (even though the authors claim that's the methods advantage)
> > > > > > >
> > > > > > >
> > > > > > > **A4.**
> > > > > > >
> > > > > > > If we understand your question correctly, you referred to the table we provided in “Few-shot Performance” on 25 Nov 2024. We want to note that in terms of average rank, our approach (TabPTM) is notably better than k-NN (1.55 vs. 4.33). We also want to note that the table only contains 5 approaches, the average rank thus would take a value within [1, 5]. Compared to ModernNCA, we not only have a better average rank. Across 18 datasets (rows), we achieve better results on 14  rows. That said, we certainly do not claim that our approach is definitely superior to k-NN and ModernNCA on all kinds of datasets, which is impossible according to the legendary no-free-lunch rule in fundamental machine learning.
> > > > > > >
> > > > > > > As a minor point, we note that ModernNCA was put on arXiv on Jul. 3, 2024. According to the ICLR 2025 Reviewer Guide (https://iclr.cc/Conferences/2025/ReviewerGuide), “We consider papers contemporaneous if they are published within the last four months. That means, since our full paper deadline is October 1, if a paper was published (i.e., at a peer-reviewed venue) on or after July 1, 2024, authors are not required to compare their own work to that paper.” Thus, we humbly think demonstrating a dominant performance over ModernNCA is not a requirement for a paper acceptance.
> > > > > > >
> > > > > > > **Continued in Part2...**

---

> > > > > > > > ### Author Response · Authors · 2024-12-04
> > > > > > > > **Thanks and further rebuttal (Part2)**
> > > > > > > >
> > > > > > > > **Q5.** There should be more analysis on the role of pretraining vs meta representation (e.g. like results for the XGBoost+MR compared to the default XGBoost). This should be done on all datasets, with different pretraining regimes (size and composition of the pre-training datasets, training longer on target datasets with regularization -- pretraining might be viewed as a better initialization that ensures faster convergence, this does not necessarily mean that shared knowledge is encoded in the MLP predictor), furthermore MR could be applied to different parametric models beyond XGBoost and MLP.
> > > > > > > >
> > > > > > > >
> > > > > > > > **A5.**
> > > > > > > >
> > > > > > > > We will certainly include more results (i.e., downstream tasks) in the final version. That said, we humbly think the most important experiment is to compare with MR+MLP (i.e., Standard TabPTM). It does not use any pre-training but directly trains an MLP with MR from scratch for each downstream dataset. A comparison between TabPTM (pre-trained) and Standard TabPTM (no pre-training) thus would demonstrate the importance of pre-training. For Standard TabPTM, we use the standard practice (L1292-1294) with longer epochs and weight decays to train it. Our table (in our last response and Table 3/Table 5 in the main paper) does show that TabPTM outperforms Standard TabPTM.
> > > > > > > >
> > > > > > > > In terms of pretraining regimes, we did study different sizes and compositions of the pre-training datasets, in Table 6 in the appendix. We apologize if the results were buried, and we will clarify it in the final version.
> > > > > > > >
> > > > > > > > **Thanks for your valuable time and insights.**

---

### Meta-Review · Area_Chair_4JnN · 2024-12-19

**Metareview:**

This paper proposes a tabular data pre-training method TabPTM, which produces meta-representations for each sample based on the relation of the sample to its closest neighbors for each class (in case of classification) or its overall closest neighbor for regression. All reviewers agree that the goal of learning a homogeneous model for a set of heterogeneous tabular problems is very intriguing. However, as pointed out by Reviewer 7pPr, the necessity of the proposed shared model is debatable. The improvement TabPTM brought compared with previous state-of-the-art methods is not significant. Moreover, as mentioned by Reviewer H7GT, representing each sample with its neighborhood similarity would cause severe semantic loss. Such a ''second-order'' representation would limit the explainability, so the downstream analysis results would be hard to interpret. The slight improvement in performance metrics is not solid enough to prove the significance of TabPTM. Furthermore, I feel the generalization ability of TabPTM is also questionable when the data distribution varies in the training and test data. Given the above concerns, I think this work is slightly below the acceptance threshold of ICLR.

**Additional Comments On Reviewer Discussion:**

All three reviewers had a thorough discussion with the authors. Reviewer 96ub was convinced by the rebuttal and raised the score. However, Reviewer 7pPr and Reviewer H7GT are still concerned about the significance and novelty. After reading the rebuttal and discussions, I agree that the concerns raised by the two reviewers limit the contributions of this work, and thus I give the reject recommendation.

---

### Decision · Program_Chairs · 2025-01-22

Reject